# How to Cure Newton for Unlearning Neural Networks? An Empirical Study from the Hessian Perspective

**Nhung Bui[1*], Xinyang Lu[1*], Rachael Hwee Ling Sim[1], See-Kiong Ng[2], Bryan Kian Hsiang Low[1]**
[1]Department of Computer Science, National University of Singapore, Singapore
[2]Institute of Data Science, National University of Singapore, Singapore
`{btcnhung,xinyang.lu,rachael.sim}@u.nus.edu`
`seekiong@nus.edu.sg, lowkh@comp.nus.edu.sg`
*Equal contribution

## Abstract

Machine unlearning enables AI practitioners to comply with data owners' "Right to be Forgotten" and post-hoc filter sensitive, noisy, or malicious data from trained models. As a theoretically justified algorithm, Newton unlearning is used in previous works to rigorously unlearn selected models, eliminating the need for expensive retraining. However, we found that Newton unlearning is highly sensitive to the Hessian degeneracy phenomenon in trained neural networks, including large language models (LLMs), leading to unlearning performance degradation. To address this challenge, we propose two new unlearning algorithms, CuReNU and CuReNUS, that tackle the Hessian degeneracy in principle based on cubic regularization and discuss their convergence guarantees. As a stochastic variant of CuReNU, CuReNUS offers an efficient second-order unlearning algorithm that is applicable even to the scale of LLMs. We demonstrated that CuReNUS can achieve comparable unlearning performance to state-of-the-art empirical algorithms across diverse settings, including batch and challenging sequential unlearning.

## 1 Introduction

Recent years have witnessed a growing number of machine learning (ML) models trained on *personal data* for widespread applications (Achiam et al., 2023; Radford et al., 2023; EDPB, 2024). As these ML models proliferate, regulatory frameworks are essential for protecting personal data and the rights of their owners. Particularly, the "Right to be Forgotten," which is codified in modern data privacy regulations such as GDPR (2016) and CCPA (2018), enables data owners to request that their personal data be deleted, extending to its *lineage* residing in trained ML models. On the other hand, previous research has shown that ML models can unintentionally leak sensitive training data or become compromised by learning from noisy or malicious data (Carlini et al., 2023; Nasr et al., 2023; Tian et al., 2022). This necessitates post-hoc methods to mitigate the influence of such data on trained models. Consequently, the field of *machine unlearning* (Bourtoule et al., 2021; Cao & Yang, 2015; Nguyen et al., 2022) has emerged to address these legal demands and technical challenges.

The massive scale of modern training datasets and growing model sizes render *retraining* ML models from scratch to facilitate unlearning impractical, e.g., OpenAI's GPT-4 reportedly costs over $100 million to train (Achiam et al., 2023). To address this challenge, machine unlearning seeks to eliminate the influence of the specific training data from trained ML models without the prohibitive cost of retraining (Bourtoule et al., 2021; Nguyen et al., 2022). However, achieving perfect unlearning with provably equivalence to a retrained model is notoriously difficult for neural networks (NNs) due to their iterative learning process and non-linear operations (Bourtoule et al., 2021; Thudi et al., 2022). This has led to the growing adoption of *approximate unlearning* approaches (Ginart et al., 2019), which focus on finding an unlearned model that closely mimics the behavior of a retrained model. While empirical approximate unlearning algorithms exist, such as maximizing the loss on the erased set or continually training with randomized labels (Zhang et al., 2024b; Golatkar et al., 2020), their heuristics often lack rigorous guarantees for truly unlearning, limiting their success in challenging settings like sequential unlearning and unlearning for Large Language Models (LLMs).

Second-order unlearning (Guo et al., 2020; Golatkar et al., 2020; Warnecke et al., 2021) represents a class of approximate unlearning algorithms that leverage both the first-order (gradient) and the second-order information (Hessian) of the loss function to achieve superior unlearning performance. Unlike most empirical approaches, second-order unlearning algorithms are naturally connected to the rigorous concept of influence functions (Koh & Liang, 2017) and often provide theoretical guarantees of fast convergence to the *same loss* as retraining (Guo et al., 2020).[1] Many second-order unlearning algorithms are derived from optimization algorithms, extending unlearning from a static operation to a dynamic, iterative process (Jia et al., 2024) where second-order information accelerates convergence compared to first-order counterparts (Neel et al., 2021; Chien et al., 2024). However, when applied to NNs/LLMs, we found that popular second-order unlearning algorithms exemplified by Newton unlearning (Guo et al., 2020; Golatkar et al., 2020) suffer from a critical problem: the Hessian matrix is severely degenerate near the local optimum.[2] Due to Hessian degeneracy, the Hessian matrix becomes non-invertible in Newton unlearning, making it inapplicable for unlearning NNs. This issue is particularly pronounced in LLMs where the degree of degeneracy is significantly higher, as we will show later in Fig. 1. This motivates our central research question: *How can we unlearn NNs/LLMs effectively with second-order unlearning while addressing the issue of Hessian degeneracy?*

Our analysis further reveals that the Newton unlearning update norm is highly sensitive to the large concentration of small eigenvalues in the Hessian of trained NNs/LLMs. To address this challenge, we leverage the fact that the update norm is a monotonically decreasing function of the Hessian damping factor $\gamma$ and reframe our question: *How can we automatically identify $\gamma$ that warrants effective Newton unlearning for NNs/LLMs?* Based on cubic regularization method in optimization (Nesterov & Polyak, 2006; Tripuraneni et al., 2018), we introduce two new unlearning algorithms, _Cubic-Regularized Newton Unlearning_ (CuReNU) and its stochastic variant (CuReNUS), that can identify the optimal $\gamma$ for effective unlearning without manual hyperparameter tuning. Notably, CuReNUS is a scalable Hessian-free unlearning algorithm[3] that remains computationally viable even at the scale of LLMs. Additionally, CuReNUS maintains a constant memory overhead of $\mathcal{O}(2d)$, where $d$ is the number of model parameters, significantly improving over the $\mathcal{O}(dn)$ complexity of the existing Hessian-free algorithm that scales linearly with the number of training samples $n$ (Qiao et al., 2025).

To summarize, our key contributions are:

1. We define a set of desiderata (effectiveness, efficiency, usability, and practicality) for unlearning algorithms (Sec. 3.1) that existing unlearning algorithms may fail to satisfy. Then, we seek to propose unlearning algorithms that can satisfy all of them;

2. We show that Hessian degeneracy is a fundamental but oft-overlooked issue in Newton unlearning, undermining its success for NNs/LLMs. Moreover, we show that common baselines like Hessian pseudo-inverse and Hessian damping fall short in addressing this issue. (Sec. 4);

3. We formulate a new problem of automatically tuning the Hessian damping factor $\gamma$ and devise two new unlearning algorithms, CuReNU and CuReNUS, that guarantee convergence to the same loss as retraining despite the presence of problematic Hessians. (Sec. 5);

4. We show that CuReNU and CuReNUS can unlock the potential of the vanilla Newton unlearning empirically on different datasets and models. Moreover, CuReNUS is efficient, scalable, and can achieve comparable unlearning performance to state-of-the-art empirical unlearning algorithms, even in challenging settings like sequential unlearning and LLMs unlearning. (Sec. 6).

We note that while our unlearning algorithms are adapted from existing optimization methods, we believe this adaptation is both necessary and non-trivial to address failure modes (i.e., the problematic Hessians) and allow second-order unlearning algorithms to apply successfully to NNs/LLMs. Our novelty thus lies in recognizing the potential of existing optimization methods to satisfy our desiderata, addressing limitations of second-order unlearning, and evaluating the methods extensively.

## 2 RELATED WORKS

**Exact Unlearning.** The goal of exact unlearning is to produce an unlearned model equivalent to a retrained model. Previous works have proposed efficient exact unlearning algorithms for conventional models like support vector machines, random forests, linear regression (Cauwenberghs & Poggio,

---

[1]Achieving the same loss as retraining is a necessary condition for unlearning (see App. L).

[2]Previous works consider linear models with convex losses, where the Hessian is positive semi-definite (Golatkar et al., 2020; Guo et al., 2020). This assumption of convexity often does not hold in NNs/LLMs.

[3]CuReNUS avoids explicit Hessian computation via Hessian-vector products, hence a Hessian-free approach.

2000; Schelter, 2019; Brophy & Lowd, 2021), and selected learning frameworks (Cao & Yang, 2015; Xiong et al., 2023). Bourtoule et al. (2021); Yan et al. (2022) propose model-agnostic, data-centric unlearning algorithms that train an ensemble of models on disjointed data subsets to isolate retraining to a few models. However, ensemble models are expensive to store and potentially compromise performance for NNs with large model sizes and massive training datasets (Zhang et al., 2020).

**Approximate Unlearning.** The goal of approximate unlearning is to produce an unlearned model that mimics the behavior of a retrained model. Common heuristics, such as maximizing the loss on erased set and training models to output random predictions, often lead to *over-forgetting*, which compromises the model's overall utility (Graves et al., 2021; Eldan & Russinovich, 2023; Zhang et al., 2024b). To mitigate this, recent empirical unlearning algorithms often incorporate an additional loss term on the retained set to preserve post-unlearning performance (Kurmanji et al., 2023; Maini et al., 2024). Differently, Guo et al. (2020); Golatkar et al. (2020) propose rigorous unlearning algorithms with unlearning guarantees using the second-order approximation of the retraining loss, which offers stronger unlearning guarantees than the first-order alternatives (Neel et al., 2021; Chien et al., 2024). From an efficiency perspective, localization techniques are often integrated with existing unlearning algorithms to restrict unlearning updates to a subset of salient parameters, thereby enabling unlearning in large-scale models (Goel et al., 2022; Yu et al., 2023; Jia et al., 2023).

**Second-Order Unlearning.** Second-order unlearning leverages both the first-order (gradient) and second-order information (Hessian) of the loss function to better approximate retraining loss and achieve more effective unlearning. Guo et al. (2020); Golatkar et al. (2020) proposed Newton-like updates to unlearn linear models under convex losses, where the Hessian is positive semi-definite. Warnecke et al. (2021) utilized influence functions (Koh & Liang, 2017) to selectively unlearn specific features and labels. Recently, Jia et al. (2024) bridged the gap between second-order optimization and classical influence functions, providing a unified perspective on these approaches. Despite their effectiveness, most second-order algorithms are hindered by the prohibitive cost of Hessian computation and storage. While Hessian approximations offer a practical workaround (Golatkar et al., 2020; Jia et al., 2024), they are prone to error accumulation in challenging sequential unlearning settings, leading to performance degradation. In contrast, CuReNUS offers a scalable Hessian-free unlearning algorithm that bypasses explicit Hessian computation/approximation through the use of Hessian-vector products (HVPs). Compared to the existing Hessian-free unlearning algorithm (Qiao et al., 2025) that incurs $\mathcal{O}(dn)$ space complexity, CuReNUS requires only $\mathcal{O}(2d)$ memory, where $n$ and $d$ denote the number of training samples and parameters, respectively. These advantages ensure that CuReNUS remains computationally viable even at the scale of LLMs.

## 3 PRELIMINARIES

### 3.1 PROBLEM FORMULATION

Let $D = \{(\mathbf{x}_i, y_i)\}_{i=1}^n \subseteq \mathcal{X} \times \mathcal{Y}$ denote the training set of $n$ samples, where $\mathbf{x}_i \in \mathcal{X}$ is the input and $y_i \in \mathcal{Y}$ is the corresponding target. Let $D_e \subseteq D$ denote the *erased set* of $n_e$ samples to be unlearned and $D_r = D \setminus D_e$ denote the *retained set* of $n_r$ remaining samples. Let $f_{\mathbf{w}^*} : \mathcal{X} \rightarrow \mathcal{Y}$, with parameters $\mathbf{w}^* \in \mathbb{R}^d$, denote the *original model* trained on $D$. The goal of machine unlearning is to remove the *lineage* of $D_e$ from $f_{\mathbf{w}^*}$ while preserving the post-unlearning performance on $D_r$. Retraining the model solely on $D_r$ can achieve this goal *exactly*; however, retraining is infeasible for large models as it requires many iterations over massive $D_r$ and scales poorly with model sizes. Therefore, the goal of an unlearning algorithm $U$ is to return an unlearned model that closely mimics the behavior of a retrained model. We provide a more formal definition of unlearning in App. K.

**Desiderata.** We define the desiderata for an effective unlearning algorithm $U$ as follows, with a detailed comparison of existing unlearning algorithms against these desiderata provided in App. A.

**D1 Effectiveness:** $U$ should approximate the performance of retraining, the goal standard for perfect unlearning, across both $D_e$ and $D_r$. This motivates our preference for second-order algorithms over first-order alternatives, as second-order information enables a higher-fidelity approximation of the retraining loss and allows us to achieve more effective unlearning. [4]

**D2 Efficiency:** $U$ should be significantly faster than retraining. While optimizing time efficiency may require storing additional statistics, $U$ should not require significantly more storage than retraining.

---

[4]On the contrary, we observed that first-order unlearning algorithms like gradient descent (GD) exhibit *under-forgetting*, as they fail to sufficiently remove the *lineage* of the erased set $D_e$ in Sec. 6.

This desideratum ensures that $U$ is applicable for unlearning large models, such as LLMs, and can efficiently handle repeated unlearning requests, such as sequential unlearning.

**D3 Usability:** To maximize usability, $U$ should require minimal hyperparameter tuning to achieve effective unlearning. We demonstrate later in Sec. 5 that CuReNU and CuReNUS can automatically identify the optimal damping factor for Newton unlearning, eliminating the need for manual hyperparameter tuning while delivering strong unlearning performance.

**D4 Practicality:** $U$ should maintain its effectiveness and efficiency across diverse real-world deployment settings. These settings may include (i) *batch unlearning*, where unlearning requests are periodically aggregated to minimize unlearning cost, and (ii) *sequential unlearning*, where unlearning requests arrive as a continuous stream. The latter is particularly challenging, as the model may accumulate errors, such as *over-forgetting* (degrading model performance on $D_r$) or *under-forgetting* (failing to sufficiently forget $D_e$), across successive unlearning rounds, leading to unlearning performance degradation.

### 3.2 Assumptions & Notations

We assume $\mathbf{w}^*$ is optimized via the common empirical risk minimization framework: $\mathbf{w}^* = \arg\min_{\mathbf{w} \in \mathbb{E}^d} \mathcal{L}(\mathbf{w}; D)$, where $\mathcal{L}(\mathbf{w}; D) \triangleq \frac{1}{|D|} \sum_{(\mathbf{x}_i, y_i) \in D} \ell(f_{\mathbf{w}}(\mathbf{x}_i), y_i)$ is the average loss over samples in $D$ under the loss function $\ell$. Moreover, we make the following assumption for $\mathcal{L}$.

**Assumption 3.1.** $\mathcal{L}$ *is twice continuously differentiable with respect to (w.r.t.)* $\mathbf{w}$.

For brevity, we use $\mathbf{g_w} \triangleq \nabla_{\mathbf{w}} \mathcal{L} \in \mathbb{R}^d$ and $\mathbf{H_w} \triangleq \nabla_{\mathbf{w}}^2 \mathcal{L} \in \mathbb{R}^{d \times d}$ to denote the gradient and the Hessian of $\mathcal{L}$ w.r.t. parameters $\mathbf{w}$. We will make additional assumptions in CuReNU and CuReNUS.

**Assumption 3.2** ($\rho$-Lipschitz gradient). *For some* $\rho > 0$, $\|\mathbf{g_w} - \mathbf{g_{w'}}\| \le \rho \|\mathbf{w} - \mathbf{w'}\|, \forall \mathbf{w}, \mathbf{w'} \in \mathbb{R}^d$.

**Assumption 3.3** ($L$-Lipschitz Hessian). *For some* $L > 0$, $\|\mathbf{H_w} - \mathbf{H_{w'}}\| \le L \|\mathbf{w} - \mathbf{w'}\|, \forall \mathbf{w}, \mathbf{w'} \in \mathbb{R}^d$.

We provide justification for Assumptions 3.1, 3.2, and 3.3 in App. B.

**Notations.** When necessary, we use superscripts $D$, $D_e$, or $D_r$ for $\mathbf{g_w}$ and $\mathbf{H_w}$ to denote the data subsets over which the loss is averaged when evaluating the gradient and Hessian. Notably, for the Hessian $\mathbf{H_w}^{D_r}$, we use $\{\lambda_1, \ldots, \lambda_d\}$ to denote its eigenvalues and $\{\mathbf{u}_1, \ldots, \mathbf{u}_d\}$ to denote its corresponding eigenvectors, where $\lambda_1 \ge \cdots \ge \lambda_d$ (sorted in non-increasing order). For any symmetric matrix $\mathbf{A}$, we write $\mathbf{A} \succ \mathbf{0}$ (or $\succeq \mathbf{0}$) to denote $\mathbf{A}$ is positive definite (or positive semi-definite), i.e., $\mathbf{z}^T \mathbf{A} \mathbf{z} > 0$ (or $\ge 0$) for all $\mathbf{z} \ne \mathbf{0}$. By default, $\|\cdot\|$ refers to the Euclidean norm.

### 3.3 Newton Unlearning

Here, we describe the Newton unlearning algorithm proposed in previous works (Guo et al., 2020; Warnecke et al., 2021). The benefits of Newton unlearning algorithm and its variants over first-order unlearning algorithms are further discussed through the lens of optimization in App. M.

The Newton unlearning algorithm involves multiple iterations, starting from $\mathbf{w}_0 = \mathbf{w}^*$, where $\mathbf{w}_t$ denotes the parameters at iteration $t$. For brevity, we use $\Delta_{t+1} \triangleq \mathbf{w}_{t+1} - \mathbf{w}_t$ to denote the parametric difference between two consecutive iterations. Recall that retraining aims to achieve the minimal loss $\mathcal{L}(\mathbf{w}; D_r)$ on the retained set, or *retraining loss*. Under Assumption 3.1, Newton unlearning seeks $\mathbf{w}_{t+1}$ that minimizes the following quadratic approximation of $\mathcal{L}(\mathbf{w}_{t+1}; D_r)$ around $\mathbf{w}_t$:

$$\min_{\mathbf{w}_{t+1}} \left[ \tilde{\mathcal{L}}(\mathbf{w}_{t+1}; D_r) \triangleq \mathcal{L}(\mathbf{w}_t; D_r) + \langle \mathbf{g}_{\mathbf{w}_t}^{D_r}, \Delta_{t+1} \rangle + \tfrac{1}{2} \langle \mathbf{H}_{\mathbf{w}_t}^{D_r} \Delta_{t+1}, \Delta_{t+1} \rangle \right]. \tag{1}$$

By solving the first-order necessary condition $\nabla_{\mathbf{w}_{t+1}} \tilde{\mathcal{L}}(\mathbf{w}_{t+1}; D_r) = \mathbf{0}$, we get the Newton update:

$$\mathbf{w}_{t+1} = \mathbf{w}_t - (\mathbf{H}_{\mathbf{w}_t}^{D_r})^{-1} \mathbf{g}_{\mathbf{w}_t}^{D_r}. \tag{2}$$

The vanilla Newton unlearning algorithm repeatedly applies the Newton update (Eq. 2) for $T$ iterations or until a stopping criterion (e.g., sufficiently small retraining loss) is met.

**Computation with $D_e$.** The Newton update can be computed with $D$ and $D_e$ through the following equations: $\mathbf{H}_{\mathbf{w}_t}^{D_r} = \frac{n}{n_r} \cdot \mathbf{H}_{\mathbf{w}_t}^{D} - \frac{n_e}{n_r} \cdot \mathbf{H}_{\mathbf{w}_t}^{D_e}$ and $\mathbf{g}_{\mathbf{w}_t}^{D_r} = \frac{n}{n_r} \cdot \mathbf{g}_{\mathbf{w}_t}^{D} - \frac{n_e}{n_r} \cdot \mathbf{g}_{\mathbf{w}_t}^{D_e}$. If $\mathbf{w}^*$ is a stationary point, we can further simplify $\mathbf{g}_{\mathbf{w}_0}^{D_e} = -\frac{n_e}{n_r} \cdot \mathbf{g}_{\mathbf{w}_0}^{D_e}$ since $\mathbf{g}_{\mathbf{w}_0}^{D} = \mathbf{0}$. Nonetheless, we justify our choice of minimizing the retraining loss $\mathcal{L}(\mathbf{w}; D_r)$ for unlearning $D_e$ in App. L.

Figure 1: **Left:** Hessian eigenspectrum for CNN $\times$ FMNIST; **Middle:** Hessian rank dynamics during training for CNN $\times$ FMNIST; **Right:** Empirical Hessian eigenspectrum density for Llama-2 $\times$ TOFU.

## 4 PROBLEMATIC HESSIANS IN NEURAL NETWORKS

Newton unlearning (Sec. 3.3) assumes that $\mathbf{H}_{\mathbf{w}_t}^{D_r}$ is invertible at $\mathbf{w}_t$, i.e., full rank with no zero eigenvalues. However, this assumption rarely holds for trained NNs/LLMs characterized by highly non-convex loss landscapes. In our experiments with CNN $\times$ FMNIST, the Hessian eigenspectrum after training shows many *zero* and near-zero eigenvalues (Fig. 1, left), and the Hessian rank rapidly diminishes as training converges (Fig. 1, middle), reflecting the increasing number of zero eigenvalues during training. We observe a similarly zero-concentrated Hessian eigenspectrum for Llama-2 $\times$ TOFU (Fig. 1, right), confirming that the invertability assumption is often invalid in practice.

Previous works have shown that Hessian degeneracy is a fundamental issue during training neural networks across various datasets (Sagun et al., 2017; Papyan, 2018; Ghorbani et al., 2019). A standard model for describing Hessian eigenspectra in deep neural networks is the so-called *spiked model*, which features a large concentration of eigenvalues near $0$ (the *bulk*), and a few isolated, large eigenvalues well separated from the bulk (the *spikes*) (Johnstone, 2001; Sagun et al., 2016; 2017). The Hessian rank deficiency (hence Hessian degeneracy) is further explained in Singh et al. (2021; 2023) to be closely connected to the effective number of model parameters that naturally decreases as training converges, especially in the over-parameterization paradigm. Besides, it is not uncommon for neural networks to converge to saddle points (Dauphin et al., 2014), where the Hessian contains negative eigenvalues, as also shown in our experiment with Llama-2 $\times$ TOFU (Fig. 1, right). The presence of negative eigenvalues can further undermine the convergence of Newton unlearning.

We summarize these properties of Hessians in trained NNs/LLMs in the following observation.

**Observation 4.1.** $\mathbf{H}_{\mathbf{w}_t}^{D_r}$ *is degenerate with many zero and possibly negative eigenvalues, i.e., there exists $k \ll d$ s.t. $\lambda_i > 0$ for $i \leq k$ and $\lambda_j \leq 0$ for $j \geq k+1$.*

**Baselines.** Due to Obs. 4.1, the Hessian is non-invertible (with potentially negative eigenvalues), rendering the vanilla Newton unlearning inapplicable. In practice, we often employ the following baselines to tackle the problematic Hessians: (1) replacing the exact inverse with the pseudo-inverse (**PINV-Newton**), and (2) adding a small diagonal matrix to the Hessian (**Damped Newton**).

**(1) Pseudo-Inverse (PINV-Newton).** The inverse of the degenerate Hessian is replaced by its unique pseudo-inverse $(\mathbf{H}_{\mathbf{w}_t}^{D_r})^\dagger$ that always exists. Applying the pseudo-inverse yields the minimum-norm solution $\Delta_{t+1}$ to the linear system $\mathbf{H}_{\mathbf{w}_t}^{D_r}\Delta_{t+1} = \mathbf{g}_{\mathbf{w}_t}^{D_r}$, specifically $\Delta_{t+1} = (\mathbf{H}_{\mathbf{w}_t}^{D_r})^\dagger \mathbf{g}_{\mathbf{w}_t}^{D_r}$.

**Remark 4.2.** *The squared norm of the Newton update using pseudo-inverse Hessian is $\|\Delta_{t+1}\|^2 = \sum_{i:\lambda_i \neq 0} \frac{1}{\lambda_i^2}(\mathbf{u}_i^T \mathbf{g}_{\mathbf{w}_t}^{D_r})^2$ (see App. C.1 for the derivation). As noted in Obs. 4.1, the presence of numerous near-zero eigenvalues $|\lambda_i| \approx 0$ can lead to $\|\Delta_{t+1}\|^2 \gg 0$.*

**(2) Damping (Damped Newton).** The degenerate Hessian is *damped* by adding a small diagonal matrix, i.e., $\mathbf{H}_{\mathbf{w}_t}^{D_r} + \gamma\mathbf{I}$, where $\gamma > \max\{0, -\lambda_d\}$ is the damping factor and $\mathbf{I}$ is the $d$-dimensional identity matrix. Formally, damping is equivalent to finding $\Delta_{t+1}$ that minimizes the *regularized* linear least squares $\|\mathbf{H}_{\mathbf{w}_t}^{D_r}\Delta_{t+1} - \mathbf{g}_{\mathbf{w}_t}^{D_r}\|^2 + \gamma\|\Delta_{t+1}\|^2$, that is $\Delta_{t+1} = (\mathbf{H}_{\mathbf{w}_t}^{D_r} + \gamma\mathbf{I})^{-1}\mathbf{g}_{\mathbf{w}_t}^{D_r}$. In practice, a small $\gamma$ is often chosen to minimize the distortion of the original second-order information.

**Remark 4.3.** *The squared norm of the Newton update using damped Hessian is $\|\Delta_{t+1}\|^2 = \sum_{i=1}^{d} \frac{1}{(\gamma+\lambda_i)^2}(\mathbf{u}_i^T \mathbf{g}_{\mathbf{w}_t}^{D_r})^2$ (see App. C.2 for the derivation). If $\gamma$ is too small, given the presence of numerous zero eigenvalues (Obs. 4.1), then $\|\Delta_{t+1}\|^2 \gg 0$.*

Remarks 4.2 and 4.3 show that both PINV-Newton and Damped Newton are prone to produce excessively large-norm updates. This behaviour leads to overshooting local minima and subsequent unlearning performance degradation (violating **D1**), as evidenced by our experiments in Sec. 6.2.

| Unlearning Algorithm | Convergence Rate | Guarantee | Proof |
|---|---|---|---|
| GD | $\mathcal{O}(\varepsilon^{-2})$ | $\varepsilon$-FOSP | Nesterov (2013) |
| SGD | $\mathcal{O}(\varepsilon^{-4})$ | $\varepsilon$-FOSP | Khaled & Richtárik (2020) |
| Newton | local quadratic | $\varepsilon$-FOSP | Nocedal & Wright (2006) |
| CuReNU | global $\mathcal{O}(\varepsilon^{-1.5})$ | $\varepsilon$-SOSP | App. D.1 |
| CuReNUS | global $\tilde{\mathcal{O}}(\varepsilon^{-3.5})$ | $\varepsilon$-SOSP | App. E.1 |

Table 1: **Convergence guarantees of various unlearning algorithms under non-convex losses.** While standard first-order algorithms (i.e., GD, SGD) only guarantee convergence to an $\varepsilon$-FOSP, CuReNU and CuReNUS achieve $\varepsilon$-SOSP convergence with superior rates.

However, Remark 4.3 also indicates that the update norm is monotonically decreasing w.r.t. $\gamma$. We posit that an effective Newton unlearning for NNs/LLMs should strike a balance: $\gamma$ must be large enough to avoid overly large-norm updates, yet small enough to avoid trivial updates and cause slow convergence. This motivates a question: *How can we automatically find an optimal $\gamma$ (**D3**) that warrants effective Newton unlearning for NNs/LLMs, satisfying **D1**?*

## 5 METHODOLOGY

Here, we describe the methodology and theoretical guarantees of CuReNU and CuReNUS that will be of interest to the ML audience, with an emphasis on their applications in unlearning. While there may be less novelty in this section, our algorithms offer principled and scalable solutions to overcome the problematic Hessians, supported by strong empirical results in the next section.

**Convergence guarantees.** Finding the local minima of retraining loss $\mathcal{L}(\mathbf{w}; D_r)$ is challenging for NNs/LLMs with highly non-convex losses. Therefore, it is often helpful to consider two relaxed definitions: $\varepsilon$-first-order stationary points ($\varepsilon$-FOSPs) and $\varepsilon$-second-order stationary points ($\varepsilon$-SOSPs).

**Definition 5.1.** *An $\varepsilon$-FOSP $\mathbf{w}$ of the function $\mathcal{L}$ satisfies $\|\mathbf{g_w}\| \leq \varepsilon$.*

**Definition 5.2.** *An $\varepsilon$-SOSP $\mathbf{w}$ of the function $\mathcal{L}$ (with $L$-Lipschitz Hessian) satisfies $\|\mathbf{g_w}\| \leq \varepsilon$ and the minimum eigenvalue of the Hessian $\lambda_{min}(\mathbf{H_w}) \geq -\sqrt{L\varepsilon}$.*

Tab. 1 summarizes the convergence guarantees of different unlearning algorithms for non-convex losses. Both CuReNU and CuReNUS provide convergence to an $\varepsilon$-SOSP, which is a stronger guarantee than the $\varepsilon$-FOSP offered by first-order algorithms like GD and SGD.[5] Moreover, CuReNU and CuReNUS provide global convergence guarantees, which are better than local convergence in Newton unlearning (that may even diverge due to degenerate Hessians). These stronger guarantees indicate that our unlearning algorithms can optimize retraining loss effectively, setting them apart from empirical unlearning algorithms such as Kurmanji et al. (2023); Zhou et al. (2025).

### 5.1 CUBIC-REGULARIZED NEWTON UNLEARNING (CuReNU)

Under Assumption 3.3, we consider the minimization problem of the cubic-regularized approximation (Nesterov & Polyak, 2006) of $\mathcal{L}(\mathbf{w}_{t+1}; D_r)$ around $\mathbf{w}_t$:

$$\min_{\mathbf{w}_{t+1}} \left[ \tilde{\mathcal{L}}(\mathbf{w}_{t+1}; D_r) \triangleq \mathcal{L}(\mathbf{w}_t; D_r) + \langle \mathbf{g}_{\mathbf{w}_t}^{D_r}, \Delta_{t+1} \rangle + \tfrac{1}{2} \langle \mathbf{H}_{\mathbf{w}_t}^{D_r} \Delta_{t+1}, \Delta_{t+1} \rangle + \tfrac{L}{6} \|\Delta_{t+1}\|^3 \right]. \quad (3)$$

Notably, $\tilde{\mathcal{L}}(\mathbf{w}_{t+1}; D_r)$ serves as a global upper bound of $\mathcal{L}(\mathbf{w}_{t+1}; D_r)$, which allows CuReNU to achieve the global convergence (see App. D.1). Such a guarantee cannot be achieved via the standard quadratic approximation (Eq. 1) used in vanilla Newton unlearning. However, unlike Newton unlearning, the problem in Eq. 3 cannot be solved directly using the first-order necessary condition.[6] We therefore consider its strong dual form by introducing the dual variable $\alpha_{t+1} \triangleq \|\mathbf{w}_{t+1} - \mathbf{w}_t\|$:

$$\sup_{\alpha_{t+1}} \xi(\alpha_{t+1}) \triangleq -\tfrac{1}{2} \left\langle \left( \mathbf{H}_{\mathbf{w}_t}^{D_r} + \tfrac{L}{2}\alpha_{t+1}\mathbf{I} \right)^{-1} \mathbf{g}_{\mathbf{w}_t}^{D_r}, \mathbf{g}_{\mathbf{w}_t}^{D_r} \right\rangle - \tfrac{L}{12}\alpha_{t+1}^3$$

$$\text{such that} \quad \alpha_{t+1} \in \mathcal{Q} \triangleq \{ \alpha \in \mathbb{R} : \mathbf{H}_{\mathbf{w}_t}^{D_r} + \tfrac{L}{2}\alpha\mathbf{I} \succ 0, \alpha \geq 0 \}. \tag{4}$$

The key observation here is that Eq. 4 represents a *convex* constrained optimization problem in $\alpha_{t+1}$. Consequently, it can be solved efficiently using standard off-the-shelf optimization algorithms, such as trust-region methods (Conn et al., 2000). Importantly, $\alpha_{t+1}$ implicitly defines the optimal damping

---

[5]An $\varepsilon$-SOSP with small $\varepsilon$ helps avoid most saddle points and sharp local maxima, which an $\varepsilon$-FOSP cannot.

[6]Doing so yields an ill-defined update $\mathbf{w}_{t+1} = \mathbf{w}_t - \left( \mathbf{H}_{\mathbf{w}_t}^{D_r} + \tfrac{L}{2}\|\Delta_{t+1}\|\mathbf{I} \right)^{-1} \mathbf{g}_{\mathbf{w}_t}^{D_r}$ since $\|\Delta_{t+1}\|$ is unknown.

factor $\gamma$ for the degenerate Hessians via the relation $\gamma = \frac{L}{2}\alpha_{t+1}$. With the optimal $\alpha_{t+1}$ (hence the corresponding $\gamma$), CuReNU repeatedly applies the following update for $T$ iterations:

$$\mathbf{w}_{t+1} = \mathbf{w}_t - \left(\mathbf{H}_{\mathbf{w}_t}^{D_r} + \frac{L}{2}\alpha_{t+1}\mathbf{I}\right)^{-1}\mathbf{g}_{\mathbf{w}_t}^{D_r}. \tag{5}$$

We detail how to solve $\alpha_{t+1}$ using trust-region methods and discuss the duality of Eq. 3 and Eq. 4 in App. D. The pseudocode of CuReNU with trust-region solvers is provided in Algo. 1. Moreover, we prove that CuReNU converges to an $\varepsilon$-SOSP of the retraining loss in $\mathcal{O}(\varepsilon^{-1.5})$ iterations in App. D.1.

**Complexity Analysis.** Although CuReNU has a fast convergence guarantee, it incurs prohibitive space and time complexity, thereby violating **D2**. Specifically, CuReNU requires storing the explicit Hessian matrix, which costs $\mathcal{O}(d^2)$ space. CuReNU also involves multiple non-trivial computations, including Hessian and its inverse that costs $\mathcal{O}(nd^2 + d^3)$ time, and the Hessian smallest eigenvalue (as $\gamma$ initialization, see Algo. 1) that costs $\mathcal{O}(kC)$ time even when using Lanczos method, where $k \leq d$ is the number of Lanczos iterations and $C$ is the cost of computing an HVP per iteration. We note that $C$ depends on the number of samples over which the Hessian is defined, which is $n_r$ in this case. The high complexity of CuReNU raises a critical question: *Can we improve the efficiency of CuReNU while preserving its convergence guarantee and ensuring scalability to large NNs?*

## 5.2 Cubic-Regularized Newton Unlearning - Stochastic variant (CuReNUS)

Here, we consider a more efficient implementation of CuReNU based on stochastic approximation of the cubic regularization (Tripuraneni et al., 2018). Let $\mathbf{g}_{\mathbf{w}}^{B_1}$ and $\mathbf{H}_{\mathbf{w}}^{B_2}$ denote a stochastic gradient and Hessian evaluated on two mini-batches $B_1, B_2 \subset \mathcal{D}_r$ with sizes $n_1$ and $n_2$.[7] Under Assumptions 3.2 and 3.3, we seek $\mathbf{w}_{t+1}$ that minimizes the following stochastic approximation of Eq. 3:

$$\min_{\mathbf{w}_{t+1}}\left[\tilde{\mathcal{L}}^{sto}(\mathbf{w}_{t+1}; D_r) \triangleq \mathcal{L}(\mathbf{w}_t; D_r) + \langle\mathbf{g}_{\mathbf{w}_t}^{B_1}, \Delta_{t+1}\rangle + \frac{1}{2}\langle\mathbf{H}_{\mathbf{w}_t}^{B_2}\Delta_{t+1}, \Delta_{t+1}\rangle + \frac{L}{6}\|\Delta_{i+1}\|^3\right]. \tag{6}$$

While alternative methods exist, solving this problem via gradient descent (GD) is particularly appealing because (1) the gradient of $\tilde{\mathcal{L}}^{sto}(\mathbf{w})$ enables efficient computation via HVPs, and (2) previous works have shown that with an appropriate learning rate, GD is an effective stochastic optimization algorithm for both convex (Bottou et al., 2018; Duchi, 2018) and non-convex functions (LeCun et al., 2015). However, the vanilla GD may fail at the so-called "hard case" (Conn et al., 2000): when $\lambda_d < 0$ and $\langle\mathbf{u}_d, \mathbf{g}_{\mathbf{w}_t}^{B_1}\rangle = 0$,[8] then $\mathbf{g}_{\mathbf{w}_t}^{B_1}$ always remains in a subspace orthogonal to $\mathbf{u}_d$, while the optimal parametric gap $\Delta_{t+1}^*$ (the global minimizer of the approximation defined in Eq. 6) can yield $\langle\mathbf{u}_d, \Delta_{t+1}^*\rangle \neq 0$.[9] To mitigate this, it is common to slightly perturb the gradient, i.e., $\tilde{\mathbf{g}}_{\mathbf{w}_t}^{B_1} = \mathbf{g}_{\mathbf{w}_t}^{B_1} + \sigma\zeta$ where $\sigma > 0$ and $\zeta \sim \text{Unif}(\mathbb{S}^{d-1})$. In practice, we often choose a small $\sigma$ ($\sigma < 1$) to preserve most of the first-order information, although we show that CuReNUS remains effective across varying $\sigma$ in App. J.2. With a learning rate $\eta$, the $s$-th iteration of GD is:

$$\Delta_{s+1} = \Delta_s - \eta\left[\tilde{\mathbf{g}}_{\mathbf{w}_t}^{B_1} + \mathbf{H}_{\mathbf{w}_t}^{B_2}\Delta_s\right]. \tag{7}$$

Given $T_{inner}$ is the number of GD iterations, CuReNUS applies the update $\mathbf{w}_{t+1} = \mathbf{w}_t + \Delta_{T_{inner}}$. Subsequently, it samples new $B_1$ and $B_2$ mini-batches and repeats the same process for $T_{outer}$ stochastic iterations or until a stopping criterion is met. We show that using a larger $T_{outer}$ often correlates with better unlearning performance in App. J.3, while a small $T_{inner}$ (around 5-10) is sufficient in most of our experiments. Algo. 2 provides the pseudocode for CuReNUS. Additionally, we show that CuReNUS converges to an $\varepsilon$-SOSP of the retraining loss in $\tilde{\mathcal{O}}(\varepsilon^{-3.5})$ stochastic gradient/HVP evaluations in App. E.1, where $\tilde{\mathcal{O}}$ hides logarithmic factors.

**Complexity Analysis.** Compared to CuReNU that costs $\mathcal{O}(2d)$ space and the Hessian-free unlearning algorithm in Qiao et al. (2025) that costs $\mathcal{O}(nd)$ space, CuReNUS significantly reduces the memory overhead to $\mathcal{O}(2d)$, which includes $O(d)$ for a gradient vector and $O(d)$ for an HVP. An HVP can be computed efficiently using Pearlmutter's trick (Pearlmutter, 1994) in a comparable time, denoted as $C$,

---

[7]Using different batches helps decorrelate the errors from stochastic estimates, which improves stability and convergence of CuReNUS in practice.

[8]Here, we abuse notation and use $\lambda_d$ and $\mathbf{u}_d$ to denote the smallest eigenvalue and eigenvector of $\mathbf{H}_{\mathbf{w}_t}^{B_2}$.

[9]We refer readers to Carmon & Duchi (2019); Bellavia et al. (2023) for more details.

Table 2: Sample-level and class-level batch unlearning on CNN × FMNIST (averaged over 3 random runs). "→" means closer to retraining is better; "↑" means higher is better; "↓" means lower is better. We use **boldface** to denote best results and underline to denote second-best results.

| Method | Sample-Level Unlearning | | | | | | Class-Level Unlearning | | | | | |
|---|---|---|---|---|---|---|---|---|---|---|---|---|
| | $D_e$ Acc. (→) | $D_r$ Acc. (→) | $D_{test}$ Acc. (→) | ToW (↑) | JS Div. (↓) | MIA (→) | $D_e$ Acc. (→) | $D_r$ Acc. (→) | $D_{test}$ Acc. (→) | ToW (↑) | JS Div. (↓) | MIA (→) |
| Retraining | 85.43 ± 0.32 | 87.41 ± 0.51 | 84.88 ± 0.44 | 1.00 ± 0.00 | 0.000 ± 0.00 | 50.40 ± 0.11 | 0.00 ± 0.00 | 91.21 ± 0.79 | 81.30 ± 0.63 | 1.00 ± 0.00 | 0.000 ± 0.00 | 51.38 ± 0.74 |
| Original | 88.85 ± 0.17 | 88.89 ± 0.04 | 87.84 ± 0.22 | 0.92 ± 0.01 | **0.001 ± 0.00** | 50.70 ± 0.14 | 85.96 ± 0.75 | 89.86 ± 0.82 | 88.38 ± 0.55 | 0.13 ± 0.01 | 0.021 ± 0.00 | 51.69 ± 0.60 |
| Rand. Lbls. | 88.31 ± 0.40 | 88.30 ± 0.39 | 87.36 ± 0.47 | 0.94 ± 0.01 | **0.001 ± 0.00** | 50.73 ± 0.12 | 9.78 ± 2.11 | 69.57 ± 14.31 | 63.03 ± 12.47 | 0.59 ± 0.17 | 0.010 ± 0.00 | 52.19 ± 0.79 |
| DELETE | 83.34 ± 4.70 | 83.25 ± 4.73 | 82.19 ± 4.67 | 0.85 ± 0.06 | 0.002 ± 0.00 | 51.59 ± 0.80 | **0.00 ± 0.00** | **90.97 ± 0.07** | 81.00 ± 0.23 | **0.99 ± 0.01** | 0.002 ± 0.00 | **51.44 ± 1.05** |
| GA | 64.29 ± 0.47 | 63.78 ± 0.47 | 63.50 ± 0.27 | 0.47 ± 0.01 | 0.008 ± 0.00 | 51.42 ± 1.00 | 7.14 ± 0.74 | 72.70 ± 13.96 | 65.54 ± 12.27 | 0.65 ± 0.19 | 0.005 ± 0.00 | 52.52 ± 0.92 |
| GD | 89.35 ± 0.16 | 89.46 ± 0.16 | 88.34 ± 0.18 | 0.91 ± 0.01 | **0.001 ± 0.00** | 50.68 ± 0.14 | 84.69 ± 0.77 | 90.12 ± 0.75 | 88.46 ± 0.43 | 0.14 ± 0.01 | 0.021 ± 0.00 | 51.69 ± 0.62 |
| GDiff | 87.06 ± 2.17 | **87.51 ± 2.05** | 85.99 ± 2.10 | 0.92 ± 0.02 | **0.001 ± 0.00** | 52.34 ± 0.58 | 5.54 ± 4.01 | 89.45 ± 0.52 | 80.18 ± 0.72 | 0.92 ± 0.04 | 0.002 ± 0.00 | 52.06 ± 1.12 |
| NPO | 82.57 ± 1.63 | 82.22 ± 1.49 | 81.59 ± 1.76 | 0.89 ± 0.04 | 0.002 ± 0.00 | 50.68 ± 0.63 | **0.00 ± 0.00** | 75.38 ± 5.25 | 66.74 ± 4.73 | 0.72 ± 0.09 | 0.004 ± 0.00 | 51.60 ± 1.46 |
| SCRUB | 83.95 ± 1.06 | 84.64 ± 1.05 | 83.30 ± 1.07 | 0.94 ± 0.04 | **0.001 ± 0.00** | **50.53 ± 0.13** | **0.00 ± 0.00** | 92.66 ± 0.23 | 82.48 ± 0.23 | 0.97 ± 0.02 | **0.001 ± 0.00** | 51.57 ± 0.45 |
| PINV-Newton | 9.74 ± 3.35 | 9.84 ± 3.44 | 9.49 ± 3.34 | 0.01 ± 0.01 | 0.026 ± 0.00 | 49.81 ± 0.07 | 1.44 ± 2.03 | 8.86 ± 1.99 | 8.39 ± 1.71 | 0.05 ± 0.01 | 0.032 ± 0.00 | 50.42 ± 0.71 |
| Damped Newton | 8.48 ± 1.07 | 8.78 ± 0.77 | 8.89 ± 0.90 | 0.01 ± 0.00 | 0.029 ± 0.00 | 49.92 ± 0.46 | 0.52 ± 0.74 | 10.07 ± 1.14 | 9.28 ± 0.77 | 0.05 ± 0.01 | 0.024 ± 0.02 | 49.97 ± 0.05 |
| CʊReNU | 86.07 ± 0.20 | 86.39 ± 0.47 | 85.20 ± 0.08 | **0.98 ± 0.00** | 0.002 ± 0.00 | 50.74 ± 0.09 | 1.37 ± 0.64 | 88.65 ± 1.66 | 79.15 ± 1.59 | 0.93 ± 0.03 | 0.002 ± 0.00 | 52.21 ± 1.12 |
| CʊReNUS | **85.93 ± 0.45** | 86.27 ± 0.59 | **85.05 ± 0.42** | **0.98 ± 0.00** | **0.001 ± 0.00** | 50.69 ± 0.18 | 0.14 ± 0.11 | 90.88 ± 0.62 | **81.01 ± 0.49** | **0.99 ± 0.00** | **0.001 ± 0.00** | 51.98 ± 0.78 |

with the gradient evaluation. Subsequently, CʊReNUS requires $\mathcal{O}(T_{outer}C + T_{outer}T_{inner}C)$ time to compute a gradient per CʊReNUS iteration and compute an HVP per GD iteration. As mentioned before, $C$ also depends on the number of samples over which the gradient and the Hessian are defined, which are $n_1, n_2 \ll n_r$ in this case. Therefore, CʊReNUS is generally efficient and satisfies **D2**.

# 6 EXPERIMENTS

## 6.1 EXPERIMENTAL SETTINGS

**Datasets and Models.** Our experiments use four datasets: (1) **FashionMNIST** (FMNIST) (Xiao et al., 2017): contains 60,000 grayscale images of 10 fashion items, (2) **CIFAR-10** (Krizhevsky et al., 2009): contains 50,000 colour images of real-life objects, (3) **AG-News** (Zhang et al., 2015): contains 120,000 news titles and descriptions in 4 topics, (4) **TOFU** (Maini et al., 2024): contains 4000 question-answer pairs fictitiously generated by GPT-4. We train CNN for FMNIS, ResNet-18 (He et al., 2016) for CIFAR-10, and fine-tune Llama-2-7B (Touvron et al., 2023) with LoRA adapters (Hu et al., 2022) for AG-News and TOFU. Training hyperparameters of our models are detailed in App. F.1.

**Unlearning Baselines.** We compare against following unlearning algorithms: **Retraining** [10], **Random Labels** (Rand. Lbls.), **Gradient Ascent** (GA), **Gradient Descent** (GD), **GDiff** (Maini et al., 2024), Direct Preference Optimization with **"I don't know"** as positive response (IDK) (Maini et al., 2024) as baselines and **SCRUB** (Kurmanji et al., 2023), **DELETE** (Zhou et al., 2025), and **Negative Preference Optimization** (NPO) (Zhang et al., 2024b) as SOTA empirical unlearning algorithms. We also include Newton unlearning algorithm with two Hessian degeneracy baselines (Sec. 4): **Hessian pseudo-inverse** (PINV-Newton) and **damping Hessian** (Damped Newton) with small damping factor $\gamma = 10^{-3}$. Hyperparameters for the unlearning baselines are detailed in App. F.2.

**Unlearning Hyperparameters.** We use $L = 5$ for FMNIST, 50 for CIFAR-10, 80 for AG-News, and 400 for TOFU. Although the exact $L$ is often hard to find [11], we show that our algorithms are robust across different empirical choices of $L$ and describe a procedure to choose a valid $L$ in App. J.1. For CʊReNU, we only use 1 iteration on FMNIST. For CʊReNUS, we set $\sigma = 0.1$ and set $\eta$ to be the same learning rate as during training. We use $n_1 = 10, n_2 = 5$ for AG-News and TOFU, and $n_1 = 128, n_2 = 64$ for the rest. In terms of stochastic iterations, we use $T_{outer} = 20, T_{inner} = 5$ for FMNIST, and use $T_{outer} = 10, T_{inner} = 5$ for the rest. We study the effect of varying $\sigma$ and the number of stochastic iterations of CʊReNUS in App. J.2 and App. J.3, respectively.

**Evaluation Metrics.** Following previous works (Kurmanji et al., 2023; Maini et al., 2024; Zhao et al., 2024), we compare **accuracy/ROUGE** on $D_e$, $D_r$, and $D_{test}$ of the unlearned models and the retrained models (smaller gap is better) and report **Tug-of-War** (ToW) score that aggregates these gaps (higher is better). For classification tasks, we compute the **Jensen-Shannon divergence** (JS Div.) between the predicted probability distribution of the unlearned models and the retrained models (smaller is better) (Chundawat et al., 2023). For TOFU, we compute **Truth Ratio** of answering with incorrect answers versus correct answers when prompted with the question in $D_e$ (higher is better) (Maini et al., 2024). Additionally, we report AUC of the **Membership Inference Attack** (MIA) using ML-Doctor (Liu et al., 2022) for classification tasks and the Min-K++ attack (Zhang et al., 2025) for text generation tasks like TOFU (a smaller gap to retraining is better). We empirically

---

[10]For Llama-2 experiments, we use retraining to refer to fine-tuning the pretrained Llama-2 model on the retain set $D_r$ from scratch.

[11]It is empirically infeasible to enumerate over parametric space to obtain exact $L$.

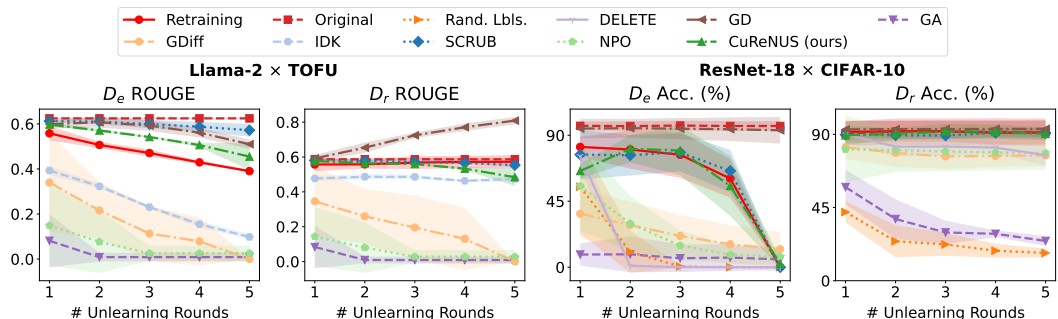

Figure 2: Sample-level sequential unlearning on Llama-2 × TOFU and class-level sequential unlearning on ResNet-18 × CIFAR-10 with 5 unlearning rounds (averaged over 5 random runs). Full results with $D_{test}$ are shown in App. H.1.

Table 3: Unlearning performance at the last round of sample-level sequential unlearning on Llama-2 × TOFU and class-level sequential unlearning on ResNet-18 × CIFAR-10 (over 5 random runs).

| Method | Llama-2 × TOFU | | | | | | ResNet-18 × CIFAR-10 | | | | | |
|---|---|---|---|---|---|---|---|---|---|---|---|---|
| | $D_e$ ROUGE ($\rightarrow$) | $D_r$ ROUGE ($\rightarrow$) | $D_{test}$ ROUGE ($\rightarrow$) | Truth Ratio ($\uparrow$) | ToW ($\uparrow$) | MIA ($\rightarrow$) | $D_e$ Acc. ($\rightarrow$) | $D_r$ Acc. ($\rightarrow$) | $D_{test}$ Acc. ($\rightarrow$) | ToW ($\uparrow$) | JS Div. ($\downarrow$) | MIA ($\rightarrow$) |
| Retraining | 0.390 ± 0.004 | 0.573 ± 0.034 | 0.731 ± 0.023 | 0.658 ± 0.007 | 1.00 ± 0.00 | 87.82 ± 5.78 | 0.000 ± 0.000 | 91.173 ± 7.363 | 77.508 ± 3.628 | 1.000 ± 0.000 | 0.000 ± 0.000 | 50.69 ± 0.64 |
| Original | 0.625 ± 0.003 | **0.587 ± 0.008** | **0.716 ± 0.035** | 0.508 ± 0.002 | 0.71 ± 0.03 | 100.00 ± 0.00 | 96.422 ± 2.883 | 90.237 ± 7.728 | 85.058 ± 3.526 | 0.033 ± 0.027 | 0.032 ± 0.004 | 49.68 ± 0.62 |
| Rand. Lbls. | - | - | - | - | - | - | 0.008 ± 0.018 | 17.065 ± 2.705 | 16.432 ± 2.806 | 0.106 ± 0.054 | 0.022 ± 0.01 | **50.93 ± 0.67** |
| DELETE | - | - | - | - | - | - | **0.000 ± 0.000** | 77.188 ± 6.152 | 66.640 ± 6.297 | 0.775 ± 0.180 | 0.015 ± 0.012 | 52.45 ± 1.09 |
| GD | 0.510 ± 0.018 | 0.809 ± 0.019 | 0.625 ± 0.054 | 0.538 ± 0.011 | 0.60 ± 0.08 | 99.81 ± 0.12 | 93.400 ± 9.144 | 93.363 ± 5.199 | 87.806 ± 1.077 | 0.057 ± 0.079 | 0.030 ± 0.007 | 51.37 ± 0.84 |
| GA | 0.009 ± 0.017 | 0.009 ± 0.017 | 0.000 ± 0.000 | 0.571 ± 0.102 | 0.07 ± 0.01 | 26.68 ± 17.08 | 5.384 ± 7.712 | 24.464 ± 2.996 | 22.180 ± 2.848 | 0.143 ± 0.049 | 0.027 ± 0.006 | 51.75 ± 0.66 |
| GDiff | 0.000 ± 0.000 | 0.000 ± 0.000 | 0.000 ± 0.000 | 0.808 ± 0.137 | 0.07 ± 0.01 | 40.25 ± 21.81 | 12.344 ± 11.709 | 77.071 ± 6.517 | 66.838 ± 6.070 | 0.669 ± 0.074 | 0.021 ± 0.009 | 51.47 ± 1.06 |
| IDK | 0.098 ± 0.012 | 0.474 ± 0.013 | 0.683 ± 0.040 | 0.566 ± 0.015 | 0.60 ± 0.02 | 99.89 ± 0.08 | - | - | - | - | - | - |
| NPO | 0.026 ± 0.030 | 0.028 ± 0.034 | 0.001 ± 0.002 | **0.831 ± 0.043** | 0.08 ± 0.02 | **78.52 ± 10.23** | 7.144 ± 5.106 | 78.498 ± 10.037 | 67.576 ± 8.286 | 0.732 ± 0.086 | 0.023 ± 0.008 | 51.94 ± 1.01 |
| SCRUB | 0.539 ± 0.033 | 0.542 ± 0.023 | 0.640 ± 0.066 | 0.512 ± 0.012 | 0.72 ± 0.03 | 100.00 ± 0.00 | **0.000 ± 0.000** | **90.704 ± 3.350** | 78.168 ± 1.708 | **0.944 ± 0.031** | 0.017 ± 0.015 | 50.16 ± 1.39 |
| CuReNUS | **0.455 ± 0.053** | 0.484 ± 0.045 | 0.706 ± 0.038 | 0.591 ± 0.043 | **0.80 ± 0.03** | 99.86 ± 0.13 | 2.320 ± 3.160 | 90.332 ± 4.003 | **77.590 ± 2.903** | 0.909 ± 0.050 | **0.011 ± 0.009** | 51.33 ± 1.26 |

observe that the MIA is largely ineffective on our models due to regularization effects (Kaya et al., 2020) and consider overfitted models in App. G. Lastly, unlearning efficiency is evaluated based on the average **unlearning time** (in seconds) and **peak memory usage** with respect to retraining.

## 6.2 Batch Unlearning

To benchmark computationally expensive unlearning algorithms (i.e., PINV-Newton, Damped Newton, and CuReNU) and affirm our analysis in Sec. 4, we perform *batch unlearning* (Sec. 3.1) on CNN × FMNIST. Following Kurmanji et al. (2023), $D_e$ is selected according to two scenarios: (1) *sample-level unlearning*, where a random subset of 80% samples in $D$ is removed[12], and 2) *class-level unlearning*, where all samples of a random class is removed.

Tab. 2 shows unlearning performance in the batch unlearning settings. Consistent with our conjecture in Sec. 4, PINV-Newton and Damped Newton exhibit poor unlearning performance due to excessively large-norm updates. In class-level batch unlearning, the Newton update norms are $3708.78 \pm 3364.67$ for PINV-Newton and $838.68 \pm 742.96$ for Damped Newton, both substantially larger than those of CuReNU ($0.36 \pm 0.07$) and CuReNUS ($0.38 \pm 0.05$). Our algorithms also maintain $D_e$ Acc. much closer to retraining than the first-order counterpart (GD), especially in class-level unlearning, which reiterates our benefits of stronger convergence guarantees. More importantly, both algorithms achieve high ToW, with CuReNUS consistently attaining the best ToW in both settings. This indicates that the outputs of the unlearned models closely approximate those of retraining. Overall, our unlearning performance, especially with CuReNUS, is comparable to SOTA empirical methods (SCRUB, DELETE) and even surpasses them on some metrics, demonstrating the potential of second-order unlearning algorithms in the realistic batch unlearning settings.

## 6.3 Sequential Unlearning

We perform *sequential unlearning* (Sec. 3.1) with 5 unlearning rounds on Llama-2 × TOFU and ResNet-18 × CIFAR-10. In each round, $D_e$ is chosen from 20% of the forget-10% split for TOFU and from 20% of a randomly selected class for CIFAR-10. We provide additional results on Llama-2 × AG-News, along with experiments on more unlearning rounds and different choices of $D_e$, in App. H. We note that many computationally expensive algorithms like PINV-Newton, Damped Newton, and CuReNU from our previous experiments are not applicable here due to large model sizes.

---

[12]We remove a large subset to induce noticeable outputs/performance changes for clearer comparison.

Table 4: Unlearning efficiency measured by running time (in seconds) and peak memory usage (in MB) of the best performing unlearning algorithms in Secs. 6.2 and 6.3 (averaged over 3 random runs).

| | CNN × FMNIST | | ResNet-18 × CIFAR-10 | | Llama-2 (+LoRA) × AG-News | | Llama-2 (+LoRA) × TOFU | |
|---|---|---|---|---|---|---|---|---|
| Trainable / Total Params | 20,728 / 20,728 | | 11,173,962 / 11,173,962 | | 1,064,960 / 6,608,424,960 | | 2,097,152 / 6,740,512,768 | |
| Metric | Unl. Time ($\downarrow$) | Peak Mem. ($\downarrow$) | Unl. Time ($\downarrow$) | Peak Mem. ($\downarrow$) | Unl. Time ($\downarrow$) | Peak Mem. ($\downarrow$) | Unl. Time ($\downarrow$) | Peak Mem. ($\downarrow$) |
| Retraining | 61.20 ± 8.70 | 1762 | 124.51 ± 10.95 | 3738 | 4792.44 ± 145.90 | 73896 | 900.71 ± 2.57 | 98340 |
| DELETE | 0.89 ± 0.10 | 1155 | 6.71 ± 0.05 | 2163 | - | - | - | - |
| SCRUB | 23.33 ± 0.43 | 1764 | 72.39 ± 4.93 | 3972 | 6796.16 ± 160.11 | 77112 | 178.52 ± 0.39 | 117000 |
| CuReNU | 6355.31 ± 127.31 | 6226 | - | - | - | - | - | - |
| CuReNUS | 35.54 ± 6.73 | 1588 | 41.79 ± 0.94 | 7404 | 85.26 ± 18.23 | 79140 | 340.24 ± 61.04 | 130826 |

Table 3 and Fig. 2 show unlearning performance in the sequential unlearning settings. Firstly, we note that many baselines struggle with either *under-forgetting* with trivial unlearning performance on $D_e$ (GD), or *over-forgetting* with degraded performance on $D_r$ and $D_{test}$ (Rand. Lbls., GA, and even DELETE). In contrast, CuReNUS is able to achieve close performance to retraining over multiple unlearning rounds. Its unlearning performance is comparable to SCRUB on CIFAR-10 and even surpasses it on TOFU across most metrics. These results suggest that CuReNUS is a practical and scalable unlearning algorithm that can effectively unlearn NNs/LLMs with minimal accumulated errors. The high MIA on TOFU, however, is likely due to the inherent distributional difference between the forget and test sets, making it easier to separate them.

**Remark 6.1.** *We note that our intention in Secs. 6.2 and 6.3 is not to claim new SOTA performance across all metrics, but rather to demonstrate that our unlearning algorithms can perform **competitively** with strong existing baselines (SCRUB, DELETE). The positive results show that our unlearning algorithms, especially CuReNUS, can unlock the potential of Newton unlearning for NNs/LLMs that previously struggled with degenerate Hessians. As no methods are strictly better across all metrics in these tables, CuReNUS is a viable unlearning algorithm worth considering in various scenarios.*

### 6.4 Unlearning Efficiency

Tab. 4 shows the unlearning time (averaged per batch) and the peak memory usage of the best performing methods (CuReNU, CuReNUS, DELETE, and SCRUB) in Secs. 6.2 and 6.3. We provide measures for the remaining unlearning algorithms in App. I. As expected, CuReNU incurs significant time and storage to store and invert the Hessians. CuReNUS is an efficient alternative of CuReNU, which only requires less than $2\times$ memory compared to retraining, consistent with our analysis in Sec. 5.2. Moreover, CuReNUS is significantly faster than SCRUB on large models like Llama-2. These results show that CuReNUS is a scalable second-order unlearning algorithm, setting it apart from the vanilla Newton unlearning. We note that SCRUB takes longer than retraining on AG-News because retraining is only performed for one epoch. On the other hand, Llama-2 experiments require more memory than ResNet-18 despite having fewer trainable parameters, as we must load both the pretrained model and the LoRA adapters into the memory.

## 7 Conclusion

While Newton unlearning has proven successful for linear models with positive, semi-definite Hessians, its application to trained NNs/LLMs is fundamentally hindered by degenerate Hessians. We bridged this gap with CuReNU and its stochastic variant CuReNUS, which are designed to tackle Hessian degeneracy in principle while the latter scales efficiently to large-scale NNs and LLMs. Our results show that our approaches, especially CuReNUS, deliver strong performance in both batch and sequential unlearning, generalizing well beyond the theoretically supported regime. We anticipate that future advances in HVP computation will further enhance the efficiency of CuReNUS.

ACKNOWLEDGMENTS

This research is supported by the National Research Foundation, Singapore under its AI Singapore Programme (AISG Award No: AISG3-RP-2022-029). This research is supported by the National Research Foundation Singapore and the Singapore Ministry of Digital Development and Innovation, National AI Group under the AI Visiting Professorship Programme (award number AIVP-2024-001).

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

## A    COMPARISON OF UNLEARNING ALGORITHMS AGAINST DESIDERATA

| Method | Effectiveness (D1) | Efficiency (D2) | Usability (D3) | Practicality (D4) |
|---|---|---|---|---|
| GD | ✗ | ✓ | ✗ | ✗ |
| Newton | ✗ | ✗ | ✗ | ✗ |
| PINV Newton | ✗ | ✗ | ✗ | ✗ |
| Damped Newton | ✗ | ✗ | ✗ | ✗ |
| Hessian-free Unlearning (Qiao et al., 2025) | ✓ | ✗ | ✓ | ✗ |
| CuReNU | ✓ | ✗ | ✓ | ✗ |
| CuReNUS | ✓ | ✓ | ✓ | ✓ |

Table 5: Comparison of different unlearning algorithms against our desiderata (Sec. 3.1).

Tab. 5 compares different unlearning algorithms with respect to our desiderata defined in Sec. 3.1. We note that **D2** is assessed based on both *time* and *memory* efficiency compared to retraining. Additionally, we also compare them regarding their applicability to challenging settings such as sequential unlearning.

**Gradient Descent (GD).**  GD is an efficient unlearning algorithm as it relies only on gradient information, satisfying **D2**. However, Sec. 6 shows that GD often fails to sufficiently remove the lineage of $D_e$ in both batch and sequential unlearning settings, hence violating **D1** and **D4**. Additionally, App. J.6 shows that its unlearning performance is susceptible to the selected learning rate, hence violating **D3**.

**Newton Unlearning.**  We note that vanilla Newton unlearning often fails to apply to neural networks due to problematic Hessians (Sec. 4), hence violating **D1** and **D4**. Without appropriate damping, it can produce large-norm updates and poor unlearning performance, indicating a failure to meet **D3**. Moreover, it requires storing and computing full Hessians, which is computationally expensive, and therefore does not satisfy **D2** and **D4**.

**PINV-Newton and Damped Newton.**  Both PINV-Newton and Damped Newton demonstrate poor unlearning performance in our experiments (Sec. 6), hence violating **D1** and **D4**. PINV-Newton does not involve explicit hyperparameters yet its unlearning performance is limited, thus violating **D3**. Damped Newton is sensitive to the choice of damping factor (Sec. 4), hence violating **D3**. Similar to the vanilla Newton method, both approaches also fail to meet **D2** and **D4** due to the high computational and storage costs of explicit Hessian computation and storage.

**CuReNU and CuReNUS.**  In Sec. 6, we show that CuReNU and CuReNUS achieve unlearning performance close to retraining, satisfying **D1**. Both algorithms remain robust across different choices of $L$ (App. J.1). CuReNU identifies the optimal damping factor via an optimization problem (Sec. 5.1), requiring minimal tuning (**D3**), but it does not meet **D2** and cannot be applied to sequential unlearning due to high computational and storage cost (violating **D4**). As its stochastic variant, CuReNUS retains robustness to hyperparameters such as the number of stochastic iterations (App. J.3) and $\sigma$ (App. J.2), which make it satisfy **D3**. Moreover, CuReNUS is computationally and memory efficient (**D2**), while being effective for sequential unlearning (Sec. 6.3), hence satisfying **D4**.

**Hessian-free Unlearning (Qiao et al., 2025).**  While the Hessian-free unlearning algorithm in Qiao et al. (2025) demonstrates effective unlearning for linear regressions and CNNs (**D1**) and robustness to varying step sizes and epochs (**D3**), it requires storing a substantial number of HVPs (Hessian-vector products) that scale with the dataset size, and therefore fails to satisfy **D2**. This high storage cost limits its overall application, including sequential unlearning (**D4**).

## B    JUSTIFICATION OF ASSUMPTIONS

For completeness, we restate our assumptions in Sec. 3.2.

**Assumption B.1.** *$\mathcal{L}$ is twice continuously differentiable with respect to $\mathbf{w}$.*

**Assumption B.2** ($\rho$-Lipschitz gradient)*. For some $\rho > 0$, $\|\mathbf{g_w} - \mathbf{g_{w'}}\| \leq \rho\|\mathbf{w} - \mathbf{w'}\|, \forall \mathbf{w}, \mathbf{w'} \in \mathbb{R}^d$.*

**Assumption B.3** ($L$-Lipschitz Hessian)*. For some $L > 0$, $\|\mathbf{H_w} - \mathbf{H_{w'}}\| \leq L\|\mathbf{w} - \mathbf{w'}\|, \forall \mathbf{w}, \mathbf{w'} \in \mathbb{R}^d$.*

Assumptions B.1, B.2 and B.3 are widely used in existing works (Zhang et al., 2024a; Sekhari et al., 2021; Guo et al., 2020). More importantly, they do not require the loss function to be convex,

allowing us to extend our analysis to non-convex settings such as neural networks. A caveat of these assumptions is that they may exclude non-differentiable activations, such as ReLU activations that are commonly used in neural networks. To address this, we empirically evaluate CuReNU and CuReNUS on ReLU-activated networks like CNN $\times$ FMNIST and ResNet-18 $\times$ CIFAR10 (Sec. 6). Our experiments show that both unlearning algorithms are able to achieve strong unlearning performance, suggesting that they generalize well beyond the theoretically limited regime.

## C   DERIVATIONS

Under Assumption B.1, $\mathbf{H}_{\mathbf{w}_t}^{D_r} \in \mathbb{R}^{d \times d}$ is symmetric and is orthogonally diagonalizable by the spectral theorem, i.e., $\mathbf{H}_{\mathbf{w}_t}^{D_r} = \mathbf{Q}\mathbf{\Lambda}\mathbf{Q}^{-1}$ with an orthornormal basis $\mathbf{Q} \triangleq [\mathbf{u}_1, \dots, \mathbf{u}_d]$ and a diagonal matrix $\mathbf{\Lambda} = \mathrm{diag}(\lambda_1, \dots, \lambda_d)$. Moreover, since $\mathbf{Q}$ is orthonormal, $\mathbf{Q}^T = \mathbf{Q}^{-1}$ and $\mathbf{u}_i^T \mathbf{u}_i = \|\mathbf{u}_i\|^2 = 1$.

### C.1   DERIVATION OF REMARK 4.2

**Remark C.1** (Restated of Remark 4.2). *The Newton update norm using the pseudo-inverse Hessian is* $\|\Delta_{t+1}\|^2 = \sum_{i:\lambda_i \neq 0} \frac{1}{\lambda_i^2}(\mathbf{u}_i^T \mathbf{g}_{\mathbf{w}_t}^{D_r})^2$.

We denote by $\mathbf{\Lambda}^\dagger \triangleq \mathrm{diag}(\frac{1}{\lambda_1}, \dots, \frac{1}{\lambda_d})$ a pseudo-inverse of $\mathbf{\Lambda}$, where $\frac{1}{\lambda_i} \triangleq 0$ if $\lambda_i = 0$. Expanding $(\mathbf{H}_{\mathbf{w}_t}^{D_r})^\dagger$, we get:

$$(\mathbf{H}_{\mathbf{w}_t}^{D_r})^\dagger \mathbf{g}_{\mathbf{w}_t}^{D_r} = (\mathbf{Q}\mathbf{\Lambda}\mathbf{Q}^{-1})^\dagger \mathbf{g}_{\mathbf{w}_t}^{D_r} = (\mathbf{Q}\mathbf{\Lambda}^\dagger \mathbf{Q}^{-1})\mathbf{g}_{\mathbf{w}_t}^{D_r} = (\mathbf{Q}\mathbf{\Lambda}^\dagger \mathbf{Q}^T)\mathbf{g}_{\mathbf{w}_t}^{D_r}$$

$$= \sum_{i=1}^{d} \frac{1}{\lambda_i}\mathbf{u}_i(\mathbf{u}_i^T \mathbf{g}_{\mathbf{w}_t}^{D_r}) = \sum_{i:\lambda_i \neq 0} \frac{1}{\lambda_i}\mathbf{u}_i(\mathbf{u}_i^T \mathbf{g}_{\mathbf{w}_t}^{D_r})$$

Since the above expression is a sum of orthonormal vectors $\mathbf{u}_i$, where each $\mathbf{u}_i$ is scaled by $\frac{1}{\lambda_i}(\mathbf{u}_i^T g_{\mathbf{w}_t}^{D_r})$, the update norm using the pseudo-inverse Hessian is given by:

$$\|(\mathbf{H}_{\mathbf{w}_t}^{D_r})^\dagger \mathbf{g}_{\mathbf{w}_t}^{D_r}\|^2 = \sum_{i:\lambda_i \neq 0} \frac{1}{\lambda_i^2}(\mathbf{u}_i^T \mathbf{g}_{\mathbf{w}_t}^{D_r})^2\|\mathbf{u}_i\|^2 = \sum_{i:\lambda_i \neq 0} \frac{1}{\lambda_i^2}(\mathbf{u}_i^T \mathbf{g}_{\mathbf{w}_t}^{D_r})^2.$$

### C.2   DERIVATION OF REMARK 4.3

**Remark C.2** (Restated of Remark 4.3). *The Newton update norm using the damped Hessian is* $\|\Delta_{t+1}\|^2 = \sum_{i=1}^{d} \frac{1}{(\gamma+\lambda_i)^2}(\mathbf{u}_i^T \mathbf{g}_{\mathbf{w}_t}^{D_r})^2$.

We denote by $\mathbf{\Lambda}_\gamma \triangleq \mathrm{diag}(\gamma + \lambda_1, \dots, \gamma + \lambda_d)$ a diagonal matrix of eigenvalues for the $\gamma$-damped Hessian $\mathbf{H}_{\mathbf{w}_t}^{D_r} + \gamma\mathbf{I}$, where $\gamma \geq \max\{0, -\lambda_d\}$. Expanding $\mathbf{H}_{\mathbf{w}_t}^{D_r} + \gamma\mathbf{I}$, we get:

$$(\mathbf{H}_{\mathbf{w}_t}^{D_r} + \gamma\mathbf{I})^{-1}\mathbf{g}_{\mathbf{w}_t}^{D_r} = (\mathbf{Q}\mathbf{\Lambda}_\gamma \mathbf{Q}^{-1})^{-1}\mathbf{g}_{\mathbf{w}_t}^{D_r} = (\mathbf{Q}\mathbf{\Lambda}_\gamma^{-1}\mathbf{Q}^{-1})\mathbf{g}_{\mathbf{w}_t}^{D_r} = (\mathbf{Q}\mathbf{\Lambda}_\gamma^{-1}\mathbf{Q}^T)\mathbf{g}_{\mathbf{w}_t}^{D_r}$$

$$= \sum_{i=1}^{d} \frac{1}{\gamma + \lambda_i}\mathbf{u}_i(\mathbf{u}_i^T \mathbf{g}_{\mathbf{w}_t}^{D_r}).$$

Therefore, as explained in App. C.1, the update norm using the $\gamma$-damped Hessian is:

$$\|(\mathbf{H}_{\mathbf{w}_t}^{D_r} + \gamma\mathbf{I})^{-1}\mathbf{g}_{\mathbf{w}_t}^{D_r}\|^2 = \sum_{i=1}^{d} \frac{1}{(\gamma + \lambda_i)^2}(\mathbf{u}_i^T \mathbf{g}_{\mathbf{w}_t}^{D_r})^2\|\mathbf{u}_i\|^2 = \sum_{i=1}^{d} \frac{1}{(\gamma + \lambda_i)^2}(\mathbf{u}_i^T \mathbf{g}_{\mathbf{w}_t}^{D_r})^2.$$

# D   DETAILS OF CUReNU

**Upper bound.** Here, we prove that the cubic regularized approximation $\tilde{\mathcal{L}}(\mathbf{w}_{t+1}; D_r)$ is a global upper bound of $\mathcal{L}(\mathbf{w}_{t+1}; D_r)$. A similar proof can be found in Lemma 1 of Nesterov & Polyak (2006).

*Proof.* For all $\mathbf{w}, \mathbf{w}' \in \mathbb{R}^d$,

$$\mathbf{g}_{\mathbf{w}'} - \mathbf{g}_{\mathbf{w}} = \int_0^1 \mathbf{H}_{\mathbf{w}+\tau(\mathbf{w}'-\mathbf{w})}(\mathbf{w}' - \mathbf{w})d\tau.$$

Then,

$$\|\mathbf{g}_{\mathbf{w}} - \mathbf{g}_{\mathbf{w}'} - \mathbf{H}_{\mathbf{w}}(\mathbf{w}' - \mathbf{w})\| \leq \left\|\int_0^1 \left(\mathbf{H}_{\mathbf{w}+\tau(\mathbf{w}'-\mathbf{w})} - \mathbf{H}_{\mathbf{w}}\right)(\mathbf{w}' - \mathbf{w})d\tau\right\|$$

$$\leq \int_0^1 \|(\mathbf{H}_{\mathbf{w}+\tau(\mathbf{w}'-\mathbf{w})} - \mathbf{H}_{\mathbf{w}})(\mathbf{w}' - \mathbf{w})\|d\tau$$

$$\leq \|\mathbf{w}' - \mathbf{w}\| \int_0^1 \|\left(\mathbf{H}_{\mathbf{w}+\tau(\mathbf{w}'-\mathbf{w})} - \mathbf{H}_{\mathbf{w}}\right)\|d\tau \quad \text{(sub-multiplicative)}$$

$$\leq \|\mathbf{w}' - \mathbf{w}\| \int_0^1 \tau L \|\mathbf{w}' - \mathbf{w}\|d\tau = \frac{L}{2}\|\mathbf{w}' - \mathbf{w}\|^2 \quad \text{(by Assumption B.3).}$$

Hence,

$$\left|\mathcal{L}(\mathbf{w}') - \mathcal{L}(\mathbf{w}) - \langle \mathbf{g}_{\mathbf{w}}, \mathbf{w}' - \mathbf{w}\rangle - \frac{1}{2}\langle \mathbf{H}_{\mathbf{w}}(\mathbf{w}' - \mathbf{w}), \mathbf{w}' - \mathbf{w}\rangle\right|$$

$$\leq \left|\int_0^1 \langle \mathbf{g}_{\mathbf{w}+\tau(\mathbf{w}'-\mathbf{w})} - \mathbf{g}_{\mathbf{w}} - \tau\mathbf{H}_{\mathbf{w}}(\mathbf{w}' - \mathbf{w}), \mathbf{w}' - \mathbf{w}\rangle d\tau\right|$$

$$\leq \int_0^1 \left|\langle \mathbf{g}_{\mathbf{w}+\tau(\mathbf{w}'-\mathbf{w})} - \mathbf{g}_{\mathbf{w}} - \tau\mathbf{H}_{\mathbf{w}}(\mathbf{w}' - \mathbf{w}), \mathbf{w}' - \mathbf{w}\rangle\right| d\tau$$

$$\leq \|\mathbf{w}' - \mathbf{w}\| \int_0^1 \|\mathbf{g}_{\mathbf{w}+\tau(\mathbf{w}'-\mathbf{w})} - \mathbf{g}_{\mathbf{w}} - \tau\mathbf{H}_{\mathbf{w}}(\mathbf{w}' - \mathbf{w})\|d\tau \quad \text{(by Cauchy-Schwarz inequality)}$$

$$\leq \|\mathbf{w}' - \mathbf{w}\| \int_0^1 \frac{L}{2}\tau^2\|\mathbf{w}' - \mathbf{w}\|d\tau = \frac{L}{6}\|\mathbf{w}' - \mathbf{w}\|^3.$$

Let the evaluated data be the retain set $D_r$, $\mathbf{w}' = \mathbf{w}_{t+1}$ and $\mathbf{w} = \mathbf{w}_t$. By solving the absolute value inequality above, for all $\mathbf{w}_t, \mathbf{w}_{t+1} \in \mathbb{R}^d$ we have:

$$\mathcal{L}(\mathbf{w}_t; D_r) + \langle \mathbf{g}_{\mathbf{w}_t}^{D_r}, \Delta_{t+1}\rangle + \frac{1}{2}\langle \mathbf{H}_{\mathbf{w}_t}^{D_r}\Delta_{t+1}, \Delta_{t+1}\rangle + \frac{L}{6}\|\Delta_{t+1}\|^3 \geq \mathcal{L}(\mathbf{w}_{t+1}; D_r), \qquad (8)$$

where $\Delta_{t+1} \triangleq \|\mathbf{w}_{t+1} - \mathbf{w}_t\|$. The left-hand side is the cubic regularized approximation $\tilde{\mathcal{L}}(\mathbf{w}_{t+1}; D_r)$.
□

## D.1   CONVERGENCE GUARANTEE

**Definition D.1** (Restated of Definition 5.2). *An $\varepsilon$-SOSP $\mathbf{w}$ of the function $\mathcal{L}$ (with L-Lipschitz Hessian) satisfies $\|\mathbf{g}_{\mathbf{w}}\| \leq \varepsilon$ and the minimum eigenvalue of the Hessian $\lambda_{min}(\mathbf{H}_{\mathbf{w}}) \geq -\sqrt{L\varepsilon}$.*

A 0-SOSP is a local minimum as $\|\nabla\mathcal{L}(\mathbf{w})\|$ is 0 and the Hessian is positive semi-definite. When $\varepsilon$ is small, the gradient norm is close to 0 and the Hessian's minimum eigenvalue is near non-negative, meaning that strongly negative curvature directions are absent or, if present, very mild. While finding an exact local minimum is computationally hard, an $\varepsilon$-SOSP with sufficiently small $\varepsilon$ can approximate a local minimum and avoid most saddle points and sharp local maxima.

**Proposition D.2** (Adapted from (Nesterov & Polyak, 2006, Theorem 1)). *For non-convex functions satisfying Assumption B.3, CuReNU converges to an $\varepsilon$-SOSP in $\mathcal{O}(\varepsilon^{-1.5})$ iterations.*

*Proof.* Let $\mathbf{w}_t$ denote the parameters in iteration $t$. We define $\mathbf{w}_{t+1}^* \triangleq \arg\min_{\mathbf{w}_{t+1}} \tilde{\mathcal{L}}(\mathbf{w}_{t+1}; D_r)$, where $\tilde{\mathcal{L}}(\mathbf{w}_{t+1}; D_r)$ is given in Eq. 3 and $\Delta_{t+1}^* \triangleq \mathbf{w}_{t+1}^* - \mathbf{w}_t$.

For $\mathbf{w}_{t+1}^*$ to be a global minimizer of $\tilde{\mathcal{L}}(\mathbf{w}_{t+1}; D_r)$, it must satisfy:

$$\mathbf{g}_{\mathbf{w}_t}^{D_r} + \mathbf{H}_{\mathbf{w}_t}^{D_r} \Delta_{t+1}^* + \frac{L}{2} \|\Delta_{t+1}^*\| \Delta_{t+1}^* = 0 \tag{9}$$

$$\mathbf{H}_{\mathbf{w}_t}^{D_r} + \frac{L}{2} \|\Delta_{t+1}^*\| \mathbf{I} \succeq 0 \tag{10}$$

Multiplying $\Delta_{t+1}^*$ once in the Eq. 9 and twice in the Inequality 10, we get:

$$\langle \mathbf{g}_{\mathbf{w}_t}^{D_r}, \Delta_{t+1}^* \rangle + \langle \mathbf{H}_{\mathbf{w}_t}^{D_r} \Delta_{t+1}^*, \Delta_{t+1}^* \rangle + \frac{L}{2} \|\Delta_{t+1}^*\|^3 = 0 \tag{11}$$

$$\langle \mathbf{H}_{\mathbf{w}_t}^{D_r} \Delta_{t+1}^*, \Delta_{t+1}^* \rangle + \frac{L}{2} \|\Delta_{t+1}^*\|^3 \geq 0 \tag{12}$$

Together, Eq. 11 and Ineq. 12 imply that:

$$\langle \mathbf{g}_{\mathbf{w}_t}^{D_r}, \Delta_{t+1}^* \rangle \leq 0. \tag{13}$$

**Step 1:** We now aim to bound the decrease in $\mathcal{L}(\cdot; D_r)$ at each iteration. Since Eq. 8 state that $\tilde{\mathcal{L}}(\mathbf{w}_{t+1}; D_r)$ is the upper bound of $\mathcal{L}(\mathbf{w}_{t+1}; D_r)$, we have:

$$\mathcal{L}(\mathbf{w}_t; D_r) - \mathcal{L}(\mathbf{w}_{t+1}^*; D_r) \geq \mathcal{L}(\mathbf{w}_t; D_r) - \tilde{\mathcal{L}}(\mathbf{w}_{t+1}^*; D_r).$$

Moreover, using Definition 3 of $\tilde{\mathcal{L}}(\cdot; D_r)$, Eq. 11 and Inequality 13, we have:

$$\mathcal{L}(\mathbf{w}_t; D_r) - \tilde{\mathcal{L}}(\mathbf{w}_{t+1}^*; D_r) = -\langle \mathbf{g}_{\mathbf{w}_t}^{D_r}, \Delta_{t+1}^* \rangle - \frac{1}{2}\langle \mathbf{H}_{\mathbf{w}_t}^{D_r} \Delta_{t+1}^*; \Delta_{t+1}^* \rangle - \frac{L}{6} \|\Delta_{t+1}^*\|^3$$

$$= -\frac{1}{2}\langle \mathbf{g}_{\mathbf{w}_t}^{D_r}, \Delta_{t+1}^* \rangle + \frac{L}{12} \|\Delta_{t+1}^*\|^3 \geq \frac{L}{12} \|\Delta_{t+1}^*\|^3.$$

Hence,

$$\mathcal{L}(\mathbf{w}_t; D_r) - \mathcal{L}(\mathbf{w}_{t+1}^*; D_r) \geq \frac{L}{12} \|\Delta_{t+1}^*\|^3.$$

**Step 2:** We set up the following auxiliary results.

**a)** From Eq. 10, we have that

$$\|\mathbf{g}_{\mathbf{w}_t}^{D_r} + \mathbf{H}_{\mathbf{w}_t}^{D_r} \Delta_{t+1}^*\| = \frac{L}{2} \|\Delta_{t+1}^*\|^2.$$

Moreover, by Inequality 2.2 in Lemma 1 of Nesterov & Polyak (2006), we get:

$$\|\mathbf{g}_{\mathbf{w}_{t+1}^*}^{D_r} - \mathbf{g}_{\mathbf{w}_t}^{D_r} - \mathbf{H}_{\mathbf{w}_t}^{D_r} \Delta_{t+1}^*\| \leq \frac{L}{2} \|\Delta_{t+1}^*\|^2.$$

By the triangle inequality, these results imply that:

$$\|\mathbf{g}_{\mathbf{w}_{t+1}^*}^{D_r}\| \leq \|\mathbf{g}_{\mathbf{w}_t}^{D_r} + \mathbf{H}_{\mathbf{w}_t}^{D_r} \Delta_{t+1}^*\| + \|\mathbf{g}_{\mathbf{w}_{t+1}^*}^{D_r} - \mathbf{g}_{\mathbf{w}_t}^{D_r} - \mathbf{H}_{\mathbf{w}_t}^{D_r} \Delta_{t+1}^*\| \leq L \|\Delta_{t+1}^*\|^2. \tag{14}$$

**b)** In view of Inequality 10, the following must hold:

$$-\lambda_{min}(\mathbf{H}_{\mathbf{w}_t}^{D_r}) \leq \frac{L}{2} \|\Delta_{t+1}^*\|. \tag{15}$$

**Step 3:** We define

$$\mu_L(\mathbf{w}) = \max\left\{ \sqrt{\frac{1}{L} \|\nabla\mathcal{L}(\mathbf{w}; D_r)\|}, -\frac{1}{L}\lambda_{min}(\nabla^2\mathcal{L}(\mathbf{w}; D_r)) \right\}. \tag{16}$$

Intuitively, $\mu_L(\mathbf{w}) \geq 0$ reflects the (non)-local optimality of $\mathbf{w}$, i.e. $\mathbf{w}$ is a local minimum with $\nabla\mathcal{L}(\mathbf{w}; D_r) = \mathbf{0}$ and $\nabla^2\mathcal{L}(\mathbf{w}; D_r) \succeq 0$ iff $\mu_L(\mathbf{w}) = 0$.

We now aim to show that $\|\Delta_{t+1}^*\| \geq \mu_L(\mathbf{w}_{t+1}^*)$. Indeed, expanding $\mu_L(\mathbf{w}_{t+1}^*)$ using Inequality 14 and Ineq. 15, we get:

$$\sqrt{\frac{1}{L}\|\mathbf{g}_{\mathbf{w}_{t+1}}^{D_r}\|} \leq \sqrt{\frac{1}{L} \cdot L\|\Delta_{t+1}^*\|^2} = \|\Delta_{t+1}^*\|$$

$$-\frac{1}{L}\lambda_{min}(\mathbf{H}_{\mathbf{w}_{t+1}^*}^{D_r}) \leq \frac{1}{L} \cdot \frac{L}{2}\|\Delta_{t+1}^*\| = \frac{1}{2}\|\Delta_{t+1}^*\|$$

Therefore, $\|\Delta_{t+1}^*\| \geq \mu_L(\mathbf{w}_{t+1}^*)$.

**Step 4:** Let $\mathcal{L}^*$ be a lower bound for $\mathcal{L}(\mathbf{w}; D_r)$. Given that CuReNU involves $T$ iterations, it follows from the results of **Step 1** and **Step 3** that:

$$\mathcal{L}(\mathbf{w}_0; D_r) - \mathcal{L}^* \geq \sum_{t=0}^{T-1}[\mathcal{L}(\mathbf{w}_t; D_r) - \mathcal{L}(\mathbf{w}_{t+1}^*; D_r)]$$

$$\geq \sum_{t=0}^{T-1}\frac{L}{12}\|\Delta_{t+1}^*\|^3$$

$$\geq \sum_{t=0}^{T-1}\frac{L}{12}\mu_L^3(\mathbf{w}_{t+1}^*) \geq \frac{TL}{12}\min_{1 \leq t \leq T}\mu_L^3(\mathbf{w}_t^*)$$

Hence,

$$\min_{1 \leq t \leq T}\mu_L(\mathbf{w}_t^*) \leq \left(\frac{12}{TL}(\mathcal{L}(\mathbf{w}_0; D_r) - \mathcal{L}^*)\right)^{1/3} \tag{17}$$

By our definition in Eq. 16, for $\min_{1 \leq t \leq T}\mu_L(\mathbf{w}_t)$ corresponding to an $\varepsilon$-SOSP (Definition 5.2), it must satisfy:

$$\min_{1 \leq t \leq T}\mu_L(\mathbf{w}_t^*) \geq \min_{1 \leq t \leq T}\sqrt{\frac{1}{L}\|\nabla\mathcal{L}(\mathbf{w}_t^*; D_r)\|} \geq \sqrt{\frac{\varepsilon}{L}}$$

$$\min_{1 \leq t \leq T}\mu_L(\mathbf{w}_t^*) \geq \min_{1 \leq t \leq T} -\frac{1}{L}\lambda_{min}\nabla\mathcal{L}(\mathbf{w}_t^*; D_r) \geq \sqrt{\frac{\varepsilon}{L}}$$

Using Inequality 17 and making $T$ the subject of the formula, we have that:

$$T \leq \frac{12\sqrt{L}(\mathcal{L}(\mathbf{w}_0; D_r) - \mathcal{L}^*)}{\varepsilon^{1.5}}.$$

Hence, $T \leq \mathcal{O}(\varepsilon^{-1.5})$. $\qquad\qquad\square$

### D.2 Duality Between the Primal and Dual Problems

Here, we restate the minimization problem of the cubic-regularized quadratic approximation of $\mathcal{L}(\mathbf{w}_{t+1}; D_r)$ as defined in Eq. 3:

$$\min_{\mathbf{w}_{t+1}}\left[\tilde{\mathcal{L}}(\mathbf{w}_{t+1}; D_r) = \mathcal{L}(\mathbf{w}_t; D_r) + \langle\mathbf{g}_{\mathbf{w}_t}^{D_r}, \Delta_{t+1}\rangle + \tfrac{1}{2}\langle\mathbf{H}_{\mathbf{w}_t}^{D_r}\Delta_{t+1}, \Delta_{t+1}\rangle + \tfrac{L}{6}\|\Delta_{t+1}\|^3\right].$$

Let $\alpha_{t+1} \triangleq \|\mathbf{w}_{t+1} - \mathbf{w}_t\|$. For the above primal problem, we have the dual optimization problem in $\alpha_{t+1}$ as described in Eq. 4:

$$\sup_{\alpha_{t+1}}\xi(\alpha_{t+1}), \quad \xi(\alpha_{t+1}) = -\tfrac{1}{2}\left\langle\left(\mathbf{H}_{\mathbf{w}_t}^{D_r} + \tfrac{L}{2}\alpha_{t+1}\mathbf{I}\right)^{-1}\mathbf{g}_{\mathbf{w}_t}^{D_r}, \mathbf{g}_{\mathbf{w}_t}^{D_r}\right\rangle - \tfrac{L}{12}\alpha_{t+1}^3$$

$$\text{s.t.} \quad \alpha_{t+1} \in \mathcal{Q} = \{\alpha \in \mathbb{R} : \mathbf{H}_{\mathbf{w}_t}^{D_r} + \tfrac{L}{2}\alpha\mathbf{I} \succ 0, \alpha \geq 0\}.$$

The duality gap is the difference between the optimized value of the primal problem $\min_{\mathbf{w}_{t+1}} \tilde{\mathcal{L}}(\mathbf{w}_{t+1}; D_r)$ and that of the dual problem $\sup_{\alpha_{t+1}} \xi(\alpha_{t+1})$. If the duality gap equals 0, which is called *strong duality*, the optimized value of the dual problem equals the optimized value of the primal problem. The following proposition, adapted from Nesterov & Polyak (2006, Theorem 10), states the duality gap between the primal and dual problems.

**Proposition B.2.1.** *For any $L > 0$, the primal and dual problems satisfy strong duality, i.e.,* $\min_{\mathbf{w}_{t+1}} \tilde{\mathcal{L}}(\mathbf{w}_{t+1}; D_r) = \sup_{\alpha_{t+1}} \xi(\alpha_{t+1})$. *Moreover, let* $\Delta_{t+1} \triangleq \left(\mathbf{H}_{\mathbf{w}_t}^{D_r} + \frac{L}{2}\alpha_{t+1}\mathbf{I}\right)^{-1} \mathbf{g}_{\mathbf{w}_t}^{D_r}$. *Then, for any* $\alpha_{t+1} \in \mathcal{Q}$, *the duality gap is* $\frac{4}{3L} \cdot \frac{\alpha_{t+1} + 2\|\Delta_{t+1}\|}{(\alpha_{t+1} + \|\Delta_{t+1}\|)^2} \cdot \xi'(\alpha_{t+1})^2 \geq 0$ *where $\xi'$ denotes the first-order derivative of $\xi$ w.r.t. $\alpha_{t+1}$.*

Note that the dual problem is a one-dimensional concave maximization problem over $\alpha_{t+1} \in \mathcal{Q}$. From Proposition B.2.1, $\tilde{\mathcal{L}}(\mathbf{w}_{t+1})$ is minimized when $\alpha_{t+1} \in \mathcal{Q}$ satisfies $\xi'(\alpha_{t+1}) = 0$, indicating a global maximizer of the dual problem. Moreover, even if finding such a maximizer is infeasible, finding $\alpha_{t+1} \in \mathcal{Q}$ with a small $\xi'(\alpha_{t+1})$ implies a small duality gap and thus the primal function value is near optimal. Additionally, the concave constrained dual problem can be solved efficiently instead of the non-convex primal problem through techniques to solve the trust-region subproblem, as we will show next.

## D.3 Solving the Dual Problem using Trust-Region Methods

Firstly, by taking the derivative of $\xi$ (defined in Eq. 4) w.r.t. $\alpha_{t+1}$ we have:

$$
\begin{aligned}
\xi'(\alpha_{t+1}) &= \frac{d}{d\,\alpha_{t+1}}\left[-\frac{1}{2}\langle(\mathbf{H}_{\mathbf{w}_t}^{D_r} + \frac{L}{2}\alpha_{t+1}\mathbf{I})^{-1}\mathbf{g}_{\mathbf{w}_t}^{D_r}, \mathbf{g}_{\mathbf{w}_t}^{D_r}\rangle - \frac{L}{12}\alpha_{t+1}^3\right] \\
&= \frac{L}{4}(\mathbf{g}_{\mathbf{w}_t}^{D_r})^T(\mathbf{H}_{\mathbf{w}_t}^{D_r} + \frac{L}{2}\alpha_{t+1}\mathbf{I})^{-1}(\mathbf{H}_{\mathbf{w}_t}^{D_r} + \frac{L}{2}\alpha_{t+1}\mathbf{I})^{-1}\mathbf{g}_{\mathbf{w}_t}^{D_r} - \frac{L}{4}\alpha_{t+1}^2 \\
&= \frac{L}{4}(\|\Delta_{t+1}\|^2 - \alpha_{t+1}^2).
\end{aligned}
$$

Setting $\xi'(\alpha_{t+1})$ to 0 and note that $\alpha_{t+1} \geq 0$ we have:

$$\|\Delta_{t+1}\| = \alpha_{t+1}. \tag{18}$$

Also, since $\Delta_{t+1} \succ 0$ we have:

$$\alpha_{t+1} > \max\left\{0, -\frac{2}{L}\lambda_d\right\}, \tag{19}$$

where $\lambda_d$ is the minimum eigenvalue of $\mathbf{H}_{\mathbf{w}_t}^{D_r}$. Here, we use trust-region methods (Conn et al., 2000) to solve Eq. 18 and Eq. 19. We consider the following trust-region subproblem:

$$\min_{\Delta_{t+1}} \langle\mathbf{g}_{\mathbf{w}_t}^{D_r}, \Delta_{t+1}\rangle + \frac{1}{2}\langle\mathbf{H}_{\mathbf{w}_t}^{D_r}\Delta_{t+1}, \Delta_{t+1}\rangle \quad \text{s.t.} \quad \|\Delta_{t+1}\| \leq \alpha_{t+1}.$$

By Corollary 7.2.2 of Conn et al. (2000), the above problem admits a global minimizer $\Delta_{t+1}^* \triangleq \left(\mathbf{H}_{\mathbf{w}_t}^{D_r} + \frac{L}{2}\alpha_{t+1}\mathbf{I}\right)^{-1}\mathbf{g}_{\mathbf{w}_t}^{D_r}$ that satisfies Eq. 18 and Eq. 19. Therefore, we adopt a standard procedure to solve the trust-region subproblem.

Our procedure starts from $\alpha_{t+1} = \max\{0, -\frac{2}{L}\lambda_d\} + \varepsilon$, where $\varepsilon$ is a small constant. Note that this choice of $\alpha_{t+1}$ yields the largest $\|\Delta_{t+1}\|$, as we showed the latter is a monotonically decreasing function of $\gamma = \frac{L}{2}\alpha_{t+1}$ (App. C.2). From this, two cases arise by comparing $\alpha_{t+1}$ and $\|\Delta_{t+1}\|$:

- **Case 1** happens when $\|\Delta_{t+1}\| \geq \alpha_{t+1}$. Empirically, we observe that this case is common for neural networks with highly non-convex losses. Then, we can use Newton's method to find the root of $\|\Delta_{t+1}\| - \alpha_{t+1} = 0$ using its derivative w.r.t. $\alpha_{t+1}$. However, we will empirically use Newton's method with a better-behaved function $\frac{1}{\|\Delta_{t+1}\|} - \frac{1}{\alpha_{t+1}}$ to avoid the tricky case when $\Delta_{t+1}$ has small eigenvalues (Conn et al., 2000, Section 7.3.3).

- **Case 2** happens when $\|\Delta_{t+1}\| < \alpha_{t+1}$. This means $\Delta_{t+1}^*$ is in the interior of the trust region (with radius $\alpha_{t+1}$), and the trust-region subproblem becomes an unconstrained optimization problem. However, by Corollary 7.2.2 of Conn et al. (2000), this implies that $\alpha_{t+1} = 0$ and the Hessian is itself positive definite, which contradicts our empirical observation (Obs. 4.1). Nonetheless, if this occurs, we will accept $\Delta_{t+1}^* = \Delta_{t+1}$.

### D.4 Pseudocode

We provide the pseudocode for CuReNU in Algorithm 1. The algorithm employs Newton's method with a trust region (App. D.3) to solve the dual problem. Following Conn et al. (2000), we use the Cholesky decomposition of the regularized Hessian matrix to enhance computational efficiency and numerical stability.

---

**Algorithm 1** CuReNU

---

**Input:** original model parameters $\mathbf{w}^*$, retained set $D_r$, objective function $\mathcal{L}$, Hessian Lipschitz constant $L$, number of unlearning iterations $T$, number of Newton's iterations $T_{inner}$, tolerance $\varepsilon$

1: Set $\mathbf{w}_0 = \mathbf{w}^*$
2: **for** $t = 0..T - 1$ **do**
3:      Get $\mathbf{g}_{\mathbf{w}_t}^{D_r} = \nabla\mathcal{L}(\mathbf{w}_t; D_r)$
4:      Get $\mathbf{H}_{\mathbf{w}_t}^{D_r} = \nabla^2\mathcal{L}(\mathbf{w}_t; D_r)$
5:      $\Delta_{t+1} = \text{SolveDualProblem}(\mathbf{H}_{\mathbf{w}_t}^{D_r}, \mathbf{g}_{\mathbf{w}_t}^{D_r}, L, \varepsilon, T_{inner})$
6:      Set $\mathbf{w}_{t+1} = \mathbf{w}_t + \Delta_{t+1}$
7: **end for**
    **Output:** unlearned model parameters: $\mathbf{w}_T$ ;

8: **function** SolveDualProblem($\mathbf{H}, \mathbf{g}, L, \varepsilon, T_{inner}$)                      ▷ See D.3
9:      Get the minimum eigenvalue $\lambda_d$ of $\mathbf{H}$
10:      Set $\gamma_0 = \max(0, -\lambda_d) + \varepsilon$
11:      Set $\alpha_0 = \frac{\gamma_0}{2L}$
12:      Factorize $\mathbf{H} + \gamma_0\mathbf{I} = \mathbf{LL}^T$                   ▷ Cholesky Decomposition
13:      Solve $\Delta$ in $(\mathbf{LL}^T)^{-1}\Delta = \mathbf{g}$
14:      **if** $\|\Delta\| \geq \alpha_0$ **then**
15:          **for** $t = 1..T_{inner}$ **do**                    ▷ Newton's Method
16:              **if** $|\|\Delta\| - \alpha_{t-1}| \leq \varepsilon$ **then**
17:                  **break**;
18:              **else**
19:                  Solve $\mathbf{Lu} = \Delta$
20:                  Set $\xi'(\alpha_{t-1}) = \frac{1}{\|\Delta\|} - \frac{1}{\alpha_{t-1}}$
21:                  Set $\xi''(\alpha_{t-1}) = \frac{\|\mathbf{u}\|^2}{\|\Delta\|^3} + \frac{1}{\gamma_{t-1}\alpha_{t-1}}$
22:                  Set $\gamma_t = \gamma_{t-1} - \frac{\xi'(\alpha_{t-1})}{\xi''(\alpha_{t-1})}$
23:                  Set $\alpha_t = \frac{\gamma_t}{2L}$
24:                  Factorize $\mathbf{H} + \gamma_t\mathbf{I} = \mathbf{LL}^T$          ▷ Cholesky Decomposition
25:                  Solve $(\mathbf{LL}^T)^{-1}\Delta = \mathbf{g}$
26:              **end if**
27:          **end for**
28:      **end if**
29:      **return** $\Delta$;
30: **end function**

---

## E Details of CuReNUS

### E.1 Convergence Guarantee

Throughout this section, we use $\tilde{\mathcal{O}}$ to hide the logarithmic factors, i.e. $\tilde{\mathcal{O}}(f(n)) = \mathcal{O}(f(n)\log^k n)$ for some constant $k$. To make the convergence analysis tractable, we need the following assumption.

**Assumption B.2.1.** *The stochastic gradient and Hessian estimates of $\mathcal{L}$ satisfy*

- $\forall \mathbf{w}, \mathbb{E}\left[\|\mathbf{g}_{\mathbf{w}}^{B_1} - \mathbf{g}_{\mathbf{w}}^{D}\|\right] \leq \sigma_1^2$ *and* $\|\mathbf{g}_{\mathbf{w}}^{B_1} - \mathbf{g}_{\mathbf{w}}^{D}\| \leq M_1$ *almost surely;*

- $\forall \mathbf{w}, \mathbb{E}\left[\|\mathbf{H}_{\mathbf{w}}^{B_2} - \mathbf{H}_{\mathbf{w}}^{D}\|\right] \leq \sigma_2^2$ *and* $\|\mathbf{H}_{\mathbf{w}}^{B_2} - \mathbf{H}_{\mathbf{w}}^{D}\| \leq M_2$ *almost surely.*

**Theorem B.2.1** (Adapted from (Tripuraneni et al., 2018, Corollary 1)). *For non-convex functions satisfying Assumptions B.3, B.2 and stochastic estimates satisfying Assumption B.2.1, with probability greater than $1 - \delta$, if $n_1 = \tilde{\mathcal{O}}\left(\frac{\sigma_1^2}{\varepsilon^2}\right)$ and $n_2 = \tilde{\mathcal{O}}\left(\frac{\sigma_2^2}{L\varepsilon}\right)$, CuReNUS can converge to an $\varepsilon$-SOSP in $\tilde{\mathcal{O}}(\varepsilon^{-3.5})$ stochastic gradient/HVP evaluations where $\varepsilon$ is sufficiently small.*

*Proof.* **Step 1:** Under Assumption B.2.1, we use the matrix Bernstein inequality Tropp et al. (2015) to derive the following concentration bounds for gradient and HVP:

- For $n_1 \geq \max(\frac{M_1}{c_1\varepsilon}, \frac{\sigma_1^2}{c_1^2\varepsilon^2})\frac{8}{3}\log\frac{2d}{\delta}$, then with probability $1 - \delta'$,

$$\|\mathbf{g}_{\mathbf{w}_t}^{B_1} - \mathbf{g}_{\mathbf{w}_t}^{D_r}\| \leq c_1\varepsilon$$

- For $n_2 \geq \max(\frac{M_2}{c_2\sqrt{L\varepsilon}}, \frac{\sigma_2^2}{c_2^2 L\varepsilon})\frac{8}{3}\log\frac{2d}{\delta}$, then with probability $1 - \delta'$,

$$\forall \mathbf{z} \in \mathbb{R}^d, \left\|\left(\mathbf{H}_{\mathbf{w}_t}^{B_2} - \mathbf{H}_{\mathbf{w}_t}^{D_r}\right)\mathbf{z}\right\| \leq c_2\sqrt{L\varepsilon}\|\mathbf{z}\|$$

For a sufficiently small $\varepsilon$, the above inequalities hold for $n_1 = \tilde{\mathcal{O}}\left(\frac{\sigma_1^2}{\varepsilon^2}\right)$ and $n_2 = \tilde{\mathcal{O}}\left(\frac{\sigma_2^2}{L\varepsilon}\right)$.

**Step 2:** We denote $\mathbf{w}_{t+1}^* \triangleq \min_{\mathbf{w}_{t+1}} \tilde{\mathcal{L}}^{sto}(\mathbf{w}_{t+1}; D_r)$ and $\Delta_{t+1}^* \triangleq \mathbf{w}_{t+1}^* - \mathbf{w}_t$, where $\tilde{\mathcal{L}}^{sto}(\mathbf{w}_{t+1}; D_r)$ is defined in Eq. 6. Here, we note that $\mathbf{w}_{t+1}^*$ is shown to be achievable by the gradient descent algorithm (perturbed by a small $\sigma$) as described in Sec. 5.2. The convergence guarantee of gradient descent is given in (Carmon & Duchi, 2019).

Similar to the proof in **Step 1** of App. D.1, we have:

$$\mathcal{L}(\mathbf{w}_t; D_r) - \tilde{\mathcal{L}}^{sto}(\mathbf{w}_{t+1}^*; D_r) \geq \frac{L}{12}\|\Delta_{t+1}^*\|^3.$$

However, unlike the results in App. D.1, it does not immediately follow that $\tilde{\mathcal{L}}^{sto}(\mathbf{w}_{t+1}; D_r)$ is the upper bound of $\mathcal{L}(\mathbf{w}_{t+1}; D_r)$. Instead, this upper bound holds only up to a certain tolerance, as we will show next.

**Step 3:** By the proof of Lemma 4 of Tripuraneni et al. (2018), if $\mathbf{w}_{t+1}^*$ is an $\varepsilon$-SOSP (Definition 5.2), then

$$\|\Delta_{t+1}^*\| \leq \frac{1}{2}\sqrt{\frac{\varepsilon}{L}}. \tag{20}$$

Otherwise, we have that $\|\Delta_{t+1}^*\| \geq \frac{1}{2}\sqrt{\frac{\varepsilon}{L}}$.

**Step 4:** Using the implication of $L$-Lipschitz Hessian (Eq. 8), Cauchy-Schwarz inequality with gradient and HVP concentration bounds in **Step 1**, and the results from **Step 2**:

$$\mathcal{L}(\mathbf{w}_{t+1}^*; D_r) - \mathcal{L}(\mathbf{w}_t; D_r)$$
$$\leq \langle \mathbf{g}_{\mathbf{w}_t}^{D_r}, \Delta_{t+1}^* \rangle + \langle \mathbf{H}_{\mathbf{w}_t}^{D_r}\Delta_{t+1}^*, \Delta_{t+1}^* \rangle + \frac{L}{6}\|\Delta_{t+1}^*\|^3$$
$$= \tilde{\mathcal{L}}^{sto}(\mathbf{w}_{t+1}^*; D_r) - \mathcal{L}(\mathbf{w}_t; D_r) + \langle \mathbf{g}_{\mathbf{w}_t}^{D_r} - \mathbf{g}_{\mathbf{w}_t}^{B_1}, \Delta_{t+1}^* \rangle + \frac{1}{2}\langle(\mathbf{H}_{\mathbf{w}_t}^{D_r} - \mathbf{H}_{\mathbf{w}_t}^{B_2})\Delta_{t+1}^*, \Delta_{t+1}^* \rangle$$
$$\leq -\frac{L}{12}\|\Delta_{t+1}^*\|^3 + c_1\varepsilon\|\Delta_{t+1}^*\| + \frac{c_2}{2}\sqrt{L\varepsilon}\|\Delta_{t+1}^*\|^2$$

We now consider the characteristics of $\mathbf{w}_{t+1}^*$:

*Case 1:* $\mathbf{w}_{t+1}^*$ is an $\varepsilon$-SOSP, then using Ineq. 20, we get:

$$\mathcal{L}(\mathbf{w}_{t+1}^*; D_r) - \mathcal{L}(\mathbf{w}_t; D_r) \leq -\frac{1}{96}\sqrt{\frac{\varepsilon^3}{L}} + \frac{c_1}{2}\sqrt{\frac{\varepsilon^3}{L}} + \frac{c_2}{8}\sqrt{\frac{\varepsilon^3}{L}} \leq -c\sqrt{\frac{\varepsilon^3}{L}},$$

where $c \geq \frac{1}{96} - \frac{c_1}{2} - \frac{c_2}{8}$ by making $c_1, c_2$ arbitrarily small (i.e. increasing $n_1$ and $n_2$).

*Case 2:* If $\mathbf{w}_{t+1}^*$ is not an $\varepsilon$-SOSP, then $\varepsilon \leq 4L\|\Delta_{t+1}^*\|^2$.

$$\mathcal{L}(\mathbf{w}_{t+1}^*; D_r) - \mathcal{L}(\mathbf{w}_t; D_r) \leq -\frac{L}{12}\|\Delta_{t+1}^*\|^3 + 4c_1 L\|\Delta_{t+1}^*\|^3 + c_2 L\|\Delta_{t+1}^*\|^3$$

$$\leq -\frac{1}{96}\sqrt{\frac{\varepsilon^3}{L}} + \frac{c_1}{2}\sqrt{\frac{\varepsilon^3}{L}} + \frac{c_2}{8}\sqrt{\frac{\varepsilon^3}{L}} \leq -c\sqrt{\frac{\varepsilon^3}{L}},$$

where $c \geq \frac{1}{96} - \frac{c_1}{2} - \frac{c_2}{8}$ by making $c_1, c_2$ arbitrarily small (i.e. increasing $n_1$ and $n_2$).

In both cases, we get the following bound:

$$\mathcal{L}(\mathbf{w}_t; D_r) - \mathcal{L}(\mathbf{w}_{t+1}^*; D_r) \geq c\sqrt{\frac{\varepsilon^3}{L}}. \tag{21}$$

**Step 5:** Let $\mathcal{L}^*$ be the lower bound of $\mathcal{L}(\mathbf{w}; D_r)$. CuReNUS involves at most $T$ iterations to decrease from $\mathcal{L}(\mathbf{w}_0; D_r)$ to $\mathcal{L}^*$, where the upper bound on per-iteration decrease is given in Inequality 21. Following the same argument as Step 4 of the proof of Prop D.2, the total number of iterations $T$ is:

$$T \leq \frac{\sqrt{L}(\mathcal{L}(\mathbf{w}_0, D_r) - \mathcal{L}^*)}{c\varepsilon^{1.5}}.$$

**Step 6:** Since $n_1 = \tilde{\mathcal{O}}(\frac{\sigma_1^2}{\varepsilon^2})$ and $n_2 = \tilde{\mathcal{O}}(\frac{\sigma_2^2}{L\varepsilon})$ from **Step 1**, each iteration (corresponding to minimizing the stochastic cubic-regularized approximation once) cost involve $n_1$ gradient and $n_2 \cdot \mathcal{T}(\varepsilon)$ HVP evaluations, where $\mathcal{T}(\varepsilon)$ is the number of steps for the gradient descent method to find a sufficiently good minimizer of $\tilde{\mathcal{L}}^{sto}$. By Lemma 1 of Tripuraneni et al. (2018), $\mathcal{T}(\varepsilon) \leq \tilde{\mathcal{O}}(\frac{\rho}{\sqrt{L\varepsilon}})$. Hence, the overall computational cost is

$$\tilde{\mathcal{O}}\left(\frac{\sqrt{L}(\mathcal{L}(\mathbf{w}_0, D_r) - \mathcal{L}^*)}{c\varepsilon^{1.5}}\left(\frac{\sigma_1^2}{\varepsilon^2} + \frac{\sigma_2^2}{L\varepsilon} \cdot \frac{\rho}{\sqrt{L\varepsilon}}\right)\right).$$

When $\varepsilon$ is sufficiently small, the above equals to $\tilde{\mathcal{O}}(\varepsilon^{-3.5})$ gradient/Hessian vector evaluations. $\square$

### E.2 PSEUDOCODE

We provide the pseudocode for CuReNUS in Algorithm 2. Following Carmon & Duchi (2019), when the stochastic gradient is large, i.e. $\|\mathbf{g}_{\mathbf{w}_t}^{B_1}\| \geq \frac{\rho^2}{L}$, we take a Cauchy step (steepest descent within the trust region), which is closed-form and computationally efficient as shown in Lines 12-13, to induce $\Omega(\sqrt{\frac{\varepsilon^3}{L}})$ decrease in the stochastic approximation $\tilde{\mathcal{L}}^{sto}$ and function value $\mathcal{L}$.

## F   DETAILED EXPERIMENTAL SETTINGS

### F.1   TRAINING HYPERPARAMETERS

We conduct our experiments on NVIDIA H100 GPUs (80GB) and NVIDIA H200 GPUs (141GB). Our evaluation is averaged across 3 random seeds $\{1, 2, 3\}$. The training hyperparameters in our experiments are detailed below.

**CNN $\times$ FMNIST.**   The dataset includes 60,000 training samples and 10,000 test samples. We adopt a small 2-layer CNN with 32 filters of size $3 \times 3$, max pooling of size $2 \times 2$, ReLU activations, and a fully connected layer with 64 hidden units. We train the CNN with the SGD optimizer with a batch size of 64 over 30 epochs, a learning rate of 0.01 with decay 0.5 every 2,000 steps.

---

**Algorithm 2** CURENUS

---

    **Input:** original model parameters $\mathbf{w}^*$, retained set $D_r$, objective function $\mathcal{L}$, gradient Lipschitz constant $\rho$, Hessian Lipschitz constant $L$, number of unlearning iterations $T$, number of gradient descent iterations $T_{inner}$, gradient perturbation parameter $\sigma$, learning rate $\eta$

1: Set $\mathbf{w}_0 = \mathbf{w}^*$
2: **for** $t = 0..T-1$ **do**
3:     Get $B_1, B_2$ independently sampled from $D_r$
4:     Get $\mathbf{g}_{\mathbf{w}_t}^{B_1} = \nabla \mathcal{L}(\mathbf{w}_t; B_1)$
5:     Get $\mathbf{H}_{\mathbf{w}_t}^{B_2} = \nabla^2 \mathcal{L}(\mathbf{w}_t; B_2)$
6:     $\Delta_{t+1} = \text{GRADIENTDESCENT}(\mathbf{H}_{\mathbf{w}_t}^{B_2}, \mathbf{g}_{\mathbf{w}_t}^{B_1}, \rho, L, T_{inner}, \sigma, \eta)$
7:     Set $\mathbf{w}_{t+1} = \mathbf{w}_t + \Delta_{t+1}$
8: **end for**
    **Output:** unlearned model parameters: $\mathbf{w}_T$;
9:
10: **function** GRADIENTDESCENT($\mathbf{H}, \mathbf{g}, \rho, L, T_{inner}, \sigma, \eta$)
11:     **if** $\|\mathbf{g}\| \geq \frac{\rho^2}{L}$ **then**                                      ▷ Cauchy Step
12:         Set $R_c = -\frac{\mathbf{g}^\top \mathbf{H} \mathbf{g}}{L\|\mathbf{g}\|^2} + \sqrt{\left(\frac{\mathbf{g}^\top \mathbf{H} \mathbf{g}}{L\|\mathbf{g}\|^2}\right)^2 + \frac{2\|\mathbf{g}\|}{L}}$
13:         Set $\Delta = -R_c \frac{\mathbf{g}}{\|\mathbf{g}\|}$
14:         **return** $\Delta$;
15:     **else**
16:         Set $\Delta_0 = \mathbf{0}$
17:         Get $\tilde{\mathbf{g}} = \mathbf{g} + \sigma\zeta$ where $\zeta \sim \text{Unif}(\mathbb{S}^{d-1})$            ▷ Gradient Descent
18:         **for** $t = 0..T_{inner}-1$ **do**
19:             $\Delta_{t+1} = \Delta_t - \eta\left(\mathbf{H}\Delta_t + \tilde{\mathbf{g}} + L\|\Delta_t\|\Delta_t\right)$
20:         **end for**
21:         **return** $\Delta_{T_{inner}}$;
22:     **end if**
23: **end function**

---

**ResNet-18 $\times$ CIFAR-10.** The dataset contains 50,000 training samples and 10,000 test samples. We train a ResNet18 (He et al., 2016) using the Adam optimizer with a batch size of 100 for 10 epochs, a learning rate of 0.001 with decay 0.5 every 5,000 steps, and a weight decay of $10^{-4}$.

**Llama-2 $\times$ AG-News.** The dataset contains 120,000 training samples and 7,600 test samples. We fine-tune the pretrained Llama-2-7B model from Hugging Face[13] using LoRA ($r = 2$, $\alpha = 2$, drop out 0.1) in bfloat16, with a batch size of 15 over 1 training epoch, learning rate of $10^{-4}$, and a warmup ratio of 0.03.

**Llama-2 $\times$ TOFU.** The training set contains 4,000 question–answer pairs fictitiously generated by GPT-4, while the test set contains 100 question–answer pairs about real-world authors. We finetune the pretrained Llama-2-7B-chat model from Hugging Face[14] using LoRA ($r = 4$, $\alpha = 16$, drop out 0.05) in bfloat16, with a batch size of 4 and 4 gradient accumulation steps over 5 training epochs with a learning rate of $10^{-3}$. We also provide results on full Llama-2-7B without LoRA in App. H.5.

We compute the empirical Hessian eigenspectrum density for Llama-2 $\times$ TOFU (Fig. 1, right) using the PyHessian package (Yao et al., 2020) with 100 iterations of the Stochastic Lanczos Quadrature algorithm. Due to memory constraints, it is infeasible to compute the Hessian eigenspectrum over the full dataset. Therefore, to mitigate variance, we repeat the process 10 times with independent mini-batches of size 8 and report the averaged results.

---

[13]https://huggingface.co/meta-llama/Llama-2-7b-hf.
[14]https://huggingface.co/meta-llama/Llama-2-7b-chat-hf.

F.2 Unlearning Baselines

Here, we describe the unlearning baselines and their hyperparameters used in our experiments. Unless otherwise specified, we conduct grid search for the best learning rate in $\{10^{-5}, 10^{-4}, 10^{-3}\}$.

- **Retraining**: trains the model from scratch on the retain set $D_r$. For Llama-2 experiments, we use retraining to refer to fine-tuning the pre-trained Llama-2 model on $D_r$ from scratch. The training/fine-tuning hyperparameters are the same as those used for training/fine-tuning the original model on the full training set $D$.

- **Random Labels** (Rand. Lbls.): fine-tunes the model on the randomly labeled $D_e$ for 1 epoch.

- **DELETE** (Zhou et al., 2025): uses the original model to generate new labels for $D_e$ (different from the true labels), then minimizes Kullback-Leibler (KL) divergence with respect to new labels for 1 epoch.

- **Gradient Descent** (GD): minimizes losses on $D_e$ via gradient descent on $D_r$. We run GD for 1 epoch on TOFU and 5 epochs for the rest, with the same learning rate as during training.

- **Gradient Ascent** (GA): maximizes losses on $D_e$ via gradient ascent for 1 epoch.

- **Gradient Difference** (GDiff) (Maini et al., 2024): minimizes the weighted average of the loss on $D_r$ and the negated loss on $D_e$. We sample a subset of $D_r$ to be of the same size as $D_e$ and assign equal weights to both loss terms. We run GDiff for 1 epoch on TOFU and 5 epochs for the rest.

- **SCRUB** (Kurmanji et al., 2023): maximizes the KL divergence to the output distribution of the original model on $D_e$ while minimizing it on $D_r$ in an alternative manner. We run SCRUB for 1 epoch on TOFU and 5 epochs for the rest.

- **PINV-Newton** (Sec. 4): replaces exact inverse with pseudo-inverse in the vanilla Newton unlearning. We run PINV-Newton for 1 epoch on FMNIST.

- **Damped Newton** (Sec. 4): adds a small diagonal matrix $\gamma\mathbf{I}$ to the degenerate Hessian in the vanilla Newton unlearning. We use $\gamma = 10^{-3}$ by default. We run Damped Newton for 1 epoch on FMNIST.

- **IDK** (Maini et al., 2024): encourages alternative answers such as "I don't know" when prompted the LLMs with questions in $D_e$. We run IDK for 1 epoch on TOFU.

- **NPO** (Zhang et al., 2024b): discourages the original answers/predictions in $D_e$. We run NPO for 1 epoch. We conduct grid search for the optimal $\beta$ in $\{0.5, 1, 2\}$.

While relevant, our experiments do not include a comparison with the Hessian-free work of Qiao et al. (2025) due to their prohibitively high computational cost and memory requirements. Particularly, Qiao et al. (2025) inherently incur substantial computational overhead due to the precomputation step that computes the Hessian-vector product for every sample in the training set. When evaluating their method on FMNIST, the precomputation step takes around $468.3$ hours ($\approx 19.5$ days) to run, which already far exceeds the time required for retraining ($\approx 61.20$ seconds) and defeats the purpose of unlearning. Additionally, their method requires $O(nd)$ memory to store HVP for every sample in the training set, where $n$ is the dataset size. The prohibitively long precomputation time and the significant memory requirement thus make Qiao et al. (2025) impractical in real-world settings.

F.3 Hessian-vector Products Computation

A common operation involving the Hessian matrix in many applications is its product with an arbitrary vector $\mathbf{v}$, known as a Hessian-vector product (HVP). Pearlmutter (1994) introduced an efficient method for computing HVPs without materializing the full Hessian $\mathbf{H}$ based on the following definition:

$$\mathbf{H}\mathbf{v} = \lim_{\varepsilon \to 0} \frac{1}{\varepsilon}[\nabla_{\mathbf{w}}\mathcal{L}(\mathbf{w} + \varepsilon\mathbf{v}) - \nabla_{\mathbf{w}}\mathcal{L}(\mathbf{w})] = \nabla_{\mathbf{w}}[\langle\nabla_{\mathbf{w}}\mathcal{L}(\cdot), \mathbf{v}\rangle](\mathbf{w}).$$

Here, HVP corresponds to the directional derivative of the gradient $\nabla_{\mathbf{w}}\mathcal{L}$ in the direction of vector $\mathbf{v}$.

Automatic differentiation (AD) can be used to compute the derivatives involved in HVP computation. At its core, AD builds a computational graph by decomposing the function into variables and

elementary operations, and applies the chain rule on this graph to compute derivatives. Depending on the order of multiplications in the chain rule, AD can be implemented in *forward mode* (propagating derivatives from inputs to outputs) or *backward mode* (propagating derivatives from outputs to inputs). Forward mode is often more memory-efficient than backward mode because it does not require storing all intermediate states in the computational graph during derivative computation.

By combining different AD modes, we can efficiently compute HVPs using two common approaches: *forward-over-backward* and *backward-over-backward*. The basic steps for each approach are outlined below.

**Forward-over-backward.**

1. Compute the gradient $\nabla_{\mathbf{w}}\mathcal{L}(\mathbf{w})$ using the backward-mode AD.
2. Compute the directional derivative of $\nabla_{\mathbf{w}}\mathcal{L}(\mathbf{w})$ in the direction of $\mathbf{v}$ using the Jacobian-vector product with forward-mode AD.

**Backward-over-backward.**

1. Compute the gradient $\nabla_{\mathbf{w}}\mathcal{L}(\mathbf{w})$ using the backward-mode AD.
2. Compute the scalar product $h(\mathbf{w}) \triangleq \langle \nabla_{\mathbf{w}}\mathcal{L}(\mathbf{w}), \mathbf{v} \rangle$.
3. Compute the gradient of $h(\mathbf{w})$ w.r.t. $\mathbf{w}$ using another backward-mode AD.

Our experiments use PyTorch's HVP implementation, which is supported through `torch.autograd.functional.hvp()`. Since forward-mode AD is not natively supported in PyTorch, HVPs are computed using the backward-over-backward approach, which has a larger memory footprint due to double backward differentiations. Moving forward, a more efficient implementation of HVP computation in PyTorch, such as the forward-over-backward approach, can be studied to further improve the efficiency of CuReNUS.

## G    Additional Results for MIA

As discussed in Sec. 6.1, we obtain the MIA results using ML-Doctor (Liu et al., 2022) by computing the AUC score based on the losses of the unlearned model on $D_e$ and $D_{test}$. However, we observe that the standard MIA based on losses of the unlearned models is not informative (e.g., Table 3), which we hypothesize is due to the following factors:

**Overfitting.** As mentioned in Kurmanji et al. (2023), the original model in their work has overfitted more than a state-of-the-art model on CIFAR-10 would, and their MIA is performed on the overfitted original and unlearned models. Additionally, the analysis in Liu et al. (2022) also shows that a higher overfitting level leads to better membership inference. However, the original model in our setting is more generalized (from Table 3, $D_{test}$ and $D_r$ accuracy are close to each other with a difference of 1.85%), and MIA on our non-overfitted models is less informative. While studies have shown that there are connections between overfitting and privacy leakage (Shokri et al., 2017; Yeom et al., 2018; Liu et al., 2022), in our setting, when the original model generalizes well on $D_{test}$, the losses on $D_e$ and $D_{test}$ would be less distinguishable, which leads to close-to-50% MIA result for both original and unlearned models. Therefore, we conduct a new experiment by training the original ResNet model and CNN model for 50 epochs to obtain overfitted models (e.g., from Table 6, $D_{test}$ accuracy is much lower than $D_r$ accuracy with a difference of 7.17%) and perform unlearning on the overfitted model. However, in practice, an overfitted model is less preferable than those with better generalization abilities. Tables 6 and 8 show the performance and MIA results for different unlearning algorithms on the overfitted CNN and ResNet models, where the MIA result is more informative. Based on the results from the overfitted ResNet, our CuReNUS successfully decreases MIA result by 6.06%. While there remains a gap between CuReNUS and the state-of-the-art method SCRUB, our CuReNUS offers advantages in theoretical support and efficiency.

**Test Loss Distributions.** In our MIA experiments on FMNIST, CIFAR, and AG-News, $D_e$ and $D_{test}$ samples are drawn from the same distribution, e.g., samples of the same class. This results in similar loss distributions on $D_e$ and $D_{test}$ samples on both the original and unlearned models. As shown in Fig. 3, both non-overfitted and overfitted models result in similar loss distributions on $D_e$ and $D_{test}$

Table 6: Performance and MIA results in the class-level sequential unlearning setting on the overfitted ResNet-18 × CIFAR-10 (averaged over 3 random runs). Results are reported at the last unlearning round.

| Method | Overfitted ResNet-18 × CIFAR-10 Class Removal | | | | | |
|---|---|---|---|---|---|---|
| | $D_e$ Acc. ($\rightarrow$) | $D_r$ Acc. ($\uparrow$) | $D_{test}$ Acc. ($\uparrow$) | ToW ($\uparrow$) | JS Div. ($\downarrow$) | MIA ($\rightarrow$) |
| Retraining | 0.00 ± 0.00 | 99.94 ± 0.01 | 84.52 ± 0.23 | 1.00 ± 0.00 | 0.0 ± 0.0 | 48.96 ± 1.21 |
| Original | 99.89 ± 0.03 | 99.90 ± 0.01 | 92.73 ± 0.07 | 0.00 ± 0.00 | 0.036 ± 0.0 | 57.61 ± 0.35 |
| Rand. Lbls. | 16.45 ± 1.92 | 94.96 ± 0.64 | 81.25 ± 1.07 | 0.77 ± 0.01 | 0.024 ± 0.0 | 60.14 ± 0.81 |
| DELETE | 0.05 ± 0.01 | 97.95 ± 0.18 | 82.44 ± 0.41 | **0.96 ± 0.01** | 0.011 ± 0.001 | 51.86 ± 0.74 |
| GD | 99.97 ± 0.03 | **99.86 ± 0.01** | **92.53 ± 0.17** | 0.00 ± 0.00 | 0.036 ± 0.0 | 57.81 ± 0.50 |
| GA | 4.35 ± 1.36 | 43.64 ± 14.91 | 37.10 ± 12.14 | 0.23 ± 0.10 | 0.029 ± 0.001 | **48.73 ± 0.86** |
| GDiff | 18.18 ± 1.33 | 78.15 ± 3.28 | 66.92 ± 3.31 | 0.53 ± 0.04 | 0.036 ± 0.0 | 49.93 ± 0.11 |
| NPO | 15.95 ± 1.14 | 90.71 ± 1.32 | 78.26 ± 1.35 | 0.72 ± 0.01 | 0.036 ± 0.0 | 51.47 ± 1.51 |
| SCRUB | **0.00 ± 0.00** | 85.92 ± 1.33 | 75.43 ± 0.83 | 0.78 ± 0.01 | **0.008 ± 0.002** | 49.68 ± 0.99 |
| CuReNUS | 1.82 ± 1.97 | 87.25 ± 0.77 | 74.52 ± 0.67 | 0.77 ± 0.02 | 0.017 ± 0.003 | 51.55 ± 1.87 |

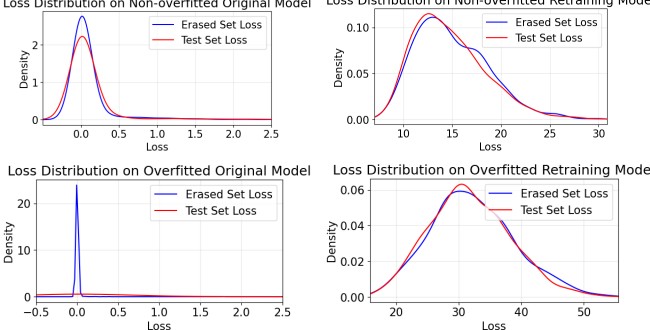

Figure 3: Loss distributions for samples on non-overfitted and overfitted, original and retraining models.

Table 7: Performance and MIA results for CuReNUS and other tested baselines in sequential unlearning setting on Llama-2 × TOFU where the test set is in a similar distribution as the forget set.

| Method | Llama-2 × TOFU | | | | | |
|---|---|---|---|---|---|---|
| | $D_e$ ROUGE ($\rightarrow$) | $D'_r$ ROUGE ($\rightarrow$) | $D'_{test}$ ROUGE ($\rightarrow$) | Truth Ratio ($\uparrow$) | ToW ($\uparrow$) | MIA ($\rightarrow$) |
| Retraining | 0.386 ± 0.004 | 0.576 ± 0.005 | 0.746 ± 0.027 | 0.664 ± 0.012 | 1.00 ± 0.00 | 63.72 ± 0.93 |
| Original | 0.628 ± 0.008 | 0.570 ± 0.007 | 0.679 ± 0.068 | 0.513 ± 0.011 | 0.69 ± 0.03 | 99.04 ± 0.26 |
| GD | 0.497 ± 0.010 | 0.759 ± 0.006 | 0.612 ± 0.033 | 0.545 ± 0.005 | 0.63 ± 0.04 | 94.89 ± 0.51 |
| GA | 0.000 ± 0.000 | 0.000 ± 0.000 | 0.000 ± 0.000 | 0.521 ± 0.024 | 0.07 ± 0.01 | **81.63 ± 16.03** |
| GDiff | 0.09 ± 0.09 | 0.09 ± 0.09 | 0.00 ± 0.00 | 0.81 ± 0.05 | 0.10 ± 0.02 | 91.98 ± 0.99 |
| IDK | 0.112 ± 0.024 | 0.462 ± 0.009 | 0.668 ± 0.025 | **0.570 ± 0.010** | 0.59 ± 0.01 | 96.17 ± 0.27 |
| SCRUB | 0.598 ± 0.020 | **0.549 ± 0.011** | 0.657 ± 0.083 | 0.508 ± 0.007 | **0.69 ± 0.05** | 98.83 ± 0.23 |
| NPO | 0.62 ± 0.16 | 0.43 ± 0.06 | 0.79 ± 0.13 | 0.59 ± 0.11 | 0.56 ± 0.01 | 71.79 ± 1.23 |
| CuReNUS | **0.477 ± 0.011** | 0.495 ± 0.006 | **0.678 ± 0.021** | **0.570 ± 0.001** | **0.76 ± 0.02** | 97.54 ± 2.13 |

Table 8: Performance and MIA results for CuReNUS and other tested baselines in batch unlearning setting on overfitted CNN × FMNIST (averaged over 3 random runs).

| Method | Overfitted CNN × FMNIST | | | | | |
|---|---|---|---|---|---|---|
| | $D_e$ Acc. ($\rightarrow$) | $D_r$ Acc. ($\rightarrow$) | $D_{test}$ Acc. ($\rightarrow$) | ToW ($\uparrow$) | JS Div. ($\downarrow$) | MIA ($\rightarrow$) |
| Retraining | 0.00 ± 0.00 | 86.07 ± 0.01 | 76.62 ± 0.11 | 1.00 ± 0.00 | 0.000 ± 0.000 | 54.04 ± 0.56 |
| Original | 92.57 ± 3.13 | 84.55 ± 0.80 | 84.32 ± 0.93 | 0.07 ± 0.03 | 0.024 ± 0.004 | 53.90 ± 0.81 |
| Rand. Lbls. | 4.70 ± 3.73 | 74.74 ± 6.43 | 67.02 ± 5.53 | 0.77 ± 0.10 | 0.007 ± 0.002 | 50.04 ± 0.48 |
| DELETE | 1.43 ± 1.29 | 82.67 ± 1.29 | 73.58 ± 1.18 | 0.92 ± 0.01 | **0.002 ± 0.001** | 50.84 ± 1.35 |
| GD | 92.21 ± 3.45 | **84.58 ± 0.78** | 84.30 ± 0.88 | 0.07 ± 0.03 | 0.023 ± 0.004 | **53.99 ± 0.37** |
| GA | 0.52 ± 0.49 | 76.82 ± 4.25 | 68.38 ± 3.77 | 0.83 ± 0.07 | 0.005 ± 0.003 | 51.34 ± 0.75 |
| GDiff | 0.18 ± 0.29 | 74.32 ± 9.96 | 66.20 ± 8.99 | 0.80 ± 0.16 | 0.006 ± 0.004 | 52.11 ± 1.44 |
| NPO | 5.81 ± 3.94 | 81.04 ± 1.21 | 72.77 ± 1.05 | 0.86 ± 0.04 | 0.003 ± 0.001 | 51.27 ± 0.67 |
| SCRUB | **0.00 ± 0.00** | 82.95 ± 0.29 | **73.90 ± 0.31** | **0.94 ± 0.01** | 0.006 ± 0.001 | 48.33 ± 3.49 |
| CuReNUS | 0.64 ± 1.04 | 81.65 ± 1.66 | 72.79 ± 1.41 | 0.91 ± 0.03 | 0.003 ± 0.001 | 51.20 ± 0.18 |

samples. This makes it more challenging for the MIA attacker to perform the binary classification, which thus results in not very high accuracy for both original and unlearned models. On the other hand, in our experiments on TOFU, the distribution of $D_{test}$ samples in the TOFU dataset is relatively different from $D_e$ samples, e.g., samples contain different question-answering content. This leads to different loss distributions on $D_e$ and $D_{test}$ samples that are naturally separable, which results in the large MIA AUC as shown in Table 3, where even Retraining reaches as high as 78.46. Therefore, we conduct a new experiment by selecting 400 samples from the original $D_r$ to form a new $D'_{test}$ to enforce a similar distribution as $D_e$. This also results in a new $D'_r$ with 3,200 samples. We train Llama-2 on the new training set, then perform unlearning and evaluate the results. Table 7 shows the performance and MIA results on this setting, where the MIA results are closer to 50 and Retraining achieves 63.72. Based on the results, while our CuReNUS does not achieve the optimal MIA result, it demonstrates competitive unlearning performance by decreasing the $D_e$ ROUGE and preserves the model utility well with high $D'_r$ and $D'_{test}$ ROUGE.

# H    SUPPLEMENTARY EXPERIMENTS FOR SEQUENTIAL UNLEARNING

## H.1    FULL RESULT FOR SEQUENTIAL UNLEARNING IN MAIN PAPER

Here, we show the full results that includes $D_{test}$ Acc. and ROUGE of sample-level sequential unlearning on Llama-2 × TOFU (Fig. 4, top) and class-level sequential unlearning on ResNet-18 × CIFAR-10 (Fig. 4, bottom). The results show that CuReNUS preserves decent performance on $D_{test}$ across the unlearning rounds in the sequential unlearning setting.

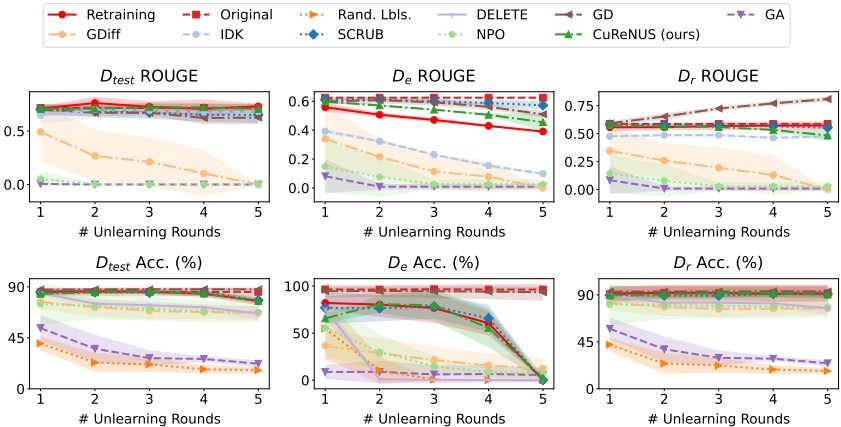

Figure 4: Full results on sample-level sequential unlearning on Llama-2 × TOFU (top) and class-level sequential unlearning on ResNet-18 × CIFAR-10 (bottom) with 5 unlearning rounds (averaged over 5 random runs).

## H.2    INCREASED UNLEARNING ROUND

In this experiment, we extended our sequential unlearning experiments to 10 rounds using ResNet-18 on the CIFAR-10 dataset. We increase $D_e$ to include two full classes (20% of the full dataset, or 10,000 samples). The experiment is conducted using the random seed 1. Despite the increased number of unlearning rounds and larger $D_e$, our results in Fig. 5 and Table 9 show that our CuReNUS can perform relatively well compared to the strong baselines by staying close to Retraining throughout the increased unlearning rounds and achieving the best ToW score at the end of sequential unlearning.

## H.3    CLASS-LEVEL SEQUENTIAL UNLEARNING ON LLAMA-2 × AG-NEWS

In this experiment, we perform class-level sequential unlearning with Llama-2-7B on the AG-News dataset (Zhang et al., 2015). The number of unlearning rounds is set to 3, which corresponds to 10,000 samples to be unlearned per round. The results in Table 10 and Fig. 6 show that our CuReNUS achieves good forgetting under the class-level setting, reaching a similar Acc. as SCRUB on $D_e$ as well as decent model utility by maintaining the $D_{test}$ Acc. and $D_r$ Acc..

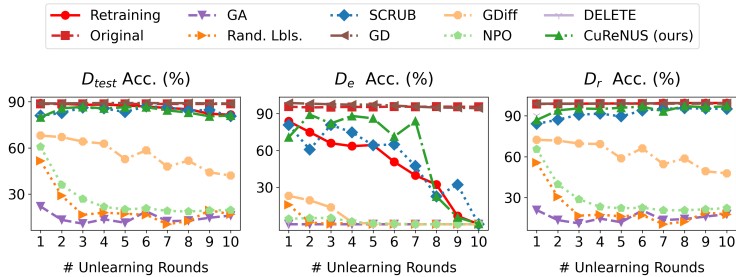

Figure 5: Extended sequential unlearning with 10 unlearning rounds on ResNet18 × CIFAR-10 (seed 1).

Table 9: Unlearning performance at the last unlearning round for extended sequential unlearning with 10 unlearning rounds on ResNet18 × CIFAR-10 (seed 1).

| Method | ResNet18 × CIFAR-10 | | | | | |
|---|---|---|---|---|---|---|
| | $D_e$ Acc. ($\rightarrow$) | $D_r$ Acc. ($\rightarrow$) | $D_{test}$ Acc. ($\rightarrow$) | ToW ($\uparrow$) | JS Div. ($\downarrow$) | MIA ($\rightarrow$) |
| Retraining | 0.00 | 99.40 | 81.56 | 1.00 | 0.000 | 53.36 |
| Original | 95.54 | 98.96 | 88.63 | 0.04 | 0.031 | 50.66 |
| Rand. Lbls. | **0.00** | 17.67 | 16.92 | 0.06 | 0.014 | 50.74 |
| DELETE | **0.00** | 97.63 | 80.23 | **0.97** | 0.007 | 50.26 |
| GD | 94.34 | **99.25** | 88.84 | 0.05 | 0.030 | 49.85 |
| GA | **0.00** | 17.82 | 16.09 | 0.06 | 0.032 | **53.18** |
| GDiff | **0.00** | 47.85 | 42.06 | 0.29 | 0.028 | 52.56 |
| SCRUB | **0.00** | 95.02 | 80.56 | 0.95 | 0.007 | 50.69 |
| NPO | **0.00** | 22.33 | 19.65 | 0.09 | 0.032 | 52.51 |
| CuReNUS | 0.38 | 96.98 | **80.98** | **0.97** | **0.006** | 53.72 |

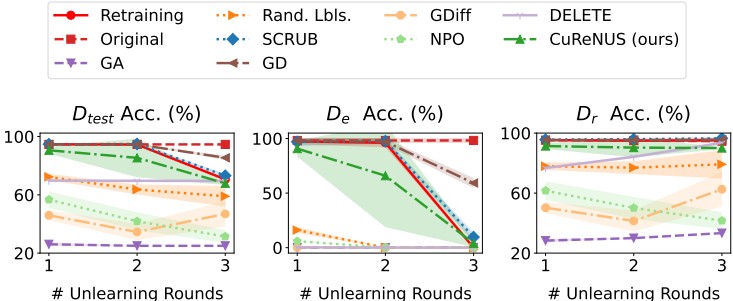

Figure 6: Class-level sequential unlearning on Llama-2 × AG-News (averaged over 3 random runs).

Table 10: Unlearning performance at the last unlearning round for 10% class-level sequential unlearning on Llama-2 × AG-News (averaged across 3 random runs).

| Method | Llama-2 × AG-News | | | | | |
|---|---|---|---|---|---|---|
| | $D_e$ Acc. ($\rightarrow$) | $D_r$ Acc. ($\rightarrow$) | $D_{test}$ Acc. ($\rightarrow$) | ToW ($\uparrow$) | JS Div. ($\downarrow$) | MIA ($\rightarrow$) |
| Retraining | 0.00 ± 0.00 | 95.20 ± 1.04 | 70.62 ± 0.85 | 1.00 ± 0.00 | 0.000 ± 0.000 | 49.96 ± 0.32 |
| Original | 98.24 ± 1.77 | 94.59 ± 0.55 | 94.60 ± 0.08 | 0.01 ± 0.01 | **0.014 ± 0.008** | 49.47 ± 1.95 |
| Rand. Lbls. | **0.00 ± 0.00** | 79.19 ± 7.75 | 59.01 ± 5.76 | 0.75 ± 0.13 | 0.019 ± 0.025 | 50.65 ± 0.89 |
| DELETE | **0.00 ± 0.00** | 93.48 ± 0.97 | **69.51 ± 0.79** | **0.97 ± 0.01** | 0.030 ± 0.008 | 49.52 ± 0.63 |
| GD | 59.21 ± 3.80 | **95.64 ± 0.66** | 85.42 ± 1.13 | 0.34 ± 0.03 | 0.045 ± 0.017 | 50.31 ± 1.31 |
| GA | **0.00 ± 0.00** | 33.33 ± 0.00 | 25.00 ± 0.00 | 0.21 ± 0.01 | 0.028 ± 0.027 | **49.90 ± 1.20** |
| GDiff | **0.00 ± 0.00** | 62.56 ± 9.37 | 46.90 ± 7.06 | 0.52 ± 0.12 | 0.015 ± 0.008 | 49.65 ± 0.41 |
| SCRUB | 9.69 ± 3.33 | 96.49 ± 0.68 | 73.14 ± 1.09 | 0.87 ± 0.04 | 0.018 ± 0.012 | 49.42 ± 1.52 |
| NPO | 0.01 ± 0.01 | 41.63 ± 4.56 | 31.37 ± 3.54 | 0.28 ± 0.05 | 0.028 ± 0.020 | 49.89 ± 0.81 |
| CuReNUS | 3.95 ± 2.85 | 90.03 ± 0.61 | 67.91 ± 0.99 | 0.89 ± 0.03 | 0.023 ± 0.007 | 51.23 ± 0.26 |

## H.4 SAMPLE-LEVEL SEQUENTIAL UNLEARNING

**Noisy Data Removal.** In this experiment, we mislabel 20% of the training set, such that the label is shifted right from the true label. We then perform sequential unlearning on mislabeled data for 10 rounds, each round removes 2% of the randomly selected mislabeled data. In this setting, unlearning can increase the model performance on $D_r$ and $D_{test}$ by removing the noisy data that harms the model utility. Tab. 11 shows our results, which demonstrate that our CURENUS performs well in removing noisy data and preserving the model utility.

Table 11: Unlearning performance at the last unlearning round for noisy data sequential unlearning on ResNet18 × CIFAR-10 (averaged over 3 random runs).

| Method | ResNet18 × CIFAR-10 | | | | | |
|---|---|---|---|---|---|---|
| | $D_e$ Acc. ($\rightarrow$) | $D_r$ Acc. ($\rightarrow$) | $D_{test}$ Acc. ($\rightarrow$) | ToW ($\uparrow$) | JS Div. ($\downarrow$) | MIA AUC ($\rightarrow$) |
| Retraining | 6.88 ± 0.10 | 100.00 ± 0.00 | 89.83 ± 0.02 | 1.00 ± 0.00 | 0.0 ± 0.0 | 53.55 ± 3.46 |
| Original | 26.41 ± 0.94 | 96.09 ± 0.05 | 81.97 ± 0.52 | 0.71 ± 0.01 | 2.9e-6 ± 0.8e-6 | **52.95** ± 1.62 |
| Rand. Lbls. | 11.70 ± 0.76 | 9.96 ± 1.66 | 9.93 ± 2.15 | 0.02 ± 0.00 | **2.8e-6** ± 1.1e-6 | 54.44 ± 0.45 |
| GD | 13.39 ± 0.58 | **97.90** ± 0.05 | **86.42** ± 0.11 | 0.88 ± 0.01 | 3.7e-6 ± 1.7e-6 | 51.80 ± 3.13 |
| GA | **6.98** ± 2.51 | 9.05 ± 0.86 | 9.04 ± 1.27 | 0.02 ± 0.00 | 19.8e-6 ± 22.7e-6 | 49.04 ± 3.19 |
| SCRUB | 2.52 ± 0.23 | 82.81 ± 1.77 | 80.43 ± 1.41 | 0.72 ± 0.02 | 3.5e-6 ± 1.3e-6 | 51.28 ± 3.54 |
| CURENUS | 9.06 ± 0.69 | 96.26 ± 0.36 | 85.92 ± 0.20 | **0.90** ± 0.00 | 3.7e-6 ± 1.7e-6 | 54.67 ± 3.43 |

**Benign Data Removal.** In this experiment, we perform sequential unlearning to iteratively remove a random training subset across multiple rounds, which we refer to as *sample-level sequential unlearning*. Fig. 7 and Tables 12, 13 show our results for sample-level sequential unlearning of 10% randomly selected training data on CIFAR-10 and AG-News. As observed, CURENUS maintains a close performance to Retraining and comparable results to SOTA methods (DELETE, SCRUB) on $D_e$ and does not degrade model performance on $D_{test}$ and $D_r$ even after multiple unlearning requests. This reinforces our argument that CURENUS is a good unlearning algorithm for long-term settings such as sequential unlearning on both class and sample levels.

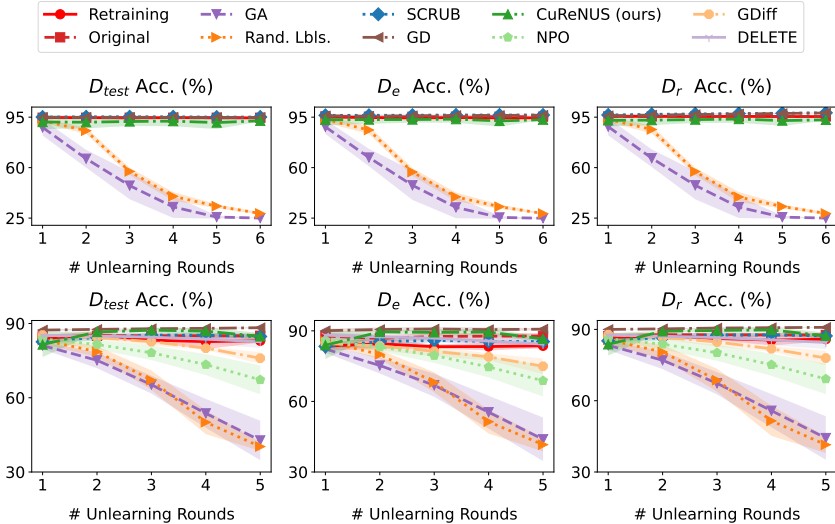

Figure 7: Sample-level sequential unlearning of 10% training data. **Top Row**: Llama-2 × AG-News (2000 unlearned samples per round). **Bottom Row**: ResNet-18 × CIFAR-10 (1000 unlearned samples per round).

## H.5 SEQUENTIAL UNLEARNING ON LLAMA-2 WITHOUT LORA

Our main results on Llama-2 focus on the LoRA-tuned setting because LoRA (Hu et al., 2022) is now a widely adopted approach for fine-tuning LLMs due to its efficiency and competitive effectiveness to full-scale fine-tuning (Schulman & Lab, 2025). Furthermore, fine-tuning datasets are often

Table 12: Unlearning performance at the last unlearning round for $10\%$ sample-level sequential unlearning on Llama-2 $\times$ AG-News (averaged over 3 random runs).

| Method | Llama-2 $\times$ AG-News | | | | | |
|---|---|---|---|---|---|---|
| | $D_e$ Acc. ($\rightarrow$) | $D_r$ Acc. ($\rightarrow$) | $D_{test}$ Acc. ($\rightarrow$) | ToW ($\uparrow$) | JS Div. ($\downarrow$) | MIA ($\rightarrow$) |
| Retraining | 94.55 ± 0.21 | 95.41 ± 0.11 | 94.53 ± 0.17 | 1.00 ± 0.00 | 0.000 ± 0.000 | 49.94 ± 0.42 |
| Original | **95.36 ± 0.10** | **95.50 ± 0.03** | **94.63 ± 0.06** | **0.99 ± 0.00** | 0.027 ± 0.013 | 50.06 ± 0.20 |
| Rand. Lbls. | 28.17 ± 0.46 | 28.21 ± 0.70 | 28.23 ± 0.72 | 0.04 ± 0.00 | **0.014 ± 0.011** | 49.71 ± 0.69 |
| GD | 96.39 ± 0.10 | 97.90 ± 0.10 | 94.84 ± 0.13 | 0.95 ± 0.00 | 0.021 ± 0.015 | 49.10 ± 0.62 |
| GA | 24.91 ± 0.08 | 25.02 ± 0.01 | 25.02 ± 0.02 | 0.03 ± 0.00 | 0.018 ± 0.002 | 50.24 ± 0.88 |
| SCRUB | 96.46 ± 0.17 | 98.00 ± 0.24 | 95.11 ± 0.09 | 0.95 ± 0.01 | 0.028 ± 0.010 | 49.86 ± 0.18 |
| CuReNUS | 91.46 ± 2.53 | 91.57 ± 2.26 | 90.35 ± 3.04 | 0.90 ± 0.07 | 0.018 ± 0.012 | **49.97 ± 0.35** |

Table 13: Unlearning performance at the last unlearning round for $10\%$ sample-level sequential unlearning on ResNet18 $\times$ CIFAR-10 (averaged over 3 random runs).

| Method | ResNet18 $\times$ CIFAR-10 | | | | | |
|---|---|---|---|---|---|---|
| | $D_e$ Acc. ($\rightarrow$) | $D_r$ Acc. ($\rightarrow$) | $D_{test}$ Acc. ($\rightarrow$) | ToW ($\uparrow$) | JS Div. ($\downarrow$) | MIA ($\rightarrow$) |
| Retraining | 83.64 ± 0.80 | 85.89 ± 0.68 | 83.01 ± 0.62 | 1.00 ± 0.00 | 0.0 ± 0.0 | 50.03 ± 0.11 |
| Original | 87.31 ± 1.05 | 87.11 ± 0.93 | 84.74 ± 0.66 | 0.94 ± 0.01 | 0.028 ± 0.004 | 51.64 ± 0.69 |
| Rand. Lbls. | 41.61 ± 2.34 | 41.46 ± 2.80 | 40.28 ± 1.42 | 0.19 ± 0.03 | 0.013 ± 0.0 | 50.41 ± 0.25 |
| DELETE | **84.35 ± 2.02** | **84.49 ± 2.28** | **82.61 ± 2.04** | **0.96 ± 0.03** | 0.029 ± 0.001 | 50.67 ± 0.63 |
| GD | 90.72 ± 0.31 | 90.86 ± 0.08 | 88.30 ± 0.19 | 0.84 ± 0.02 | **0.002 ± 0.0** | 51.55 ± 0.70 |
| GA | 43.90 ± 7.56 | 44.29 ± 7.48 | 42.75 ± 6.51 | 0.22 ± 0.07 | 0.032 ± 0.0 | **50.23 ± 0.32** |
| GDiff | 74.95 ± 2.69 | 77.95 ± 2.47 | 75.98 ± 1.93 | 0.78 ± 0.04 | 0.030 ± 0.001 | 51.23 ± 0.31 |
| NPO | 68.80 ± 5.49 | 69.14 ± 5.23 | 67.24 ± 4.78 | 0.60 ± 0.10 | 0.029 ± 0.002 | 51.23 ± 1.46 |
| SCRUB | 85.42 ± 0.84 | 87.26 ± 0.83 | 84.74 ± 0.82 | 0.95 ± 0.02 | 0.003 ± 0.0 | 50.50 ± 0.48 |
| CuReNUS | 86.54 ± 1.01 | 87.25 ± 1.18 | 84.77 ± 1.17 | 0.93 ± 0.03 | 0.003 ± 0.0 | 50.57 ± 0.24 |

task-specific datasets that are more privacy-sensitive than the public datasets used for full-scale LLM training, making them particularly relevant for unlearning. Nonetheless, we conducted an additional experiment on Llama-2-7B without LoRA on the TOFU dataset with a similar sequential unlearning setup as Sec. 6.3 to verify the scalability of our method to full-scale LLM fine-tuning.

We provide our results in Tab. 14, which show a similar trend to our results for LoRA-tuned LLM unlearning in Tab. 3. While CuReNUS does not achieve the best forgetting efficiency, as shown by a $D_e$ ROUGE gap of 0.162 from retraining, it shows a better unlearning trade-off between forgetting and utility preservation than SOTA methods like SCRUB via higher ToW score. More importantly, the fact that the Original model achieves the best ToW score amongst all tested methods shows that most of them are likely insufficient for large-scale unlearning, leading to either worse forgetting efficiency (observed in GD, CuReNUS) or worse utility preservation (observed in GA, GDiff, IDK, NPO, SCRUB). This challenging setting calls for the development of more robust unlearning methods in the future.

Table 14: Unlearning performance at the last unlearning round for sample-level sequential unlearning on Llama-2 $\times$ TOFU without LoRA (seed 1).

| Method | Llama-2 $\times$ TOFU | | | | | |
|---|---|---|---|---|---|---|
| | $D_e$ ROUGE ($\rightarrow$) | $D_r$ ROUGE ($\rightarrow$) | $D_{test}$ ROUGE ($\rightarrow$) | Truth Ratio ($\uparrow$) | ToW ($\uparrow$) | MIA ($\rightarrow$) |
| Retraining | 0.399 | 0.672 | 0.861 | 0.695 | 1.000 | 88.93 |
| Original | 0.692 | **0.686** | 0.894 | 0.539 | **0.674** | 100.00 |
| GD | 0.677 | 0.904 | **0.891** | 0.559 | 0.538 | 99.62 |
| GA | 0.000 | 0.000 | 0.000 | 0.344 | 0.027 | 10.44 |
| GDiff | 0.032 | 0.207 | 0.198 | 0.484 | 0.114 | 6.68 |
| IDK | 0.066 | 0.373 | 0.502 | 0.616 | 0.300 | 99.20 |
| NPO | 0.078 | 0.082 | 0.002 | **0.805** | 0.039 | 55.37 |
| SCRUB | **0.321** | 0.404 | 0.611 | 0.674 | 0.506 | **79.21** |
| CuReNUS | 0.561 | 0.535 | 0.695 | 0.547 | 0.603 | 99.99 |

# I  FULL RESULTS FOR UNLEARNING EFFICIENCY

Tab. 15 (an expanded version of Tab. 4 in the main paper) shows the running time comparison (in seconds) among different unlearning algorithms to unlearn a batch of erased data points across various datasets and models. As anticipated, the unlearning algorithms that utilize the second-order information, such as PINV-Newton, Damped Newton, and CuReNU, have the longest running times and even exceed that of Retraining on FMNIST. Therefore, these algorithms are impractical for large-scale experiments with ResNet18 and Llama-2. On the other hand, GA, NPO, and Rand. Lbls. are fast unlearning algorithms but tend to significantly degrade model performance post-unlearning, especially in long-term settings such as sequential unlearning (Sec. 6.3 and App. H.4). GD and GDiff are also fast unlearning algorithms, but they do not unlearn effectively. In contrast, DELETE is a strong state-of-the-art method that performs unlearning both efficiently and effectively. Meanwhile, CuReNUS can maintain decent efficiency across various datasets and models and be more efficient than the state-of-the-art method SCRUB despite being a second-order method by leveraging the fast HVPs and the stochastic setup, while maintaining a decent erasing quality and post-unlearning performance.

Table 15: Running time comparison (in seconds) across different datasets and models (averaged over 3 random runs).

| Dataset | FMNIST | | CIFAR-10 | | AG-News | | TOFU | |
|---|---|---|---|---|---|---|---|---|
| Model | 2-layer CNN | | ResNet18 | | Llama-2-7B (+LoRA) | | Llama-2-7B (+LoRA) | |
| Trainable Parameters | 20,728 | | 11,173,962 | | 1,064,960 | | 2,097,152 | |
| Retraining | 61.20 ± 8.70 | 1.0× | 124.51 ± 10.95 | 1.0× | 4792.44 ± 145.90 | 1.0× | 900.71 ± 2.57 | 1.0× |
| Rand. Lbls. | 1.70 ± 0.19 | 0.03× | 2.58 ± 0.10 | 0.02× | 144.63 ± 1.83 | 0.03× | - | - |
| DELETE | 0.89 ± 0.10 | 0.01× | 6.71 ± 0.05 | 0.05× | 133.54 ± 3.90 | 0.03× | - | - |
| GD | 9.04 ± 0.82 | 0.1× | 19.16 ± 4.03 | 0.2× | 4641.50 ± 407.93 | 0.96× | 181.45 ± 0.41 | 0.20× |
| GA | 2.28 ± 0.58 | 0.03× | 5.78 ± 0.26 | 0.04× | 105.46 ± 2.85 | 0.02× | 5.61 ± 0.63 | 0.01× |
| GDiff | 1.34 ± 0.03 | 0.02× | 6.51 ± 0.07 | 0.05× | 482.11 ± 103.48 | 0.10× | 50.38 ± 1.22 | 0.06× |
| PINV-Newton | 6185.72 ± 804.94 | 101.1× | - | - | - | - | - | - |
| Damped Newton | 6228.82 ± 739.82 | 101.7× | - | - | - | - | - | - |
| SCRUB | 23.33 ± 0.43 | 0.4× | 72.39 ± 4.93 | 0.6× | 6796.16 ± 160.11 | 1.4× | 178.52 ± 0.39 | 0.20× |
| IDK | - | - | - | - | - | - | 37.94 ± 0.38 | 0.04× |
| NPO | 0.80 ± 0.02 | 0.01× | 0.87 ± 0.05 | 0.01× | 134.34 ±2.27 | 0.03× | 31.02 ± 0.24 | 0.03× |
| CuReNU | 6355.31 ± 127.31 | 103.8× | - | - | - | - | - | - |
| CuReNUS | 35.54 ± 6.73 | 0.6× | 41.79 ± 0.94 | 0.3× | 85.26 ± 18.23 | 0.02× | 340.24 ± 61.04 | 0.38× |

# J  ABLATION STUDIES

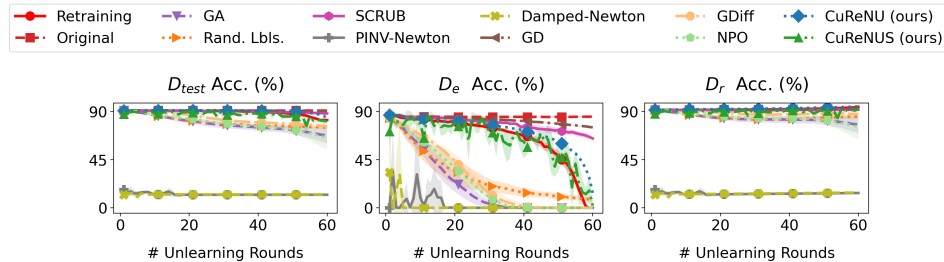

Figure 8: Class-level sequential unlearning performance on FMNIST (averaged over 3 random runs).

Throughout our ablation studies of CuReNU and CuReNUS, we adopt a class-level sequential unlearning setting with 60 unlearning rounds on the FMNIST dataset. We choose class 2 to be unlearned and report the averaged results over 3 random seeds $\{125, 126, 127\}$. Fig. 8 presents the sequential unlearning performance for this setup. The remaining ablation results are reported at the last unlearning round.

### J.1 EFFECT OF VARYING $L$

Since the exact Hessian Lipschitz constant $L$ is often hard to find, we treat it as a hyperparameter and analyze the effectiveness of our algorithms with varying choices of $L$ in Fig. 9. As can be seen, both algorithms exhibit consistent performance and remain close to the retraining baseline across different $L$ values. This highlights the robustness of CuReNU and CuReNUS with respect to the choice of $L$, in contrast to the strong dependence on learning rates in the many first-order unlearning algorithms such as GD, GA, and Rand. Lbls..

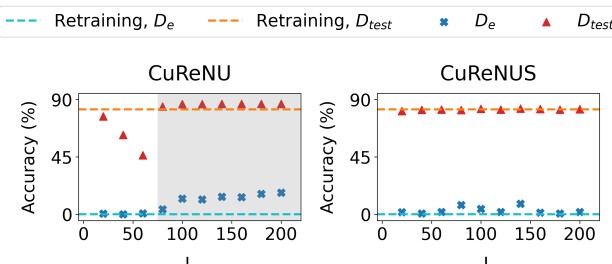

Figure 9: Performance of CuReNU and CuReNUS for different empirical choices of $L$ (seed 125).

On a separate note, it is important to keep $L$ large enough (e.g., approximately $L = 70$ for CuReNU, as marked by the shaded region) to ensure the validity of the cubic approximation and maintain convergence guarantees. Nonetheless, an appropriate value of $L$ can be determined during training and before unlearning actually happens.

**How to find a valid empirical $L$?** It is infeasible to empirically estimate the exact Hessian Lipschitz constant $L$ in neural networks, as it would require enumerating over the entire space of model parameters. To address this, we have adopted a more practical approach that is also suggested in Section 5.2 of Nesterov & Polyak (2006): starting with a random guess of $L$, we increase $L$ if the model fails to converge (since a valid upper bound would induce convergence); otherwise, we can use $L$, or decrease $L$ and check if a reduced $L$ still induces convergence. Since the actual $L$ is an upper bound, this procedure is guaranteed to return a valid empirical $L$.

**Caveats for non-Lipschitz models.** When the model fails to satisfy $L$-Lipschitz Hessian (i.e., no valid $L$ exists), selecting a sufficiently large $L$ remains beneficial as it can act as an effective regularizer to prevent large steps (large-norm updates) in suboptimal directions.

### J.2 EFFECT OF VARYING $\sigma$ IN STOCuReNU

Tab. 16 presents unlearning performance for varying levels of gradient perturbation $\sigma$ on CNN $\times$ FMNIST. We find that CuReNUS achieves comparable unlearning performance to retraining across a wide range of $\sigma$. This observation excludes $\sigma = 100$, where the stochastic gradient is heavily perturbed. Since gradient perturbation is intended to prevent the "hard case" (Conn et al., 2000), it is often sufficient to use a small $\sigma$, such as $\sigma < 1$, in our experiments while maintaining the fidelity of the stochastic gradient.

| $\sigma$ | $D_e$ Acc. | $D_r$ Acc. | $D_{test}$ Acc. |
|---|---|---|---|
| $10^{-3}$ | $0.072 \pm 0.097$ | $87.568 \pm 1.640$ | $76.525 \pm 1.161$ |
| $10^{-2}$ | $0.000 \pm 0.000$ | $87.922 \pm 3.056$ | $76.854 \pm 2.324$ |
| $10^{-1}$ | $1.072 \pm 1.857$ | $86.638 \pm 2.379$ | $75.820 \pm 1.582$ |
| $1$ | $0.000 \pm 1.279$ | $87.914 \pm 0.887$ | $76.712 \pm 0.977$ |
| $10$ | $0.072 \pm 0.125$ | $87.696 \pm 2.834$ | $76.729 \pm 2.533$ |
| $100$ | $0.016 \pm 0.016$ | $43.193 \pm 9.832$ | $37.441 \pm 8.486$ |
| Retraining | $0.000 \pm 0.000$ | $92.466 \pm 0.883$ | $80.225 \pm 0.963$ |

Table 16: Effect of $\sigma$ in CuReNUS (averaged over 3 random runs).

### J.3 EFFECT OF VARYING $T$ IN STOCURENU

Fig. 10 shows the unlearning performance when varying the number of stochastic iterations $T$ in CURENUS. As $T$ increases, CURENUS better approximates the retraining performance on $D_{test}$, $D_e$, and $D_r$. This illustrates the inherent trade-off between unlearning performance and computational efficiency in CURENUS. Nonetheless, we observe that CURENUS can obtain good unlearning performance with only around 10-20 iterations, offering a significant advantage over full retraining. In practice, the number of stochastic iterations in CURENUS can be tuned to meet specific unlearning requirements.

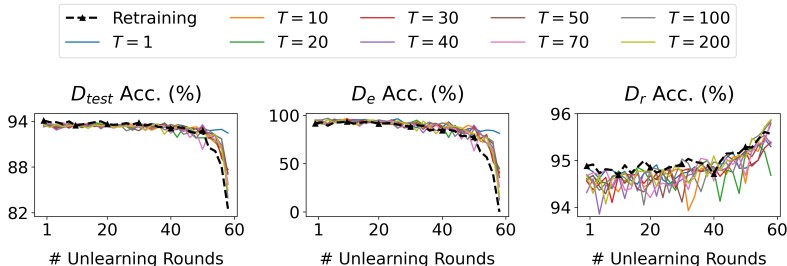

Figure 10: Effect of the number of stochastic iterations $T$ in CURENUS (seed 125).

### J.4 EFFECT OF VARYING BATCH SIZE IN STOCURENU

We assume the same batch sizes are used for gradient evaluation and Hessian-vector product evaluation in CURENUS, i.e., $n_1 = n_2$.[15] Tab. 17 presents unlearning performance on CNN $\times$ FMNIST for varying batch sizes. Our results show that CURENUS with mini-batch sampling can achieve comparable performance (within 2-4% difference on $D_e/D_r/D_{test}$ Acc.) to the full-batch setting (i.e, batch size $= |D_r|$), suggesting that the sampling-induced stochasticity might have minimal impact on unlearning effectiveness.

| Batch Size | $D_e$ Acc. | $D_r$ Acc. | $D_{test}$ Acc. |
|---|---|---|---|
| 32 | 1.839 ± 2.899 | 87.445 ± 1.585 | 76.400 ± 1.471 |
| 64 | 1.322 ± 1.002 | 88.863 ± 2.724 | 77.683 ± 2.711 |
| 128 | 1.344 ± 1.078 | 89.954 ± 1.604 | 78.958 ± 1.388 |
| 256 | 1.383 ± 2.073 | 90.383 ± 2.084 | 79.138 ± 1.426 |
| 512 | 4.544 ± 7.513 | 91.042 ± 1.747 | 79.950 ± 0.850 |
| 1024 | 0.133 ± 0.217 | 89.820 ± 3.593 | 78.321 ± 3.157 |
| 2048 | 0.150 ± 0.246 | 89.958 ± 3.461 | 78.433 ± 3.043 |
| $|D_r|$ | 0.206 ± 0.178 | 90.018 ± 3.360 | 78.542 ± 2.860 |
| Retraining | 0.000 ± 0.000 | 92.466 ± 0.883 | 80.225 ± 0.963 |

Table 17: Effect of batch size in CURENUS (averaged over 3 random runs).

### J.5 ANALYSIS OF DUAL VARIABLE $\alpha$ IN CURENU

We set the number of unlearning iterations $T = 1$ for CURENU and observe the value of $\alpha$ during class-level sequential unlearning on FMNIST in Fig. 11. Compared to the damping factor $\gamma = 10^{-3}$ in Damped Newton, the dual variable $\alpha$ in CURENU often admits larger values, which effectively prevents the model from excessively large norm updates.

During sequential unlearning, the value of $\alpha$ consistently decreases until around round 50. This may imply that the Hessian becomes more well-behaved over time, and less regularization is needed. After round 50, however, $\alpha$ increases again. This allows for larger norm updates, which may be needed for class unlearning.

---

[15]In practice, CURENUS can sample batches of different sizes for gradient evaluation and Hessian-vector product evaluation.

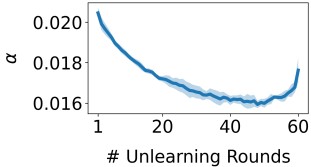

Figure 11: The dynamics of $\alpha$ in our CuReNU method (averaged over 3 random runs).

### J.6 Effect of varying learning rate in GD

Prior works have noted that the performance of gradient descent (GD) is highly sensitive to the choice of learning rate (Schaul et al., 2013). Here, we provide empirical evidence that supports this claim when GD is applied for unlearning in Fig. 12. As we can see, its performance on $D_e$ varies significantly across different learning rates. This suggests that identifying the optimal learning rate for GD to achieve unlearning performance close to retraining may be challenging in practice.

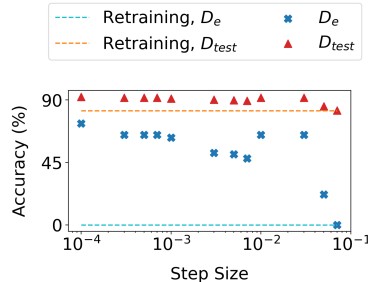

Figure 12: Effect of varying learning rates in GD (seed 125).

## K Formal Definition of Machine Unlearning

Here, we provide a formal definition of machine unlearning used in (Nguyen et al., 2022).

Let $D = \{(\mathbf{x}_i, y_i)\}_{i=1}^n \subseteq \mathcal{X} \times \mathcal{Y}$ denote the training set of $n$ samples, where $\mathbf{x}_i \in \mathbb{R}^d$ is the input and $y_i \in \mathbb{R}$ is the corresponding target. Let $D_e \subseteq D$ denote the *erased set* of $n_e$ samples to be unlearned and $D_r = D \setminus D_e$ denote the *retained set* of $n_r$ remaining samples.

Let $\Pr(A(D))$ denote the model distributions trained on $D$ using a randomized learning algorithm $A$. We denote by $U$ an unlearning algorithm that takes the training set $D$, the erased set $D_e$, the trained model $A(D) \in \mathcal{H}$, and returns an unlearned model in $\mathcal{H}$. $U$ is deemed *exact unlearning* iff

$$\forall \mathcal{T} \subseteq \mathcal{H}, D \subseteq \mathcal{X} \times \mathcal{Y}, D_e \subseteq D, \quad \text{then} \quad \Pr(A(D_r) \in \mathcal{T}) = \Pr(U(D, D_e, A(D)) \in \mathcal{T}).$$

Exact unlearning algorithms, such as Bourtoule et al. (2021); Yan et al. (2022), are often expensive to perform, especially for neural networks due to massive $D$ and large model sizes. Therefore, our goal is to achieve a relaxed notion of *approximate unlearning*, i.e.,

$$\forall \mathcal{T} \subseteq \mathcal{H}, D \subseteq \mathcal{X} \times \mathcal{Y}, D_e \subseteq D, \quad \text{then} \quad \Pr(A(D_r) \in \mathcal{T}) \approx \Pr(U(D, D_e, A(D)) \in \mathcal{T}).$$

We can view $A(D)$ as a function mapping any input $\mathbf{x}$ to real outputs (e.g., predicted logits for classification tasks) and $\mathcal{H}$ defines the set of all such models. Hence, our target $U$ must achieve *similar outputs* to retraining on any data. This motivates us to achieve the same retraining loss in our work.

## L Justification of Unlearning Goal

The common goal of unlearning algorithms in Sections 3.3, 5.1, and 5.2 is to minimize the loss $\mathcal{L}(\mathbf{w}; D_r)$ on the retained set to achieve unlearning of $D_e$. This choice is widely used in existing

works (Guo et al., 2020; Neel et al., 2021; Sekhari et al., 2021) and adopted in our work because it forms a necessary condition to approximate a retrained model on $D_r$, i.e., if an unlearned model does not minimize $\mathcal{L}(\mathbf{w}; D_r)$, it cannot be the same as a retrained model. In special cases like strongly-convex losses with at most one minimum, this necessary condition is also a sufficient condition. Nonetheless, our unlearning algorithms can be extended to other loss formulations, such as a weighted combination of $\mathcal{L}(\mathbf{w}; D_r)$ and $-\mathcal{L}(\mathbf{w}; D_e)$. Empirically, we find that incorporating a small negated gradient on $D_e$ helps prevent the model from being trapped at the original solution when minimizing the retraining loss.

## M  BACKGROUND IN OPTIMIZATION

As demonstrated in Sections 3.3, 5.1 and 5.2, we can view unlearning as an optimization process that starts from a local minimum of the loss on $D$ (the original model) and seeks a nearby local minimum of the loss on $D_r$ (a retrained model). To approach unlearning, we can study the broader optimization literature, which broadly categorizes optimization methods into *first-order* methods and *second-order* methods.[16] These optimization methods are further distinguished by their behavior in *convex* and *non-convex* settings, with unlearning neural networks typically situated in the more challenging non-convex setting.

**Convex vs. Non-Convex Optimization.**  Convex optimization involves minimizing a convex objective function (or maximizing a concave one) over a convex feasible set. In convex optimization problems, any local minimum is also a global minimum. Moreover, if the objective function is strictly convex, the local minimum, if exists, is unique. In contrast, non-convex optimization arises when the objective function is non-convex. In non-convex optimization, a *first-order* stationary point (where the gradient is close to 0) may correspond to a local minimum, a local maximum, or a saddle point. In contrast, a *second-order* stationary point (where the gradient is close to 0 and the Hessian is nearly positive semi-definite) can help avoid most saddle points and local maxima with strong negative curvature. Although finding a local minimum may not correspond to a global minimum, empirical studies show that many local minima in non-convex problems tend to have objective values nearly as good as the global minimum (Kashyap, 2022).

**First-Order Methods.**  Gradient descent (GD) and its variants (e.g., stochastic gradient descent (SGD), momentum (Sutskever et al., 2013), and Adam (Kinga et al., 2015)) are widely used first-order optimization methods in machine learning due to their scalability to high-dimensional data and complex models, such as deep neural networks. We subsequently restate the convergence results for GD and SGD. Interested readers can refer to (Nesterov, 2013; Garrigos & Gower, 2023; Khaled & Richtárik, 2020) for more details.

For a convex function with Lipschitz continuous gradient, GD achieves a convergence rate of $\mathcal{O}(\frac{1}{k})$ (sublinear convergence), where $k$ is the number of iterations. If the function is strongly convex, GD enjoys a linear convergence rate of $\mathcal{O}(c^k)$ for some constant $0 < c < 1$. For non-convex optimization, GD can only guarantee convergence to an $\varepsilon$-first-order stationary point ($\varepsilon$-FOSP) in $\mathcal{O}(\varepsilon^{-2})$ iterations.

In the stochastic setting, SGD converges in expectation at a rate of $\mathcal{O}(\frac{1}{\sqrt{k}})$ for convex functions and $\mathcal{O}(\frac{1}{k})$ for strongly convex functions. If the function is non-convex, SGD can only guarantee convergence to an $\varepsilon$-FOSP in $\mathcal{O}(\varepsilon^{-4})$ iterations.

**Second-Order Methods.**  Many real-world functions, such as objective functions in neural networks, are inherently non-convex. Although first-order optimization methods like GD and SGD are computationally efficient and scalable, they often suffer from slow convergence in ill-conditioned problems and may get stuck at saddle points if the function is non-convex. In contrast, Newton's method leverages the second-order information of the function to accelerate the convergence. When the function is strongly convex with a Lipschitz continuous Hessian, and the initialization is sufficiently close to the minimizer, Newton's method converges locally at a quadratic rate (Nocedal & Wright, 2006). In general, however, its global convergence is not guaranteed. Moreover, Newton's method may instead converge to saddle points or maxima, as it does not distinguish among stationary points.

---

[16]While higher-order methods exist, they are rarely adopted due to significant computational and numerical instability.

To achieve global convergence, cubic regularization methods (Nesterov & Polyak, 2006) construct a global upper bound of the objective function by adding a cubic term to the quadratic approximation. For a function (possibly non-convex) with Lipschitz continuous Hessian, cubic-regularized Newton's method can achieve convergence to an $\varepsilon$-second-order stationary point ($\varepsilon$-SOSP) in $\mathcal{O}(\varepsilon^{-1.5})$ iterations. For small $\varepsilon$, converging to $\varepsilon$-SOSP can help avoid most saddle points and sharp local maxima.

In the stochastic setting, Kohler & Lucchi (2017) proposes a variant of cubic regularization that utilizes subsampled gradients and Hessians, but does not provide its asymptotic analysis. Xu et al. (2020) considers stochastic Hessians but still requires access to the full gradients. Leveraging HVPs to avoid explicit Hessian computation, Tripuraneni et al. (2018) proposes a scalable Hessian-free variant that converges to an $\varepsilon$-SOSP in $\tilde{\mathcal{O}}(\varepsilon^{-3.5})$ gradient/HVP evaluations.

