# OpenReview forum: "How to Cure Newton for Unlearning Neural Networks? An Empirical Study from the Hessian Perspective"
_ICLR.cc/2026/Conference — ICLR 2026 Poster_

### Official Review · Reviewer_wc7S · 2025-10-29

**Soundness:** 2
**Presentation:** 3
**Contribution:** 2
**Rating:** 6
**Confidence:** 3

**Summary:**

The paper identifies a key limitation in second-order unlearning methods like damped Newton, where Hessian degeneracy near local optima causes unstable and excessively large parameter updates that harm performance. To address this, the authors propose Cubic-Regularized Newton Unlearning (CuReNU) and its scalable variant StoCuReNU, which apply cubic regularization to automatically control the Hessian damping factor. This approach mitigates degeneracy, stabilizes update norms, and provides theoretical convergence guarantees to second-order stationary points.

**Strengths:**

Unlearning is an important topic and the authors clearly explain and evaluate their proposal.

**Weaknesses:**

The evaluation is limited to small models or LoRA ona small LLM. Thus it is unclear how this will scale to large models where retraining is not realistic, and approximate unlearning is needed. Some parts of the evaluation technique are unclear.

**Questions:**

I would have appreciated some analysis of the results in Figure 2. In particular, the experiment for unlearning on Llama-2 still has 40\% accuracy $D_e$ regardless of whether StoCuReNU or retraining is used, which suggests that the samples being targeted for unlearning are not unique within the training set. As a result, it is not clear if the experiments can be conclusive in this case.

I was also confused on how the authors obtained a retraining time measurement for Llama2-7B. If I understand this should be the time to retrain Llama2-7B from scratch minus the data in the dataset they want forgotten. However, it's more likely this is the time to fine-tune Llama2-7B on the dataset. Thus, it seems misleading to call this the retraining time for Llama2-7B since that is not how a fully exact retraining method would work.

---

> ### Author Response · Authors · 2025-11-26
>
> > The evaluation is limited to small models or LoRA ona small LLM. Thus it is unclear how this will scale to large models where retraining is not realistic, and approximate unlearning is needed. Some parts of the evaluation technique are unclear.
>
> Our evaluation focuses on the LoRA-tuned setting because LoRA is now a **widely adopted** approach for fine-tuning LLMs due to its efficiency and competitive effectiveness to full-scale fine-tuning (Schulman \& Thinking Machines Lab, 2025). Furthermore, fine-tuning datasets are often task-specific datasets that are more privacy-sensitive than the public datasets used for full-scale LLM training, making them particularly relevant for unlearning.
>
> Nonetheless, we conducted an additional experiment to evaluate our StoCuReNU and other baseline methods on full-scale Llama-2 without LoRA, which contains around 7B trainable parameters, on the TOFU dataset. Our results, reported below and included in **Table 14 (Appendix H.5) of our revised paper**, show a **similar trend** to our results for LoRA-tuned LLM unlearning in Table 3:
> * While StoCuReNU does not achieve the best forgetting efficiency, as shown by a $D_e$ ROUGE gap of $0.162$ from retraining, it shows a better unlearning trade-off between forgetting and utility preservation than SOTA methods like SCRUB via higher ToW score.
> * The fact that the Original model achieves the best ToW score amongst all tested methods shows that most of them are likely insufficient for large-scale unlearning, leading to either worse forgetting efficiency (observed in GD, StoCuReNU) or worse utility preservation (observed in GA, GDiff, IDK, NPO, SCRUB). This challenging setting calls for the development of more robust unlearning methods in the future.
>
> Would you like to clarify which parts of the evaluation technique are unclear? We are willing to address them by improving our explanations in the revision.
>
> |Methods|$D_e$ ROUGE (→)|$D_r$ ROUGE (→)|$D_{test}$ ROUGE (→)|Truth Ratio (↑)|ToW (↑)|MIA AUC (→)|
> |--|:-:|:-:|:-:|:-:|:-:|:-:|
> |Retraining|0.399|0.672|0.861|0.695|1.000|88.93|
> |Original|0.692|**0.686**|$\underline{0.894}$|0.539|**0.674**|100.00|
> |GD|0.677|0.904|**0.891**|0.559|0.538|99.62|
> |GA|0.000|0.000|0.000|0.344|0.027|10.44|
> |GradDiff|0.032|0.207|0.198|0.484|0.114|6.68|
> |IDK|0.066|0.373|0.502|0.616|0.300|$\underline{99.20}$|
> |NPO|0.078|0.082|0.002|**0.805**|0.039|55.37|
> |SCRUB|**0.321**|0.404|0.611|$\underline{0.674}$|0.506|**79.21**|
> |StoCuReNU (ours)|$\underline{0.561}$|$\underline{0.535}$|0.695|0.547|$\underline{0.603}$|99.99|
>
> **References**
>
> Schulman, John and Thinking Machines Lab, "LoRA Without Regret", Thinking Machines Lab: Connectionism, Sep 2025.
>
> > I would have appreciated some analysis of the results in Figure 2. In particular, the experiment for unlearning on Llama-2 still has 40\% accuracy $D_e$ regardless of whether StoCuReNU or retraining is used, which suggests that the samples being targeted for unlearning are not unique within the training set. As a result, it is not clear if the experiments can be conclusive in this case.
>
> Thank you for the insightful question. While we agree that the experiments on sample-level unlearning cannot conclusively show that unlearning is successful, these experiments are meaningful as they can help us **identify cases where unlearning is unsuccessful** by revealing unintended unlearning behaviors, such as _over-forgetting_ (i.e., degrading unlearned model performance due to excessive forgetting) and _under-forgetting_ (i.e., retaining excessive influence of the unlearned samples). In Llama-2 experiments shown in Figure 2, we observe that GDiff, NPO, and GA achieve much lower $D_e$ and $D_r$ ROUGE than retraining, indicating over-forgetting effects. Conversely, $D_e$ ROUGE for SCRUB is close to that of the original model, suggesting the unlearned samples are under-forgotten. Among the tested methods, the trajectory of StoCuReNU stays closest to retraining on both $D_e$ ROUGE and $D_r$ ROUGE, avoiding over-forgetting and under-forgetting.
>
> Additionally, we note that **the class-level unlearning experiments** (Figure 2 (right), Tables 2-3) offer a **more conclusive evaluation** of unlearning success. For example, in the ResNet-18 experiment in Figure 2, $D_e$ accuracy is expected to drop to zero for the unlearned class. This is achieved by our method, but not by some of the baseline methods such as GD or GDiff.
>
> ---
>
> **[Continue below.]**

---

> > ### Author Response · Authors · 2025-11-26
> >
> > > I was also confused on how the authors obtained a retraining time measurement for Llama2-7B. If I understand this should be the time to retrain Llama2-7B from scratch minus the data in the dataset they want forgotten. However, it's more likely this is the time to fine-tune Llama2-7B on the dataset. Thus, it seems misleading to call this the retraining time for Llama2-7B since that is not how a fully exact retraining method would work.
> >
> > Thank you for your suggestion. Indeed, in the Llama-2-7B experiments, the "retraining" time refers to the time to fine-tune the model on the retained dataset. We used "retraining" to remain consistent with prior works (Shi et al., 2025.). However, we agree that it may be misleading, and have **clarified the definition of retraining for Llama-2 in our revised paper**.
> >
> > **References**
> >
> > Shi, W., et al. (2025). MUSE: Machine unlearning six-way evaluation for language models. In Proc. ICLR, 2025.
> >
> > ---
> >
> > Thank you again for reviewing our work. We hope that our clarifications and additional experiments will improve your evaluation of our paper. We would be happy to provide further clarification.

---

### Official Review · Reviewer_znzs · 2025-10-30

**Soundness:** 3
**Presentation:** 3
**Contribution:** 2
**Rating:** 4
**Confidence:** 4

**Summary:**

This paper investigates Newton-like unlearning algorithms to address the Hessian degeneracy challenge. It proposes two methods, CuReNU and StoCuReNU, based on cubic-regularized optimization and analyzes the convergence guarantee. StoCuReNU achieves performance comparable to state-of-the-art empirical unlearning methods across diverse settings. The authors demonstrate that StoCuReNU is scalable with comparable unlearning performance for various settings,including batchand sequential unlearning.

**Strengths:**

This paper investigates Newton-like unlearning algorithms to address the Hessian degeneracy challenge. It proposes two methods, CuReNU and StoCuReNU, based on cubic-regularized optimization. StoCuReNU achieves performance comparable to state-of-the-art empirical unlearning methods across diverse settings. Theoretic analysis of convergence guarantees seem solid.

**Weaknesses:**

1. This is not the first work to propose second-order unlearning for neural networks, so the contributions appear limited. For example, Qiao et al. (2025) and Zhang et al. (2024b) also introduce second-order unlearning algorithms for non-convex objectives. The paper therefore needs stronger motivation and a clearer articulation of its novel contributions.

2. The authors compare against only a subset of existing unlearning algorithms; for instance, Qiao et al. (2025) is not included. Moreover, in the experimental evaluations, the proposed method does not perform noticeably better than the baselines, which further limits the contributions of this work.

3. The experiments use only five unlearning rounds, averaged over three random runs, which seems quite limited.

4. Accuracy is not a very convincing metric to evaluate the unlearning performance, Other common evaluation methods, such as membership inference attacks (MIA), are not considered.

**Questions:**

1. Since this is not the first work to propose second-order unlearning for neural networks, the authors should restate the central question more clearly: How can we unlearn neural networks effectively using second-order methods?

2. StoCuReNU claims smaller space complexity than Qiao et al. (2025); how does its time complexity compare?

3. Can lower-complexity approaches be used to approximate or invert the Hessian in CuReNU, such as Hessian–vector products or related techniques?

4. Line 375: “Tug-of-War (ToW) score that aggregates these gaps (smaller is better)” — did you mean “larger is better”?

---

> ### Author Response · Authors · 2025-11-26
>
> Thank you for your comments on our work! We would like to address your concerns as follows.
>
> ---
>
> > This is not the first work to propose second-order unlearning for neural networks, so the contributions appear limited. For example, Qiao et al. (2025) and Zhang et al. (2024b) also introduce second-order unlearning algorithms for non-convex objectives. The paper therefore needs stronger motivation and a clearer articulation of its novel contributions.
>
>
> While our work is not the first to propose second-order unlearning for neural networks, we articulated in the introduction that our novel contributions lie in addressing the _limitations_ of existing second-order methods, such as Hessian degeneracy, inefficiency, and high memory usage (for Qiao et al, 2025). Particularly, our contribution, which involves investigating a fundamental issue of degenerate Hessians in well-trained neural networks and providing a principled remedy via cubic regularization, is useful for developing more advanced and effective second-order unlearning methods in the future.
>
> To further clarify the _novel_ contributions, we will add the following line at the end of the introduction: "The novelty lies in recognizing the potential of existing optimization methods to satisfy our desiderata, addressing limitations of second-order unlearning, and evaluating the methods extensively empirically."
>
> We are unsure whether Zhang et al. (2024b) is the intended citation, as their work does not propose second-order unlearning algorithms for non-convex objectives to our understanding.
>
> > Since this is not the first work to propose second-order unlearning for neural networks, the authors should restate the central question more clearly: How can we unlearn neural networks effectively using second-order methods?
>
> We are slightly unsure about this comment, as this is the question stated at the end of page 1. Could the reviewer clarify whether they are suggesting that the question be reformulated? Since our work seeks to address the limitations in the existing second-order methods, a possible restatement might be: "How can we unlearn neural networks effectively using second-order unlearning methods while avoiding problems faced by existing methods?"
>
> If the reviewer has a preferred phrasing, we would be happy to discuss and incorporate it.
>
> > The authors compare against only a subset of existing unlearning algorithms; for instance, Qiao et al. (2025) is not included.
>
> > StoCuReNU claims smaller space complexity than Qiao et al. (2025); how does its time complexity compare?
>
> Qiao et al. (2025) inherently incur **substantial computational overhead** due to the precomputation step that computes the Hessian-vector product for _every_ sample in the training set. When evaluating their method on FMNIST, the precomputation step takes around 468.3 hours (\~19.5 days) to run, which already far exceeds the time required for retraining (~61.20 seconds) and defeats the purpose of unlearning. We note that Qiao et al. (2025) originally tested on a subset of FMNIST with only 4k samples, whereas we use the full FMNIST dataset of 60k samples in our experiments.
>
> The prohibitively long precomputation time and the significant $O(nd)$ memory requirement make (Qiao et al., 2025) **impractical** in real-world settings. We have included this discussion of Qiao et al. (2025) in Appendix F.2 of our revised paper.
>
>
> > Moreover, in the experimental evaluations, the proposed method does not perform noticeably better than the baselines, which further limits the contributions of this work.
>
> We understand your concern. We would like to gently point out that while StoCuReNU is not always better than other baselines, it is consistently among the best or second-best performers across most metrics in Tables 2 and 3. In contrast, we also observe that no tested method, including strong baselines such as SCRUB and DELETE, consistently dominates others across all evaluation metrics. These observations indicate that StoCuReNU remains a viable and competitive method worth considering in various scenarios.
>
> Additionally, in Remark 6.1, we describe that our intention is not to claim new SOTA unlearning performance for StoCuReNU. Rather, **our main contribution** is to demonstrate a principled remedy for the Hessian degeneracy faced by existing second-order unlearning methods and to unlock their potential for unlearning large-scale neural networks. Our empirical results show that the proposed remedy substantially narrows the unlearning performance gap between second-order approaches (StoCuReNU) and first-order SOTA methods such as SCRUB and DELETE. We view this as an important step towards more effective second-order unlearning methods in the future.
>
> ---
> **[Continue below.]**

---

> ### Author Response · Authors · 2025-11-26
>
> > The experiments use only five unlearning rounds, averaged over three random runs, which seems quite limited.
>
> Firstly, we'd like to clarify that an extended sequential unlearning experiment with **10 rounds** on ResNet × CIFAR-10 is reported in our **Appendix H.2** (as mentioned in Sec. 6.3). StoCuReNU continues to approximate well the performance of retraining (Fig. 5), and achieves the best ToW score of 0.97 among all tested methods (Tab. 9). This indicates that StoCuReNU's effectiveness persists well beyond the five unlearning rounds.
>
> We report our results averaged over **5 random seeds** for Llama-2 × TOFU and ResNet-18 × CIFAR-10 as follows. These results can also be found in **Tab. 3 & Fig. 2 of our revised paper**. We find that increasing the number of runs generally does not alter our qualitative conclusions.
>
> |Methods|$D_e$ ROUGE ($\rightarrow$)|$D_r$ ROUGE ($\rightarrow$)|$D_{test}$ ROUGE ($\rightarrow$)|Truth Ratio ($\uparrow$)|ToW ($\uparrow$)|MIA AUC ($\rightarrow$)|
> |--|:-:|:-:|:-:|:-:|:-:|:-:|
> |Retraining|0.390 ± 0.004|0.573 ± 0.034|0.731 ± 0.023|0.658 ± 0.007|1.00 ± 0.00|87.82 ± 5.78|
> |Original|0.625 ± 0.003|**0.587 ± 0.008**|**0.716 ± 0.035**|0.508 ± 0.002|0.71 ± 0.03|100.00 ± 0.00|
> |GD|$\underline{0.510 ± 0.018}$|0.809 ± 0.019|0.625 ± 0.054|0.538 ± 0.011|0.60 ± 0.08|$\underline{99.81 ± 0.12}$|
> |GA|0.009 ± 0.017|0.009 ± 0.017|0.000 ± 0.000|0.571 ± 0.102|0.07 ± 0.01|26.68 ± 17.08|
> |GradDiff|0.000 ± 0.000|0.000 ± 0.000|0.000 ± 0.000|$\underline{0.808 ± 0.137}$|0.07 ± 0.01|40.25 ± 21.81|
> |IDK|0.098 ± 0.012|0.474 ± 0.013|0.683 ± 0.040|0.566 ± 0.015|0.60 ± 0.02|99.89 ± 0.08|
> |NPO|0.026 ± 0.030|0.028 ± 0.034|0.001 ± 0.002|**0.831 ± 0.043**|0.08 ± 0.02|**78.52 ± 10.23**|
> |SCRUB|0.539 ± 0.033|$\underline{0.542 ± 0.023}$|0.640 ± 0.066|0.512 ± 0.012|$\underline{0.72 ± 0.03}$|100.00 ± 0.00|
> |StoCuReNU (ours)|**0.455 ± 0.053**|0.484 ± 0.045|$\underline{0.706 ± 0.038}$|0.591 ± 0.043|**0.80 ± 0.03**|99.86 ± 0.13|
>
> |Methods|$D_e$ Acc ($\rightarrow$)|$D_r$ Acc ($\rightarrow$)|$D_{test}$ Acc ($\rightarrow$)|ToW ($\uparrow$)|JS Div. ($\downarrow$)|MIA AUC ($\rightarrow$)|
> |--|:-:|:-:|:-:|:-:|:-:|:-:|
> |Retraining|0.000 ± 0.000|91.173 ± 7.363|77.508 ± 3.628|1.000 ± 0.000|0.000 ± 0.000|50.69 ± 0.64|
> |Original|96.422 ± 2.883|90.237 ± 7.728|85.058 ± 3.526|0.033 ± 0.027|0.032 ± 0.004|49.68 ± 0.62|
> |Rand. Lbls.|$\underline{0.008 ± 0.018}$|17.065 ± 2.705|16.432 ± 2.806|0.106 ± 0.054|0.022 ± 0.01|**50.93 ± 0.67**|
> |DELETE|**0.000 ± 0.000**|77.188 ± 6.152|66.640 ± 6.297|0.775 ± 0.180|$\underline{0.015 ± 0.012}$|52.45 ± 1.09|
> |GD|93.400 ± 9.144|93.363 ± 5.199|87.806 ± 1.077|0.057 ± 0.079|0.030 ± 0.007|51.37 ± 0.84|
> |GA|5.384 ± 7.712|24.464 ± 2.996|22.180 ± 2.848|0.143 ± 0.049|0.027 ± 0.006|51.75 ± 0.66|
> |GradDiff|12.344 ± 11.709|77.071 ± 6.517|66.838 ± 6.070|0.669 ± 0.074|0.021 ± 0.009|51.47 ± 1.06|
> |NPO|7.144 ± 5.106|78.498 ± 10.037|67.576 ± 8.286|0.732 ± 0.086|0.023 ± 0.008|51.94 ± 1.01|
> |SCRUB|**0.000 ± 0.000**|**90.704 ± 3.350**|$\underline{78.168 ± 1.708}$|**0.944 ± 0.031**|0.017 ± 0.015|$\underline{50.16 ± 1.39}$|
> |StoCuReNU (ours)|2.320 ± 3.160|$\underline{90.332 ± 4.003}$|**77.590 ± 2.903**|$\underline{0.909 ± 0.050}$|**0.011 ± 0.009**|51.33 ± 1.26|
>
> > Accuracy is not a very convincing metric to evaluate the unlearning performance, Other common evaluation methods, such as membership inference attacks (MIA), are not considered.
>
> We would like to clarify that **Tables 2 and 3** include results based on non-accuracy unlearning metrics such as JS divergence and MIA. For example, our StoCuReNU can achieve an MIA AUC of 49.78, close to that of Retraining's 49.61 (Table 3). These metrics are introduced in the "evaluation metrics" paragraph of **Sec 6.1**, and our **Appendix G** contains additional MIA results on overfitted models.
>
> > Can lower-complexity approaches be used to approximate or invert the Hessian in CuReNU, such as Hessian–vector products or related techniques?
>
> While using Hessian-vector products to reduce the complexity of CuReNU is a reasonable suggestion, the method remains costly for large-scale models and datasets, as it requires computing these products across **the entire retained set**. StoCuReNU mitigates this issue by using **mini-batch updates** on the retained set while preserving CuReNU's theoretical convergence guarantee, making it an efficient method for large-scale unlearning.
>
> > Did you mean "larger is better" in line 375, "Tug-of-War (ToW) score that aggregates these gaps (smaller is better)"?
>
> Thank you for the correction. We have corrected it in our revised paper.
>
> ---
>
> We hope that our clarifications and additional results will help improve your evaluation of our paper. Thank you again for reviewing our work and providing insightful comments.

---

> > ### Comment · Reviewer_znzs · 2025-11-27
> > **Response to the authors**
> >
> > The authors have largely addressed most of my comments, and the contributions are now clearer to me. I have therefore raised the score.
> >
> > Regarding the following point from my previous comments:
> >
> > "Since this is not the first work to propose second-order unlearning for neural networks, the authors should restate the central question more clearly: How can we unlearn neural networks effectively using second-order methods?"
> >
> > What I meant was to refine the question a bit to better reflect the contribution of this paper. It would be helpful to be more specific about these issues, such as Hessian degeneracy in the revised version: "How can we unlearn neural networks effectively using second-order unlearning methods while avoiding problems faced by existing methods?"

---

> > > ### Author Response · Authors · 2025-11-28
> > >
> > > We thank the reviewer for the positive feedback.
> > >
> > > Following your suggestion, we have updated the introduction in the revised paper to explicitly state our research question as: "How can we effectively unlearn neural networks using second-order methods while addressing the issue of Hessian degeneracy?" We hope this revision clarifies the focus and motivation of our work.

---

### Official Review · Reviewer_9cot · 2025-10-31

**Soundness:** 4
**Presentation:** 4
**Contribution:** 4
**Rating:** 8
**Confidence:** 4

**Summary:**

This paper studies the machine unlearning problem and proposes cubic regularized method to handle the potential problem of Hessian degeneracy. The proposed method CuReNU introduces damping to stabilize the Newton's method and have a systematic way to avoid stability issues. The authors further propose a stochastic version StoCuReNU method to bypass the $O(d^3)$ involved in matrix inversion using efficient Hessian Vector Product computations. The numerical experiments also supports the method favorably.

**Strengths:**

Machine unlearning is a very timely topic and Hessian degeneracy can be serious issue for computation stability. The adaptation of tools from classical nonconvex optimization (cubic regularization) into ML context is very well motivated. The usage of second order information is something the community has often overlooked and it is great to see bring brought back to the stage. The paper is technically sound, clearly presented.

**Weaknesses:**

The CuReNU and StoCuReNU are both adapted from existing algorithm in different setting and hence the convergene and theoretical guarantees are inherited. The authors say that " this adaptation is both necessary and non-trivial to address failure modes". Please clarify further what exactly had to be modified.

While the theory shows favorable memory usage, the empirical results in Table 4 show STOCURENU's practical peak memory can be higher than its baselines. Is this due to a large constant factor (e.g., loading the base model, LoRA adapters, and HVP buffers) 1 and that the $\mathcal{O}(2d)$ benefit is about asymptotic scaling as the dataset size $n$ grows? Is the problem too small scale for the asymptotic to kick in?

In Appendix G, the authors test on overfitted models and note that while STOCURENU is effective, a performance gap remains in Membership Inference Attack (MIA) mitigation compared to the SOTA empirical method, SCRUB. Please discuss this further. Is this gap expected for other scenarios as well or in general there is a gap?

**Questions:**

In Appendix J.3.1 the authors  show that $T$ represents a direct trade-off between computational efficiency and unlearning effectiveness. Could you please provide a principled heuristic in selecting $T$? Or maybe an early stopping criterion?

---

> ### Author Response · Authors · 2025-11-26
>
> > The authors say that " this adaptation is both necessary and non-trivial to address failure modes". Please clarify further what exactly had to be modified.
>
> Our modification of existing algorithms for unlearning involves: initializing from the fully trained neural network (instead of random initialization) and defining the unlearning objective based on the retained dataset (rather than the full training set). While other algorithmic details remain the same, we believe our adaptation is necessary as it addresses the issue of Hessian degeneracy in a principled manner, and the benefit of our adaptation is non-trivial as it unlocks the potential effectiveness of existing second-order unlearning methods for unlearning neural networks. We would like to note that instead of the algorithm, "non-trivial" refers to our main contributions of identifying the fundamental "failure modes" of second-order unlearning (Hessian degeneracy), proposing a principled remedy via cubic regularization, and extensively evaluating its effectiveness in real-world settings. This contribution is useful for developing a more effective second-order unlearning in the future.
>
>
> > While the theory shows favorable memory usage, the empirical results in Table 4 show STOCURENU's practical peak memory can be higher than its baselines. Is this due to a large constant factor (e.g., loading the base model, LoRA adapters, and HVP buffers) 1 and that the $O(2d)$ benefit is about asymptotic scaling as the dataset size grows? Is the problem too small scale for the asymptotic to kick in?
>
> - Table 4 does not show the memory benefit of StoCuReNU because we are comparing against SOTA first-order methods with inherent lower memory requirements. Particularly, the baselines (Retraining, SCRUB, DELETE) only require around $O(d)$ memory to compute the gradient, while StoCuReNU requires $O(2d)$ memory to compute both the gradient and the HVP, regardless of the dataset sizes. Despite higher memory usage, we showed that StoCuReNU can achieve better unlearning performance than DELETE, while being comparable and sometimes better than SCRUB in Tables 2 and 3.
> - We note that the theoretical benefit of $O(2d)$ memory arises when compared StoCuReNU with existing second-order methods, which require $O(d^2)$ memory for storing the full Hessian matrix in the vanilla Newton unlearning and CuReNU, and $O(nd)$ memory in (Qiao et al., 2025), where $n$ being the dataset sizes.
>
> > In Appendix G, the authors test on overfitted models and note that while STOCURENU is effective, a performance gap remains in Membership Inference Attack (MIA) mitigation compared to the SOTA empirical method, SCRUB. Please discuss this further. Is this gap expected for other scenarios as well or in general there is a gap?
>
> Thank you for the question. Firstly, we note that while present, the MIA gap between StoCuReNU and SCRUB is **relatively small** (<2\% AUC), as shown in Tables 6-7 (Appendix G).
>
> Furthermore, in an additional MIA experiment on an overfitted CNN × FMNIST model, we observe that StoCuReNU achieves an **MIA AUC closer to retraining** than the SOTA empirical method SCRUB. The corresponding results are shown below and in Table 8 (Appendix G) of our revised paper. This suggests that the observed MIA gap is likely scenario-dependent, and StoCuReNU can achieve better MIA results than SOTA methods like SCRUB under certain scenarios.
>
> | Method | MIA AUC (→) |
> | :-: | :-: |
> | Retraining | 54.04 ± 0.56 |
> | Original | $\underline{53.90 ± 0.81}$ |
> | Rand. Lbls. | 50.04 ± 0.48 |
> | DELETE | 50.84 ± 1.35 |
> | GD | **53.99 ± 0.37** |
> | GA | 51.34 ± 0.75 |
> | GDiff | 52.11 ± 1.44 |
> | NPO | 51.27 ± 0.67 |
> | SCRUB | 48.33 ± 3.49 |
> | StoCuReNU (ours) | 51.20 ± 0.18 |
>
> > In Appendix J.3.1 the authors show that $T$ represents a direct trade-off between computational efficiency and unlearning effectiveness. Could you please provide a principled heuristic in selecting $T$? Or maybe an early stopping criterion?
>
> A principled heuristic in selecting $T$ can start by selecting a small $T$ (e.g., $T = 10-20$ as in our experiments) and gradually increasing $T$ if the unlearning performance is not yet satisfactory and if computational resources allow. We note that this process can be economical because the unlearned model in iteration $T-1$ can serve as initialization for iteration $T$, ensuring that early computations are not wasted.
>
> A reasonable early stopping criterion can be based on non-retraining metrics, such as MIA for forgetting efficacy, combined with a thresholded loss on the retained set to ensure that utility is preserved.
>
> ---
>
> We sincerely thank the reviewer for your time and effort in evaluating our work. We hope that our responses and clarifications have addressed your concerns and helped improve your opinion of the paper. If any questions remain, we would be happy to provide further clarification.

---

### Official Review · Reviewer_mMKS · 2025-11-01

**Soundness:** 3
**Presentation:** 3
**Contribution:** 2
**Rating:** 6
**Confidence:** 3

**Summary:**

This paper studies the challenge of machine unlearning i.e. removing the influence of specific training data from a trained neural network without retraining from scratch. The authors focus on improving second-order unlearning methods such as Newton Unlearning, which rely on Hessian information to approximate retraining. They identify a critical limitation: Hessian degeneracy (presence of many small, zero or negative eigenvalues) in trained neural networks, which leads to unstable or divergent updates during unlearning. To overcome this, the paper proposes two novel algorithms: CuReNU (Cubic-Regularized Newton’s Unlearning) – uses cubic regularization to automatically determine an appropriate damping factor, thereby stabilizing updates and it's stochastic variant StoCuReNU (Stochastic CuReNU) – a scalable, Hessian-free variant using Hessian-vector products (HVPs). Empirically, on FashionMNIST, CIFAR-10, AG-News, and TOFU datasets, CuReNU and StoCuReNU achieve competitive unlearning performance compared to state-of-the-art empirical methods (e.g., SCRUB, DELETE) in both batch and sequential unlearning settings.

**Strengths:**

1. The paper is very well-written, including notational consistency, and is very easy to understand.
2. The problem formulation is very clean and I believe the authors point towards and study an important problem.
3. The proposed CuReNU and StoCuReNU are derived from established optimization theory. The adaptation for unlearning is technically sound and mathematically well-justified.
4. Experiments span both batch and sequential unlearning on diverse datasets (vision, text). Metrics (accuracy, JS divergence, ToW score, etc.) are carefully chosen and clearly reported.
5. The figures and tables are clear and informative.

**Weaknesses:**

1. The proposed methods are direct adaptations of known optimization techniques. The novelty lies primarily in contextual application rather than new algorithms.
2. The main results focus on small to medium models (CNN, ResNet-18, LoRA-tuned LLaMA-2). Full-scale LLM unlearning remains untested.
3. Ablation studies that isolate the impact of cubic regularization vs. stochasticity are missing.
4. Discussion around more efficient Hessian vector product is missing.
5. While there are a lot of results that are included, the proposed methods are not always a clear winner.

**Questions:**

1. How sensitive are CuReNU and StoCuReNU to the cubic regularization coefficient 𝐿?
2. Can the authors provide empirical Hessian spectra for larger models (e.g., LLaMA-2 layers) to substantiate degeneracy at scale?
3. How does the unlearning error accumulate across rounds? Is there a mechanism to “recalibrate” the model to prevent drift after many sequential unlearning steps?
4. For large foundation models with mixed data sources, how feasible is StoCuReNU in practice?
5. Can authors include runtime comparisons at least for a subset of experiments w.r.t. baselienes?

Any discussion around these will help improve the paper over it's current state.

---

> ### Author Response · Authors · 2025-11-26
>
> Thank you for your helpful review of our work! We will address your comments below.
>
> ---
>
> > The novelty lies primarily in contextual application rather than new algorithms.
>
> Although the novelty lies primarily in contextual application, our paper still makes a significant contribution of identifying and resolving the key limitation that hinders the effectiveness of existing second order machine unlearning approaches. Specifically, we study the fundamental issue of degenerate Hessians in well-trained neural networks and provide a principled remedy via cubic regularization. This study is useful for developing more advanced and effective second-order unlearning methods in the future and can be impactful even without inventing new algorithms.
>
> > The main results focus on small to medium models (CNN, ResNet-18, LoRA-tuned LLaMA-2). Full-scale LLM unlearning remains untested.
>
> Our main results on Llama-2 focus on the LoRA-tuned setting because LoRA is now a **widely adopted** approach for fine-tuning LLMs due to its efficiency and competitive effectiveness to full-scale fine-tuning (Schulman \& Thinking Machines Lab, 2025). Furthermore, fine-tuning datasets are often task-specific datasets that are more privacy-sensitive than the public datasets used for full-scale LLM training, making them particularly relevant for unlearning.
>
> Nonetheless, we conducted an additional experiment to evaluate our StoCuReNU and other baseline methods on full-scale Llama-2 without LoRA, which contains around 7B trainable parameters, on the TOFU dataset. Our results, reported below and included in **Table 14 (Appendix H.5) of our revised paper**, show a **similar trend** to our results for LoRA-tuned LLM unlearning in Table 3:
> * While StoCuReNU does not achieve the best forgetting efficiency, as shown by a $D_e$ ROUGE gap of $0.162$ from retraining, it shows a better unlearning trade-off between forgetting and utility preservation than SOTA methods like SCRUB via higher ToW score.
> * The fact that the Original model achieves the best ToW score amongst all tested methods shows that most of them are likely insufficient for large-scale unlearning, leading to either worse forgetting efficiency (observed in GD, StoCuReNU) or worse utility preservation (observed in GA, GDiff, IDK, NPO, SCRUB). This challenging setting calls for the development of more robust unlearning methods in the future.
>
> |Methods|$D_e$ ROUGE (→)|$D_r$ ROUGE (→)|$D_{test}$ ROUGE (→)|Truth Ratio (↑)|ToW (↑)|MIA AUC (→)|
> |--|:-:|:-:|:-:|:-:|:-:|:-:|
> |Retraining|0.399|0.672|0.861|0.695|1.000|88.93|
> |Original|0.692|**0.686**|$\underline{0.894}$|0.539|**0.674**|100.00|
> |GD|0.677|0.904|**0.891**|0.559|0.538|99.62|
> |GA|0.000|0.000|0.000|0.344|0.027|10.44|
> |GradDiff|0.032|0.207|0.198|0.484|0.114|6.68|
> |IDK|0.066|0.373|0.502|0.616|0.300|$\underline{99.20}$|
> |NPO|0.078|0.082|0.002|**0.805**|0.039|55.37|
> |SCRUB|**0.321**|0.404|0.611|$\underline{0.674}$|0.506|**79.21**|
> |StoCuReNU(ours)|$\underline{0.561}$|$\underline{0.535}$|0.695|0.547|$\underline{0.603}$|99.99|
>
> **References**
>
> Schulman, John and Thinking Machines Lab, "LoRA Without Regret", Thinking Machines Lab: Connectionism, Sep 2025.
>
> > Discussion around more efficient Hessian vector product is missing.
>
> In **Appendix F.3 of our revised paper**, we include a discussion of efficient Hessian-vector product computation. While relevant, we note that HVP computation is complementary to our work, and the efficiency of StoCuReNU would further improve with more efficient HVP computations.
>
> > The proposed methods are not always a clear winner.
>
> We understand your concern. We would like to gently point out that while StoCuReNU is not always a clear winner, it is consistently among the best or second-best performers across most metrics in our Tables 2 and 3. Meanwhile, we also observe that **no tested method, including strong baselines such as SCRUB and DELETE, consistently dominates others** across all evaluation metrics. These observations indicate that StoCuReNU remains a viable and competitive method worth considering in various scenarios.
>
> Additionally, in Remark 6.1, we describe that **our intention is not to claim new SOTA unlearning performance for StoCuReNU**. Rather, we seek to demonstrate a principled remedy for the Hessian degeneracy faced by existing second-order unlearning methods and to unlock their potential for unlearning in the age of large-scale neural networks. We view this as a useful step towards more effective unlearning methods in the future.
>
> > Can authors include runtime comparisons at least for a subset of experiments w.r.t. baselienes?
>
> Thank you for your suggestion. We have provided the runtime comparisons in our current paper between our methods and some best-performing baseline methods in **Sec. 6.4 (Table 4)**, and the full comparisons against all baselines in **Appendix I** (Table 13 of our current paper, corresponding to Table 15 of our revised paper).
>
> ---
> **[Continued below.]**

---

> ### Author Response · Authors · 2025-11-26
>
> > Ablation studies that isolate the impact of cubic regularization vs. stochasticity are missing.
>
> Since the meaning of "impact of cubic regularization **vs.** stochasticity" is not clear to us, we assume that the reviewer is referring to two separate ablation studies: one on cubic regularization and one on stochasticity.
>
> - **Ablation on cubic regularization**: This experiment is incorporated in our Table 2, where we compare Newton unlearning with cubic regularization (CuReNU) and without cubic regularization (PINV-Newton and Damped Newton) on CNN × FMNIST. Our results show that the absence of cubic regularization results in poor unlearning performance, as evidenced by significantly low $D_r$ Acc., $D_{test}$ Acc. and ToW for PINV-Newton and Damped Newton.
> - **Ablation on stochasticity**: We provide additional analysis of the stochasticity in terms of batch size (BS) in **Appendix J.4 of our revised paper**, with the main results shown below.
>
> |BS|$D_e$ Acc. (→)|$D_r$ Acc. (→)|$D_{test}$ Acc. (→)|
> |:-:|:-:|:-:|:-:|
> |32|1.839 ± 2.899|87.445 ± 1.585|76.400 ± 1.471|
> |64|1.322 ± 1.002|88.863 ± 2.724|77.683 ± 2.711|
> |128|1.344 ± 1.078|89.954 ± 1.604|78.958 ± 1.388|
> |256|1.383 ± 2.073|90.383 ± 2.084|79.138 ± 1.426|
> |512|4.544 ± 7.513|91.042 ± 1.747|79.950 ± 0.850|
> |1024|0.133 ± 0.217|89.820 ± 3.593|78.321 ± 3.157|
> |2048|0.150 ± 0.246|89.958 ± 3.461|78.433 ± 3.043|
> |$\|D_r\|$|0.206 ± 0.178|90.018 ± 3.360|78.542 ± 2.860|
> |Retraining|0.000 ± 0.000|92.466 ± 0.883|80.225 ± 0.963|
>
> Our results show that StoCuReNU with mini-batch sampling can achieve comparable performance (within 2-4\% difference on $D_e/D_r/D_{test}$ Acc.) to the full-batch setting (i.e, BS = $|D_r|$), suggesting that the sampling-induced stochasticity might have minimal impact on unlearning effectiveness.
>
> - Additionally, our **Appendix J.3** shows that increasing the number of stochastic iterations in StoCuReNU leads to an improvement in unlearning performance, and the optimal unlearning results can be achieved early within 10-20 iterations.
>
> We are happy to discuss further if the reviewer has other ablation studies in mind.
>
> > How sensitive are CuReNU and StoCuReNU to the cubic regularization coefficient $L$?
>
> As shown in **Appendix J.1**, CuReNU and StoCuReNU maintain competitive unlearning performance to retraining across various cubic regularization coefficients $L$, except when $L$ becomes too small (i.e., $L < 70$ for CuReNU). Since $L$ serves as an upper bound on the Hessian difference between any two model weights (Assumption 3.3), this result is expected as such a small $L$ would lead to an invalid upper bound.
>
> > Can the authors provide empirical Hessian spectra for larger models (e.g., LLaMA-2 layers) to substantiate degeneracy at scale?
>
> Thank you for your suggestion. We provide empirical Hessian spectra for Llama-2 × TOFU in **Figure 1 of our revised paper**. Our results show that the Hessian degeneracy is not limited to small-scale models but also persists in large-scale settings.
>
> > How does the unlearning error accumulate across rounds? Is there a mechanism to “recalibrate” the model to prevent drift after many sequential unlearning steps?
>
> Unlearning errors can accumulate across rounds because each unlearning round starts from the model obtained in the previous round. If an earlier round introduces approximation errors, such as _over-forgetting_ (i.e., the utility of the unlearned model is degraded) or _under-forgetting_ (i.e., excessive influence of the unlearned data is retained), these errors can propagate forward in next rounds, worsening the model utility and/or accumulating the residual influence of the unlearned data.
>
> To "recalibrate" the model after many sequential unlearning steps, one may eventually need to retrain the model to fully mitigate approximation errors. From Fig. 2, methods such as GA, NPO, DELETE, Rand. Lbls., and IDK, may require such recalibration after every unlearning round due to their significant deviation from retraining. In contrast, the gap between StoCuReNU and retraining remains small across multiple rounds, **reducing the need for frequent recalibration** via retraining.
>
> > For large foundation models with mixed data sources, how feasible is StoCuReNU in practice?
>
> For large foundation models trained on very large datasets with mixed data sources, the retained set might be very large and hard to access. Therefore, it would become **a challenge for all unlearning methods** that depend on the retained set, including StoCuReNU. One practical solution to make StoCuReNU (and other unlearning methods) feasible in this case is to consider **a smaller core set** that approximates the distribution of the large retained set and to perform unlearning using the core sets instead of the large data sources.
>
> ---
>
> Thank you again for reviewing our work. We hope that our clarifications and additional empirical results will improve your opinion of our work. We would be happy to provide further clarification.

---

### Author Response · Authors · 2025-12-03
**Summary of Reviews (Part 1/2)**

Dear Area Chairs,

To conclude the discussion phase, we summarize the reviewers' feedback and our responses as follows.

As of November 27, one reviewer has responded and [raised their score](#:~:text=I%20have%20therefore%20raised%20the%20score); overall, all reviewers have given our paper a positive rating ($\geq 6$).

**Highlighted Strengths:**

- The reviewers noted that our paper is **well written, clearly explained, and technically sound** ([mMKS](#:~:text=very%20well,to%20understand), [9cot](#:~:text=The%20paper%20is%20technically%20sound%2C%20clearly%20presented), [wc7S](#:~:text=the%20authors%20clearly%20explain)), with all reviewers rating the presentation between `3: good` and `4: excellent`.
- The studied problem of unlearning neural networks is **well formulated, timely, and important** ([mMKS](#:~:text=The%20problem,important%20problem), [9cot](#:~:text=Machine%20unlearning%20is%20a,computation%20stability), [wc7S](#:~:text=Unlearning%20is%20an%20important%20topic)).
- Our methods (CuReNU \& StoCuReNU) are **well motivated**, **technically sound**, and **mathematically justified** with **solid theoretical analysis** ([mMKS](#:~:text=The%20adaptation,justified.), [9cot](#:~:text=The%20adaptation%20of%20tools,very%20well%20motivated.), [znzs](#:~:text=Theoretic,seem%20solid.)).
- The empirical evaluation is **clear and diverse** across unlearning settings and datasets ([mMKS](#:~:text=Experiments%20span,reported.), [znzs](#:~:text=across%20diverse%20settings.%20The%20authors%20demonstrate%20that%20StoCuReNU%20is%20scalable%20with%20comparable%20unlearning%20performance%20for%20various%20settings%2Cincluding%20batchand%20sequential%20unlearning.), [wc7S](#:~:text=the%20authors%20clearly,proposal.)).

---
**[Continued below].**

---

> ### Author Response · Authors · 2025-12-03
> **Summary of Reviews (Part 2/2)**
>
> **Reviewers' Concerns & Our Response:**
>
> - _Novelty and Significance of our methods_ ([mMKS](#:~:text=known%20optimization%20techniques.-,The%20novelty%20lies,new%20algorithms.,-The%20main%20results), [9cot](#:~:text=The%20CuReNU,different%20setting), [znzs](#:~:text=not%20the,neural%20networks)): While our methods build on existing optimization algorithms, we have **already articulated** in our introduction and **added the following line** in our revised paper to further clarify our novel contributions: "The novelty lies in recognizing the potential of existing optimization methods to satisfy our desiderata, addressing limitations of second-order unlearning, and evaluating the methods extensively empirically." Reviewer [znzs](#:~:text=the%20contributions%20are%20now%20clearer%20to%20me) positively responded that "the contributions are now clearer" to them.
> - _Performance of StoCuReNU is not always SOTA_ ([mMKS](#:~:text=the%20proposed%20methods%20are%20not%20always%20a%20clear%20winner), [znzs](#:~:text=the%20proposed%20method%20does,baselines)): We noted that StoCuReNU consistently ranks among the best-performing methods, and no single method consistently dominates others across all metrics. Importantly, we clarified that our objective is not to claim new SOTA results, **as stated in Remark 6.1**. Instead, we aim to present a principled solution to the Hessian degeneracy issue in second-order unlearning, which is shown to significantly narrow the performance gap with SOTA methods such as SCRUB and DELETE (Tabs. 2-3 \& Fig. 2). Reviewer [znzs](#:~:text=The%20authors%20have,my%20comments) noted that our response has largely addressed their concerns.
> - _Comparison to Qiao et al. (2025)_ ([znzs](#:~:text=StoCuReNU%20claims,complexity%20compare%3F,-Can%20lower%2Dcomplexity)): We attempted to [empirically evaluate](#:~:text=When%20evaluating%20their,experiments.) the Hessian-free method proposed by Qiao et al. (2025) on FMNIST. However, unlike StoCuReNU, we found that their method is impractical in real-world settings due to prohibitively long precomputation time (~19.5 days) and substantial memory to compute and store Hessian-vector products for every sample in the training set. We have included this comparison in **Appendix F.2** of our revised paper.
> - _Additional experiments suggested by the reviewers_:
>     - Regarding the suggestion on full-scale LLM unlearning ([mMKS](#:~:text=Full%2Dscale%20LLM%20unlearning%20remains%20untested.), [wc7S](#:~:text=The%20evaluation%20is,needed)), we have clarified that using fine-tuning/LoRA is standard as in existing works. Nonetheless, we also included an additional experiment on [full-scale LLM unlearning](#:~:text=we%20conducted,future) in Appendix H.5. This addressed the only weakness noted by Reviewer wc7S.
>     - We have incorporated additional experiments, including [ablation on stochasticity of StoCuReNU](#:~:text=We%20provide,below.) suggested by Reviewer mMKS in Appendix J.4 and [main experiment results with 5 random seeds](#:~:text=We%20report,conclusions.) suggested by Reviewer znzs in Tab. 3 \& Fig. 2.
> - _Broader discussion_: We have engaged in discussion about [Hessian-vector product computation](#:~:text=In%20Appendix%20F.3%20of%20our%20revised%20paper%2C%20we%20include%20a%20discussion%20of%20efficient%20Hessian%2Dvector%20product%20computation.%20While%20relevant%2C%20we%20note%20that%20HVP%20computation%20is%20complementary%20to%20our%20work%2C%20and%20the%20efficiency%20of%20StoCuReNU%20would%20further%20improve%20with%20more%20efficient%20HVP%20computations.) (mMKS; included in Appendix F.3), ["recalibration" method for sequential unlearning](#:~:text=To%20%22recalibrate%22%20the%20model%20after,for%20frequent%20recalibration%20via%20retraining.) (mMKS), [unlearning with large datasets](#:~:text=For%20large%20foundation%20models%20trained,of%20the%20large%20data%20sources.) (mMKS), and [heuristics for selecting number of stochastic iterations in StoCuReNU](#:~:text=early%20stopping%20criterion%3F-,A%20principled%20heuristic%20in%20selecting,loss%20on%20the%20retained%20set%20to%20ensure%20that%20utility%20is%20preserved.) (9cot).
> - _Minor suggestions \& corrections_: We have added evidence for [Hessian degeneracy in large models](#:~:text=We%20provide%20empirical,paper.) (mMKS; included in Fig. 1), [clarified retraining definition in Llama-2 experiment](#:~:text=the%20%22retraining%22%20time,dataset) (wc7S), and [corrected typos](#:~:text=(smaller%20is%20better),better) in our revised paper.
>
> ---
>
> We greatly appreciate all reviewers' and the area chair's time, efforts, and insightful feedback, which have helped us further strengthen our work.
>
> Regards,
>
> The Authors.

---

### Meta-Review · Area_Chair_BDNk · 2025-12-24

**Summary:**

This paper addresses the problem of machine unlearning in neural networks. While second-order unlearning methods based on Newton updates are theoretically appealing because they closely approximate retraining, the authors show that these methods fail in practice for neural networks due to  Hessian degeneracy the prevalence of zero or negative eigenvalues near trained optima. This degeneracy causes standard Newton unlearning, as well as common fixes like Hessian pseudo-inverses to produce large updates. As a result, existing second-order approaches often underperform or diverge when applied to modern deep models. To overcome this limitation, the paper proposes two new algorithms: Cubic-Regularized Newton Unlearning (CuReNU) and its scalable variant  Stochastic CuReNU (StoCuReNU) . These methods adapt cubic regularization from non-convex optimization to automatically determine an appropriate Hessian damping factor, thereby stabilizing updates in the presence of degenerate Hessians. Extensive experiments on image classification and language modeling tasks demonstrate that the proposed methods substantially improve unlearning performance. The reviewers like the work for its clarity, technical soundness, and relevance. The proposed methods are grounded in established nonconvex optimization theory and their adaptation to unlearning is justified and well presented.

**Reviewer Concerns:**

The rebuttal did address several concerns but some that remain include the fact that the paper’s novelty is limited in terms of algorithmic innovation beyond instantiating known cubic-regularized methods for the objective. The fact that the empirical results do not cleanly present a mechanistic understanding of what to do in different regimes and why suggest there is more to explore on this front (this shows up as different methods being better in different regimes without a clear underlying rationale tying the work together).

**Reviewer Scores:**

znzs raised their score and the remaining had scores >=6. I think overall there was consensus this paper should be accepted.

---

### Decision · Program_Chairs · 2026-01-26

Accept (Poster)